# Class 3 PI3K coactivates the circadian clock to promote rhythmic de novo purine synthesis

Chantal Alkhoury[1,2,3,16], Nathaniel F. Henneman[1,2,3,16], Volodymyr Petrenko[4,5,6,7], Yui Shibayama[1,2,3], Arianna Segaloni[1,2,3], Alexis Gadault[1,2,3], Ivan Nemazanyy[8], Edouard Le Guillou[1,2,3], Amare Desalegn Wolide[9,10,11], Konstantina Antoniadou[1,2,3], Xin Tong[12], Teruya Tamaru[13], Takeaki Ozawa[14], Muriel Girard[1,2,3], Karim Hnia[15], Dominik Lutter[9,11], Charna Dibner[4,5,6,7] & Ganna Panasyuk[1,2,3] ✉

Metabolic demands fluctuate rhythmically and rely on coordination between the circadian clock and nutrient-sensing signalling pathways, yet mechanisms of their interaction remain not fully understood. Surprisingly, we find that class 3 phosphatidylinositol-3-kinase (PI3K), known best for its essential role as a lipid kinase in endocytosis and lysosomal degradation by autophagy, has an overlooked nuclear function in gene transcription as a coactivator of the heterodimeric transcription factor and circadian driver Bmal1–Clock. Canonical pro-catabolic functions of class 3 PI3K in trafficking rely on the indispensable complex between the lipid kinase Vps34 and regulatory subunit Vps15. We demonstrate that although both subunits of class 3 PI3K interact with RNA polymerase II and co-localize with active transcription sites, exclusive loss of Vps15 in cells blunts the transcriptional activity of Bmal1–Clock. Thus, we establish non-redundancy between nuclear Vps34 and Vps15, reflected by the persistent nuclear pool of Vps15 in Vps34-depleted cells and the ability of Vps15 to coactivate Bmal1–Clock independently of its complex with Vps34. In physiology we find that Vps15 is required for metabolic rhythmicity in liver and, unexpectedly, it promotes pro-anabolic de novo purine nucleotide synthesis. We show that Vps15 activates the transcription of *Ppat*, a key enzyme for the production of inosine monophosphate, a central metabolic intermediate for purine synthesis. Finally, we demonstrate that in fasting, which represses clock transcriptional activity, Vps15 levels are decreased on the promoters of Bmal1 targets, *Nr1d1* and *Ppat*. Our findings open avenues for establishing the complexity for nuclear class 3 PI3K signalling for temporal regulation of energy homeostasis.

Metabolic stress from nutrient fluctuations is anticipated and resolved through coordinated activity of the circadian-clock and nutrient-sensing signalling pathways[1]. The circadian clock functions as a transcriptional–translational molecular pacemaker, generating metabolic oscillations of approximately 24 h. In mammals the central clock in the suprachiasmatic nuclei of the hypothalamus synchronizes peripheral clocks in virtually all organs through neural and humoral signals[2]. In a coordinated manner, peripheral clocks, like in liver, are sensitive to entrainment by nutrients to generate cycling metabolic outputs[3,4]. The molecular clock is driven by a heterodimer of two transcription

---

factors, Bmal1 and Clock, that bind E-box motifs on gene promoters to initiate expression of its own repressors, period and cryptochrome (Per1–3, Cry1 and Cry2) as well as Rev-Erb (Rev-Erbα and Rev-Erbβ)[5]. Feeding-activated nutrient-sensing signalling pathways regulate the clock at a post-translational level. Numerous phosphorylation events driven by pro-anabolic mTOR–class 1 phosphatidylinositol-3 kinase (PI3K) signalling inhibit Bmal1–Clock by promoting complex destabilization, nuclear exclusion and non-transcriptional functions[6–8]. They also affect the localization and stability of other clock components[9,10]. Conversely, pro-catabolic AMPK protein kinase potentiates the transcriptional activity of Bmal1 by promoting proteasomal degradation of Cry1 and Per2 (refs. [11,12]). Furthermore, lysosomal degradation by autophagy of Cry1 and chaperone-mediated autophagy of clock components was reported[13,14]. However, none of these aforementioned signalling components act on chromatin to modify the activity of the Bmal1–Clock transcriptional complex. Moreover, although the clock drives metabolic rhythmicity especially in liver[1], the mechanisms of how specific metabolic pathways are regulated are still poorly understood. A substantial portion of these nutrient-sensitive metabolic cascades are driven by the lipid second messengers phosphoinositides. They orchestrate cellular trafficking through different cellular compartments to ensure nutrient uptake and metabolism[15]. However, the mechanistic links between clock-driven metabolic rhythmicity and phosphoinositide signalling remain poorly explored. Notably, class 3 PI3K is an essential component of nutrient-sensing networks acting both downstream and upstream of mTOR–class 1 PI3K and AMPK signalling as well as activating lysosomal degradation by autophagy[16–22]. It functions as a complex of lipid kinase Vps34 and its indispensable regulatory subunit Vps15, which is required for Vps34 stability, localization and lipid kinase activity[23–25]. This is highlighted by the Vps34 loss-of-function phenotypes of Vps15 mutants from yeast to mammals and the cryo-electron microscopy-determined structure of the complex[23–26]. Class 3 PI3K is the major source of PI3P, a lipid-membrane second messenger in multiple steps of endocytosis and autophagy[16,27]. Reports of circadian lysosomal activity and autophagy, two processes downstream of class 3 PI3K, drove us to explore the functional interaction of class 3 PI3K and the molecular clock[28,29].

## Results

### Class 3 PI3K controls the liver clock

To test whether class 3 PI3K impacts the circadian clock, we focused on the liver as Bmal1–Clock activity is highly sensitive to nutrients[3,30]. We used class 3 PI3K-mutant mice bearing a liver-specific chronic deletion of *Vps15* (*AlbCre*[+];*Vps15*[f/f] mice are hereafter referred to as Vps15LKO)[31,32]. In line with the essential function of Vps15 in the class 3 PI3K complex, its deletion leads to depletion of Vps34 and subsequent autophagy block, as witnessed by p62 protein accumulation (Extended Data Fig. 1a)[32]. Around-the-clock transcript and protein expression analyses

showed profound deregulation of the entire clock machinery in the Vps15LKO mice (Fig. 1a,b and Extended Data Fig. 1b,c). The transcript levels as well as circadian amplitudes of two bona fide Bmal1 targets, *Nr1d1* (gene coding for Rev-Erbα) and *Dbp*, were decreased (Fig. 1a, Extended Data Fig. 1b and Supplementary Table 1)[33,34]. These were mirrored at the protein level, demonstrating hardly detectable Rev-Erbα expression, whereas in the dark phase Bmal1 and Clock were decreased in the livers of Vps15LKO mice (Fig. 1b and Extended Data Fig. 1c). To comprehend the breadth of potential transcriptional dysfunction of the circadian clock, we performed Bmal1 chromatin immunoprecipitation with sequencing (ChIP–Seq) in Vps15LKO livers collected at Zeitgeber time 6 (ZT6), a zenith of Bmal1 transcriptional activity[35]. We coupled this with ChIP–Seq of acetylated K27 of histone H3 (H3K27Ac), a mark of transcriptionally active chromatin[36]. The Bmal1 ChIP–Seq revealed a global decrease of its enrichment at transcription start sites (TSS) in the livers of Vps15LKO mice (Fig. 1c,d), which was validated by its lower occupancy on E-box regions in the promoters of *Nr1d1* and *Dbp* (Fig. 1e and Extended Data Fig. 1d). The H3K27Ac ChIP–Seq demonstrated that among 7,256 differentially enriched peaks in the livers of Vps15LKO mice, the vast majority (7,228) were significantly decreased (Fig. 1d). Moreover, reduced enrichment of both Bmal1 and H3K27Ac was observed in 54% (539 of 999) of the peaks identified in samples of Vps15LKO mouse livers (Fig. 1f). Gene ontology (GO)-pathway analyses of biological processes for genes in which enrichment of Bmal1 and H3K27Ac was reduced in Vps15LKO mice revealed circadian rhythm and metabolic responses (catabolic processes, lipid metabolism, purine and nucleotide metabolism) potentially downstream of class 3 PI3K (Fig. 1f and Supplementary Table 2). In line with this inhibition of Bmal1 transcription in the livers of Vps15 mutants, the levels of soluble nuclear as well as cytosolic Bmal1, and subsequently its nuclear Bmal1–Clock complex, were reduced both at its peak (ZT0) and trough (ZT12; Extended Data Fig. 1e,f). Consistent with the biochemical findings, immunohistological analyses of fixed liver tissue collected at ZT0 showed lower nuclear levels of Bmal1 in the Vps15LKO mice (Extended Data Fig. 1g,h). Notably, differences in detection of cytosolic Bmal1 in fractionation and histological analyses are likely to be due to detergent-free extraction and crosslinking, respectively. Together, these findings show marked clock dysfunction following chronic Vps15 inactivation.

### Class 3 PI3K drives the clock in a cell-autonomous manner

As class 3 PI3K is ubiquitously expressed, to test its cell-autonomous role in clock control, we used immortalized mouse embryonic fibroblasts (MEFs) from *Vps15*[f/f] mice, which were efficiently synchronized by a dexamethasone pulse (rhythmic expression of Rev-Erbα; Extended Data Fig. 2a). To acutely delete Vps15, *Vps15*[f/f] MEFs were transduced with adenoviral vectors expressing Cre recombinase, resulting in depletion of Vps15 protein accompanied by Vps34 degradation and decreased PI3P levels (Fig. 1g and Extended Data

---

**Fig. 1 | Bmal1 transcriptional activity is inhibited following Vps15 inactivation. a**, Relative transcript levels of the indicated genes in the livers of 5-week-old Vps15LKO and control male mice collected at the indicated ZT times. The mice were fed ad libitum and kept under a 12-h light–dark regimen (grey shading). Data are the mean ± s.e.m. fold increase compared with the WT ($n = 3$ Vps15LKO$_{ZT0,ZT18}$, $n = 4$ WT$_{ZT0,ZT6,ZT18}$ and Vps15LKO$_{ZT6,ZT12}$, and $n = 5$ WT$_{ZT12}$ animals). *$P < 0.05$; two-tailed unpaired Student's *t*-test. **b**, Immunoblot analysis of total protein extracts from the livers of the mice in **a**. **c**, Bmal1-binding peak profile (top) and heatmap (bottom) of the livers of male WT ($n = 2$) and Vps15LKO ($n = 2$) mice collected at ZT6. Peaks are ordered by their signal strength. Each row shows the region from −3 kb to +3 kb from the TSS. **d**, The log intensity ratio versus mean log intensity (MA) plot showing differential binding of Bmal1 and histone H3K27Ac in the livers of the WT and Vps15LKO mice in **c** (false detection rate (FDR) < 0.15). Concentration represents the normalized read counts within peaks; fold change was calculated relative to the WT. Blue line, log$_2$(fold change) = 0; pink line, nonlinear LOESS fit curve of the coverage levels and fold changes.

**e**, Bmal1 recruitment to the indicated gene promoters, determined by chromatin immunoprecipitation with quantitative PCR (ChIP–qPCR), in the livers (collected at ZT6–ZT12 and ZT18–ZT24) of mice ($n = 3$) treated as in **a**. Data are the mean ± s.e.m. *$P < 0.05$; one-way analysis of variance (ANOVA) with Benjamini–Hochberg correction. **f**, Number of overlapping and non-overlapping Bmal1 and H3K27Ac peaks reduced in Vps15LKO mice (top). The GO biological process analysis of the genes corresponding to 539 overlapping peaks is presented (bottom). **g,h**, Immunoblot analysis of total protein extracts ($n = 4$; **g**) and relative transcript levels of the indicated genes (independent repeats, $n = 4$ for CRE$_{CT32,CT36}$ and $n = 5$ for all other groups; **h**) in dexamethasone-synchronized control (green fluorescent protein, GFP) and Vps15-depleted (CRE) MEFs. Densitometric analyses of Rev-Erbα levels normalized to Gapdh. Data are the mean ± s.e.m. fold change compared with GFP-treated cells. Δex2, deletion of exon2 in Vps15 locus. *$P < 0.05$; two-tailed unpaired Student's *t*-test. **a,g,h**, Rhythmicity was determined using JTK_CYCLE (Supplementary Table 1). Source data and unprocessed blots are provided.

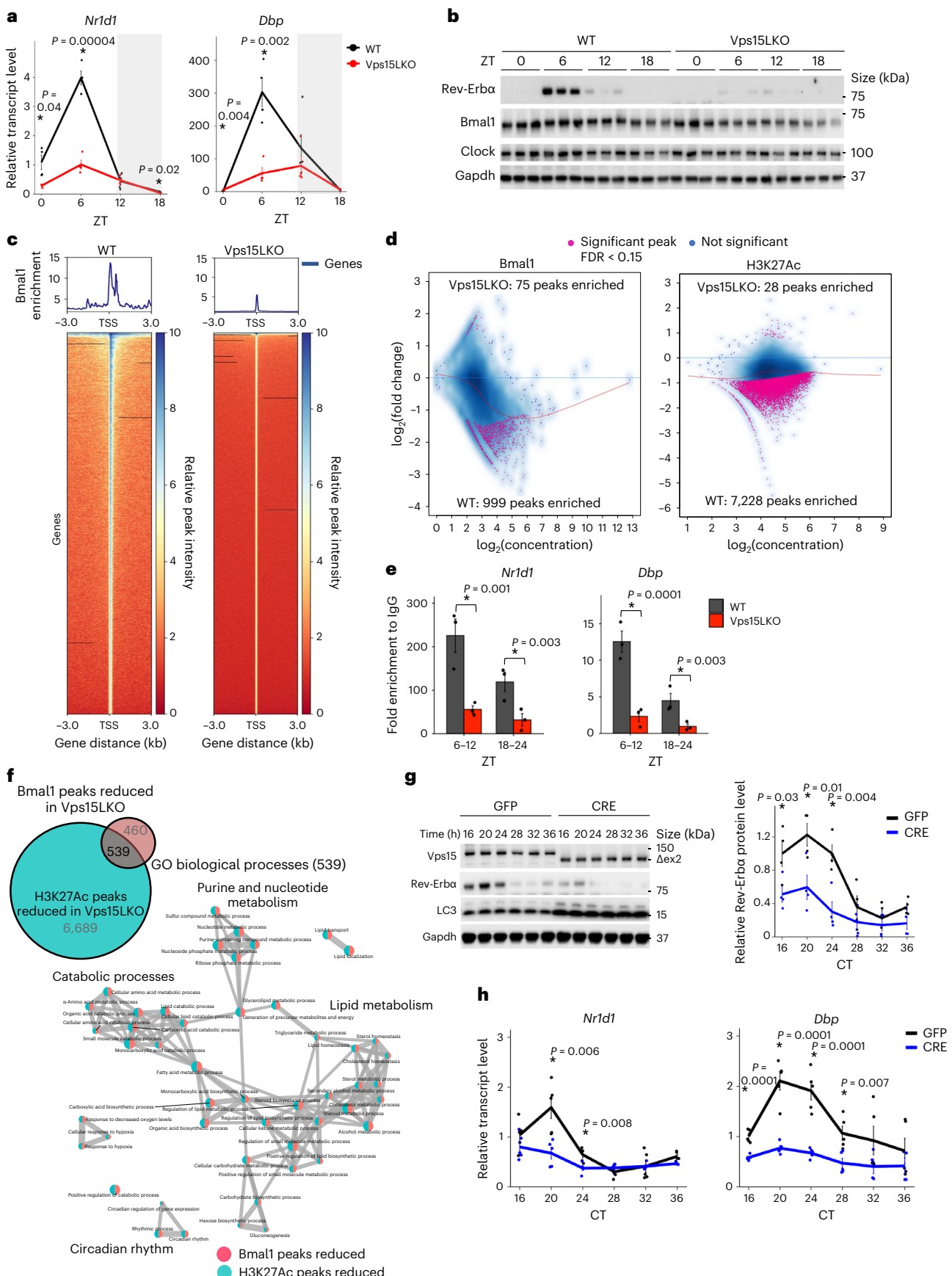

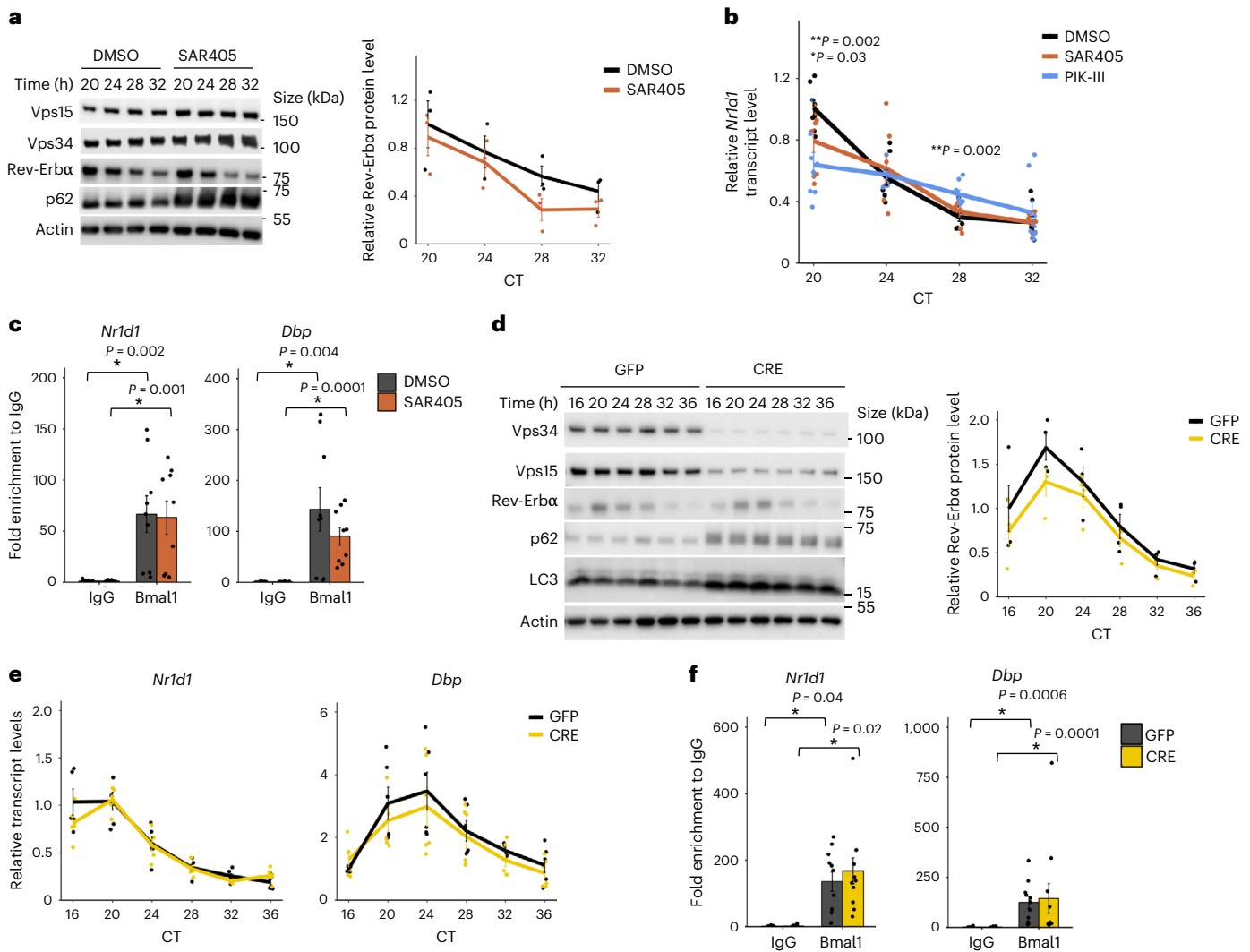

**Fig. 2 | The lipid kinase activity of Vps34 is dispensable for Rev-Erbα protein expression. a,b,** Immunoblot analysis (*n* = 3) of dexamethasone-synchronized MEFs treated with dimethyl sulfoxide (DMSO, control) or Vps34 inhibitor (SAR405). Densitometry analyses of Rev-Erbα protein levels, normalized to Actin, presented as the fold change over DMSO-treated MEF cells (right). **b,** Relative transcript levels (*n* = 8; **b**) of dexamethasone-synchronized MEFs treated with DMSO (control) or Vps34 inhibitors (SAR405 and PIK-III). *\*P* < 0.05 for SAR405 versus DMSO and *\*\*P* < 0.05 for PIK-III versus DMSO; two-way ANOVA with Benjamini−Hochberg correction. **a,b,** Data are the mean ± s.e.m. from three independent experiments. **c,** ChIP–qPCR of Bmal1 recruitment to the *Nr1d1* and *Dbp* promoters in MEFs treated with SAR405 or DMSO as in **a** collected at 24 h post synchronization. Data are the mean ± s.e.m. fold enrichment from three independent experiments (*n* = 9). *\*P* < 0.05; two-way ANOVA with Benjamini−

Hochberg correction. **d,** Immunoblot analysis of total protein extracts (*n* = 4). Densitometric analyses of Rev-Erbα levels normalized to Actin (right). **e,** Relative transcript levels of the indicated genes in dexamethasone-synchronized control (GFP) and Vps34-depleted (CRE) Vps34*^{f/f}* MEFs. **d,e,** Data are the mean ± s.e.m. fold change compared with GFP-treated MEFs from three independent experiments (*n* = 5 for GFP_{CT16} and *n* = 6 for all other groups). Rhythmicity was determined using JTK_CYCLE (Supplementary Table 1). **f,** ChIP–qPCR of Bmal1 recruitment to the *Nr1d1* and *Dbp* promoters in GFP and CRE MEFs collected at 24 h post synchronization. Data are the mean ± s.e.m. from three independent experiments (*n* = 10 for GFP ChIP-IgG and *n* = 11 for all other groups). *\*P* < 0.05; two-way ANOVA with Benjamini−Hochberg correction. Source data and unprocessed blots are provided.

Fig. 2b,c[24]. Notably, recombination of the *PIK3R4* locus results in expression of truncated Vps15 protein, which does not interact with Vps34 as we reported[37]. In agreement with this, Vps15 depletion manifested in autophagy defects, evidenced by the accumulation of LC3 and p62 proteins—which are required for autophagosome formation and cargo delivery, respectively—as well as findings of defective autophagy flux using a mRFP−EGFP-LC3 reporter (Fig. 1g and Extended Data Fig. 2b,d,e). Importantly, Vps15 depletion in dexamethasone-synchronized MEFs resulted in decreased rhythmic expression of *Nr1d1* transcripts and reduced Rev-Erbα protein levels (Fig. 1g,h and Supplementary Table 1). Bmal1 inhibition in Vps15-depleted MEFs was further supported by findings of decreased

expression of *Dbp* and *Per2* accompanied by a significant reduction of rhythmic cycle amplitude, whereas only *Cry1* lost its rhythmicity (Fig. 1h, Extended Data Fig. 2f and Supplementary Table 1). Unlike chronic Vps15 deletion in liver, the nuclear levels of Bmal1 protein were unmodified in Vps15-depleted MEFs (Extended Data Fig. 2g,h). This was further backed by a similar half-life of nuclear Bmal1 in control and Vps15-depleted MEFs (Extended Data Fig. 2i). Thus, unlike following acute Vps15 deletion, prolonged dysfunction of class 3 PI3K accompanied by chronic inhibition of endocytosis and autophagy might impact the expression and/or turnover of Bmal1 protein. Together, acute inactivation of class 3 PI3K in cells manifests as inhibition of Bmal1 activity without impacting its protein turnover.

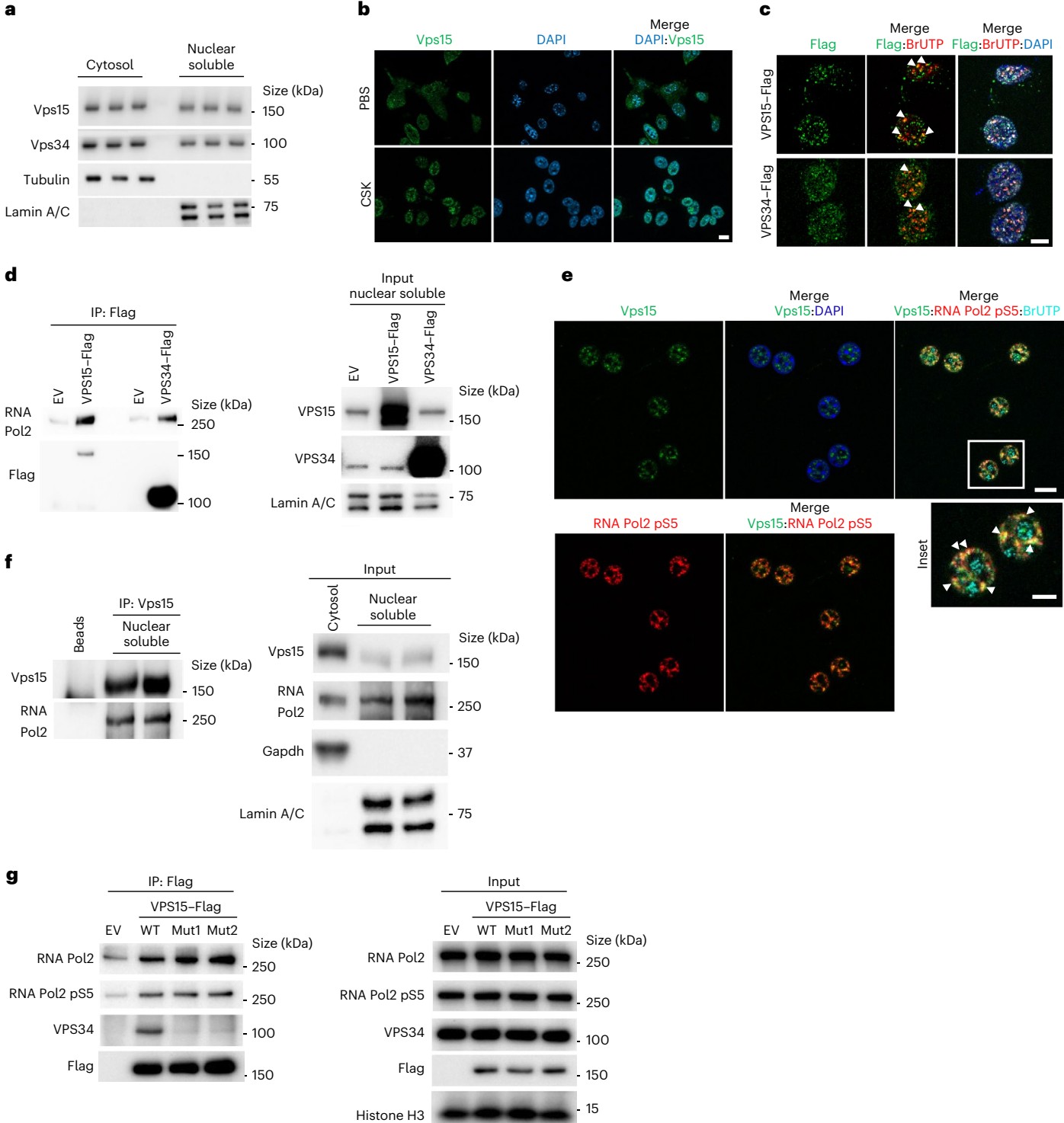

**Fig. 3 | Nuclear class 3 PI3K interacts with RNA Pol2. a**, Immunoblot analysis of soluble nuclear and cytosolic fractions from MEFs using antibodies to the indicated proteins. Tubulin and lamin A/C serve as controls for loading and fraction cross-contamination. The experiment was performed seven times. **b**, Immunofluorescence microscopy analyses of Vps15 in MEF cells treated with CSK buffer for 2 min. Nuclei were stained with 4,6-diamidino-2-phenylindole (DAPI). Scale bar, 10 μm. **c**, Transcription assay with ectopically expressed VPS15–Flag and VPS34–Flag in MEFs. To label de novo transcription sites, MEFs were incubated with BrUTP 24 h post transfection. For co-localization, immunofluorescence microscopy was performed with anti-Flag and anti-BrUTP. Scale bar, 5 μm. **d**, Immunoblot analyses, using antibodies to the indicated proteins, of the Flag-containing immunoprecipitates from the soluble nuclear fraction of HEK293T cells transfected with empty vector (EV), or VPS15–Flag- or VPS34–Flag-expressing vectors. **e**, Immunofluorescence microscopy analyses of the co-localization of endogenous RNA Pol2 phospho-S5 (pS5), Vps15 and de novo transcription sites labelled with BrUTP in primary mouse hepatocytes. Scale bars, 10 μm and 5 μm (inset). **c,e**, The white triangles point to co-localization. **f**, Immunoblot analyses, using anti-RNA Pol2, of Vps15-containing immunoprecipitates from the soluble nuclear fractions of MEF cells. **b**–**f**, The experiments were performed three times. **g**, Immunoblot analyses of VPS15-containing complexes immunoprecipitated from HEK293T cells transiently transfected with Flag-conjugated WT or mutant VPS15 ((VPS15Mut1 (Mut1) and VPS15Mut2 (Mut2)). The cells were collected 48 h post transfection and VPS15 was immunoprecipitated using anti-Flag. The experiment was performed five times. Representative blots (**a,d,f,g**) or microscopy fields of view (**b,c,e**) are shown. IP, immunoprecipitation. Unprocessed blots are provided.

## Kinase-independent role of class 3 PI3K in clock control

To address the possible connection between the lipid kinase activity of class 3 PI3K and Bmal1-driven transcriptional rhythmicity, we used two selective and structurally distinct inhibitors[38,39]. As expected, both SAR405 and PIK-III efficiently inhibited Vps34, as witnessed by p62 accumulation due to autophagy block, without affecting the levels of Vps34 or Vps15 protein (Fig. 2a and Extended Data Fig. 3a). In contrast to the findings in Vps15LKO mice and Vps15-depleted MEFs, neither inhibitor reduced the levels of Rev-Erbα protein despite having a minor effect on its transcript levels (Fig. 2a,b). However, treatment with the Vps34 inhibitors resulted in inconsistencies, such as decreased levels of *Dbp* transcripts at all time points in response to SAR405 but only a minor effect in response to PIK-III (decrease at circadian time 20 (CT20); Extended Data Fig. 3b). Similarly, *Cry1* expression was unmodified by PIK-III but SAR405 treatment resulted in increased transcript levels at CT20 and CT28 (Extended Data Fig. 3b). However, both compounds decreased *Per2* transcripts at all time points (Extended Data Fig. 3b). Notably, SAR405 treatment did not modify Bmal1 chromatin enrichment on the *Nr1d1* and *Dbp* promoters (Fig. 2c). In light of these different effects with two inhibitors, we treated Vps15-depleted MEFs with a third compound, the highly potent Vps34-IN1 (ref. 40), which decreased the levels of PI3P and resulted in autophagy block (Extended Data Fig. 3c,d). Consistent with other inhibitors, Vps34-IN1 treatment did not modify the Rev-Erbα levels. Surprisingly, it resulted in increased *Nr1d1* and *Cry1* transcript levels, and no difference in the *Dbp* levels (Extended Data Fig. 3d,e). Finally, treatment with the three inhibitors at two different doses that led to Vps34 inhibition, witnessed by autophagy block, did not show any major effect on the rhythmic activity of the *Nr1d1*-luciferase (*Nr1d1-Luc*) reporter (Extended Data Fig. 3f,g). These non-overlapping effects with different Vps34 inhibitors probably stem from their off-targets for which functional links with the clock were reported, including CDK2a, GSKb, ABL1, MAPK1 and SMG1 (refs. 41–45). In summary, the three efficient Vps34 inhibitors did not result in consistent repression of circadian *Nr1d1* transcription and, thus, did not phenocopy Vps15-null models.

## Vps34 protein is not required for Bmal1 function

Our findings suggested that Vps15 or Vps34 proteins might directly modulate Bmal1 activity. To test the implication of Vps34, we established immortalized *Vps34*[f/f] MEFs. Similar to *Vps15*[f/f] MEFs, the genetic deletion of *Vps34* resulted in instability of both subunits and manifested in autophagy block (Fig. 2d and Extended Data Fig. 4a,b). Mirroring pharmacological inhibition of Vps34 lipid kinase, the transcript levels of *Per2* were decreased in Vps34-depleted MEFs without impacting its rhythmicity (Extended Data Fig. 4c and Supplementary Table 1). However, neither the transcript rhythmicity of *Dbp*, *Cry1*

and *Nr1d1* nor the rhythmicity of Rev-Erbα protein were modified in dexamethasone-synchronized Vps34-null MEFs (Fig. 2d,e, Extended Data Fig. 4c and Supplementary Table 1). Moreover, as in Vps15-null MEFs, the levels of cytosolic and nuclear Bmal1 protein were unchanged following Vps34 depletion (Extended Data Fig. 4d). Finally, in line with its unmodified protein levels, Bmal1 chromatin recruitment to the promoters of its target genes was similar between Vps34-null and control cells (Fig. 2f). Thus, these findings favour a Vps34-idependent role of Vps15 upstream of the circadian clock.

## Class 3 PI3K is present in the nucleus

So far, class 3 PI3K functions in animal cells were limited to its role in cytosol as a lipid kinase in vesicular trafficking. Given that lipid kinase inhibition of Vps34 did not change the transcriptional activity of Bmal1, we investigated whether Vps15 and Vps34 proteins have a role in nuclear transcription. Biochemical fractionation showed that both Vps15 and Vps34 were detected in the nucleus of mouse and human cells as well as in mouse liver (Fig. 3a and Extended Data Fig. 5a–c). In liver Vps15 had modest diurnal rhythmicity in the soluble nuclear fraction and euchromatin, similar to its transcript (Extended Data Fig. 5b–e and Supplementary Table 1). However, only the Vps34 euchromatin protein levels exhibited rhythmicity (Extended Data Fig. 5b–e and Supplementary Table 1). In the nuclear fractions the levels of Vps15 and Vps34 protein were lower during the dark phase compared with the light phase, coinciding with transcriptionally active Bmal1 (Extended Data Fig. 5d). Next, we tested their nuclear localization using immunofluorescence microscopy. Although we could not validate commercially available antibodies to Vps34, endogenous Vps15 showed cytosolic–nuclear staining (Fig. 3b). Moreover, nuclear staining of Vps15 was unmasked by treatment with high-sucrose buffer (CSK), which removes cytosolic proteins before fixation (Fig. 3b). Furthermore, treatment of MEFs with ivermectin, an inhibitor of importin α and β, decreased the nuclear levels Vps15 and Vps34, suggesting their active nuclear transport (Extended Data Fig. 5f). This was further supported by findings that endogenous importin α co-immunoprecipitated with Vps15 and to a lesser extent with Vps34 (Extended Data Fig. 5g). Finally, using a proximity ligation assay (PLA), we found that the Vps15–Vps34 complex is also present in the nucleus (Extended Data Fig. 5h). Together, these findings favour the presence of an actively transported nuclear pool of class 3 PI3K.

## Nuclear class 3 PI3K interacts with RNA Pol2

Next, we tested whether nuclear Vps15 and Vps34 are involved in transcription. To label the de novo transcription sites, cells ectopically expressing Vps15 and Vps34 were pulsed with bromouridine-triphosphate (BrUTP). Strikingly, both Vps15 and Vps34

**Fig. 4 | Vps15 interacts with Bmal1. a,** Immunoblots of Bmal1-containing immunoprecipitates from total extracts of MEFs. Right: clock levels normalized to Bmal1 in immunoprecipitates presented as the mean ± s.e.m. (independent repeats, *n* = 3). **b,** ChIP–qPCR analysis of Bmal1 recruitment to the indicated gene promoters in GFP and CRE MEFs 24 h post dexamethasone synchronization. Data are the mean ± s.e.m. fold enrichment from three independent experiments (*n* = 6 for GFP-IgG, *n* = 7 for GFP-Bmal1 and *n* = 8 for all other groups). **c,** Left: immunoblot of BMAL1 immunoprecipitates from the total extracts of HEK293T cells transfected with increasing amounts of VPS15WT. Right: CLOCK levels normalized to BMAL1 in BMAL1 immunoprecipitates are presented as the fold difference to the cells transfected with empty vector. Data are the mean (independent repeats, *n* = 2 for VPS15WT–Flag 2.5 µg and *n* = 3 for all other groups). **d,** Immunoblot of VPS15–Flag-containing complexes immunoprecipitated from HEK293T cells transfected with the indicated constructs. **e,** Proximity ligation assay of endogenous VPS15 and Flag–BMAL1 in HEK293T cells (co-transfected with GFP). The 'no antibodies' condition served as the negative control. Scale bar, 10 µm. Right: data are the mean ± s.e.m. number of proximity puncta per cell (*n* = 8 no antibody and *n* = 25 anti-Flag + anti-VPS15

fields) from three independent experiments (>500 cells per condition). **b,e,** *P* < 0.05; two-tailed unpaired Student's *t*-test. **f,** Immunoblot, using the indicated antibodies, of Vps15 immunoprecipitates from the soluble nuclear fractions of liver tissue of WT 5-week-old male mice (ZT6; fed ad libitum). **g,** Immunoblot analyses of VPS15WT- and VPS15Mut1-interacting proteins from HEK293T cells immunoprecipitated with anti-Flag. **h,** Left: immunoblot of the total extracts of GFP and CRE MEFs transduced with adenoviral vectors expressing GFP, VPS15WT or VPS15Mut1 and synchronized by dexamethasone. Right: densitometric analyses of Rev-Erbα normalized to Actin presented as the fold change compared with the GFP-GFP condition. Data are the mean ± s.e.m. from three independent repeats. *P* < 0.05 for GFP versus CRE-GFP and **P* < 0.05 for CRE-VPS15WT/Mut1 versus GFP-CRE; two-tailed unpaired Student's *t*-test. **i,** Left: domain organization of mouse full-length Bmal1 protein (Protein Data Bank, 4F3L). Right: mapping of its truncated constructs. Middle: Immunoblot of Bmal1 immunoprecipitates from total extracts of HEK293T cells. The experiment was performed three times. Representative blots are shown. aa, amino acids. Source data and unprocessed blots are provided.

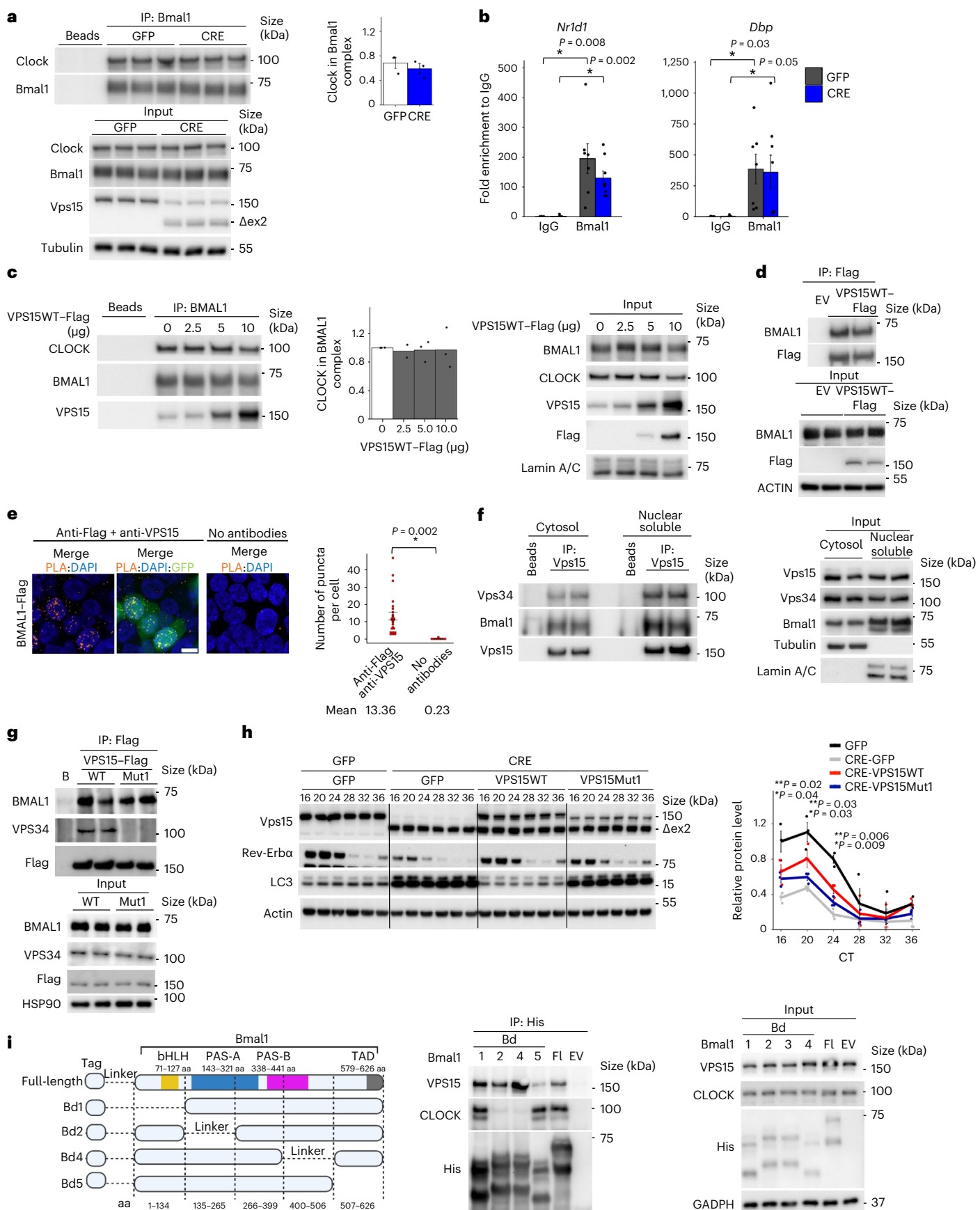

were readily co-localized with BrUTP sites (Fig. 3c). Moreover, both Flag-tagged VPS15 and VPS34 were co-immunoprecipitated with RNA polymerase II (RNA Pol2; Fig. 3d). Furthermore, in primary hepatocytes, endogenous Vps15 co-localized in BrUTP-positive sites with transcription elongation active RNA Pol2 phosphorylated on S5 (Fig. 3e). Consistent with these data, Vps15 and RNA Pol2 were co-immunoprecipitated from the nuclear fraction (Fig. 3f). Finally, we used two distinct Vps15 mutants reported in yeast, in which the E200R (referred to as VPS-15Mut1) or D165R (referred to as VPS15Mut2) amino-acid substitutions were shown to prevent its binding with Vps34 (ref. 24). Similar to findings in yeast, these amino-acid substitutions in human VPS15 (E200R or D165R) abolished complex formation between VPS15 and endogenous VPS34 (Fig. 3g). However, both the VPS15 mutants co-immunoprecipitated with RNA Pol2 (Fig. 3g). In summary, these findings suggest that both Vps15 and Vps15–Vps34 might be implicated in nuclear transcription. Importantly, the interaction of Vps15 with RNA Pol2 does not rely on Vps34, without excluding additional roles of nuclear Vps34 or the Vps15–Vps34 complex.

## Vps15 interacts with Bmal1

To explore the mechanisms of Bmal1 dysfunction in the absence of Vps15, we investigated whether Vps15 impacts the Bmal1–Clock interaction and chromatin recruitment. Co-immunoprecipitation in control and Vps15-depleted MEFs showed no differences in the amount of endogenous Clock in complex with Bmal1 (Fig. 4a). Vps15 depletion in MEFs also did not change the recruitment of Bmal1 to the promoters of its target genes (Fig. 4b). Furthermore, VPS15 overexpression neither impacted BMAL1–CLOCK complex formation nor Bmal1 or RNA Pol2 loading on chromatin (Fig. 4c and Extended Data Fig. 6a). Surprisingly, in immunoblots using anti-VPS15, VPS15 was detected in increasing amounts in BMAL1 immunoprecipitates, correlating with its overexpression (Fig. 4c). This finding was validated by pulling down endogenous BMAL1 with ectopically expressed Flag-conjugated wild-type (WT) VPS15 (VPS15WT–Flag; Fig. 4d). It was further supported by the detection of ectopically expressed BMAL1 in proximity to endogenous VPS15 in the nucleus (Fig. 4e). In line with these results, interaction of endogenous Vps15 and Bmal1 proteins was detected predominantly in nuclear extracts of MEF cells and liver tissue (Fig. 4f and Extended Data Fig. 6b). This was concordant with the detection of proximity puncta between endogenous VPS15 and BMAL1, and their partial co-localization in MEF cells (Extended Data Fig. 6c,d). Nuclear Vps15–Vps34 complex was also found in MEFs and liver extracts, albeit to a lesser extent compared with its cytosolic interaction (Fig. 4f and Extended Data Fig. 6b,c). Furthermore, the interaction between endogenous Vps15 and Bmal1 was detected in liver tissue around the peak of Bmal1 nuclear localization (Extended Data Fig. 6e). To test whether the Vps15–Vps34 complex is required for the Vps15–Bmal1 interaction, we used a VPS15 point mutant, which we showed does not interact with Vps34 and cannot restore the autophagy when expressed in Vps15-null MEFs (Fig. 4g,h). Similar to VPS15WT, the VPS15Mut1 protein co-precipitated with endogenous BMAL1 (Fig. 4g). Moreover, expression of VPS15Mut1 in Vps15-null MEFs partially restored the levels of Rev-Erbα (Fig. 4h). Finally, to investigate the interaction region of Bmal1 mediating binding with VPS15, we employed a panel of truncated Bmal1 mutants that span its identified domains (Fig. 4i; Protein Data Bank, 4F3L)[46,47]. Note that although the exact boundaries of the PAS-A and PAS-B domains essential for heterodimerization of the Bmal1–Clock vary somewhat in different sources (Protein Data Bank, 4F3L[47,48]), consistent with the published observations[47], Clock binding was abrupted even by partial deletion of the PAS-A (Bmal1 domain (Bd) 2) or PAS-B (Bd4) domains in Bmal1 (Fig. 4i). Curiously, we found that CLOCK and Vps15 co-immunoprecipitated via different regions of Bmal1. To this end, deletion of the carboxy (C)-terminal transactivation domain (TAD) region of Bmal1 was detrimental for its interaction with VPS15 without affecting CLOCK binding (Fig. 4i). Thus, the TAD

domain of Bmal1, a binding region for Cry1 and CBP[49–51], also mediates its interaction with VPS15. Reciprocally, to investigate the region of VPS15 necessary for its interaction with Bmal1, we generated a panel of truncated VPS15 proteins by deleting its domains[26] (Extended Data Fig. 6f). Co-immunoprecipitation showed that all fragments of VPS15 interact with Bmal1 (Extended Data Fig. 6f). These findings point to the HEAT domain of Vps15 mediating direct or indirect interaction with Bmal1, without ruling out multiple binding sites.

## Vps15 coactivates Bmal1 independently of Vps34

Bmal1–Clock inhibition in Vps15-null models and the interaction of Vps15 with Bmal1 prompted us to investigate whether Vps15 could act as a transcriptional coactivator of Bmal1. For this, we employed a luciferase assay in HEK293T cells following ectopic expression of BMAL1 and CLOCK in the presence of the E-box-LUC reporter. Notably, VPS15WT and VPS15Mut1 overexpression coactivated BMAL1–CLOCK, comparable to its known coactivators p300 and CBP (Fig. 5a). Next, given the interaction between Vps15 and the TAD region of Bmal1, we queried whether ectopic expression of VPS15 interferes with the repressive action of Cry1. As expected, co-expression of Cry1 repressed the transcriptional activity of BMAL1–CLOCK (Fig. 5b). Importantly, co-expression of VPS15WT or VPS15Mut1 together with Cry1 could partially restore BMAL1–CLOCK activity (Fig. 5b).

The findings that (1) VPS15Mut1 does not interact with Vps34 but coactivates BMAL1–CLOCK and (2) the presence of a functional clock in Vps34-null MEFs are in favour of Vps15 being a coactivator of Bmal1–Clock independently of Vps34. On the other hand, Vps34 is present in the nucleus, co-localizes with de novo transcription sites, interacts with RNA Pol2 and its deletion leads to decreased *Per2* transcript levels. Thus, we further investigated the capacity of ectopically expressed VPS34 to interact with and coactivate BMAL1–CLOCK. First, proximity puncta were observed between overexpressed VPS34 and endogenous BMAL1 (Extended Data Fig. 7a). Second, when increasing amounts of VPS15 or VPS34 were co-expressed with BMAL1–CLOCK in luciferase assays, we observed that although moderate VPS15 overexpression (2.4-fold, dose 2) led to BMAL1–CLOCK coactivation, only high levels of VPS34 overexpression (tenfold, dose 3) coactivated the E-box-Luc reporter (Extended Data Fig. 7b). Moreover, given the finding that VPS15 overexpression interfered with the repressive effect of Cry1 on BMAL1–CLOCK (Fig. 5b), we tested whether a high level of VPS34 overexpression would produce a similar effect. Remarkably, although VPS34 co-expression activated BMAL1–CLOCK, it did not interfere with the repressor action of Cry1 (Extended Data Fig. 7c).

Finally, the findings of a functional clock in Vps34-null MEFs prompted us to investigate the nuclear Vps15 levels in Vps34-depleted models. Within current state-of-the-art, Vps15 and Vps34 form a constitutive complex and are indispensable for their reciprocal stability. In support of a potential Vps34-independent role of Vps15 for clock control, the levels of Vps15 in Vps34-null cells were decreased by 70% in total protein fractions and only by 40% in nuclear extracts (Extended Data Fig. 7d). This was further corroborated by expression analyses in Vps15- and Vps34-depleted MEFs; the levels of Vps15 protein in the nuclear extracts of Vps34-depleted MEFs were twofold higher than those of Vps15-null cells (Extended Data Fig. 7e). Collectively, these findings advocate a coactivator function of Vps15 on Bmal1 that probably does not rely on its interaction with Vps34.

## Vps15 and Bmal1 co-regulate metabolic gene networks

Our finding that Vps15 coactivates BMAL1–CLOCK prompted us to investigate whether Vps15 resides on chromatin in regions bound by Bmal1. We found enrichment of Vps15 at Bmal1-bound E-box regions in the promoters of circadian clock-related genes (*Nr1d1* and *Dbp*) in the livers of mice collected at the peak of Bmal1 chromatin binding as well as in synchronized MEFs (Fig. 5c and Extended Data Fig. 7f). Notably, Vps15 enrichment was considerably lower compared with

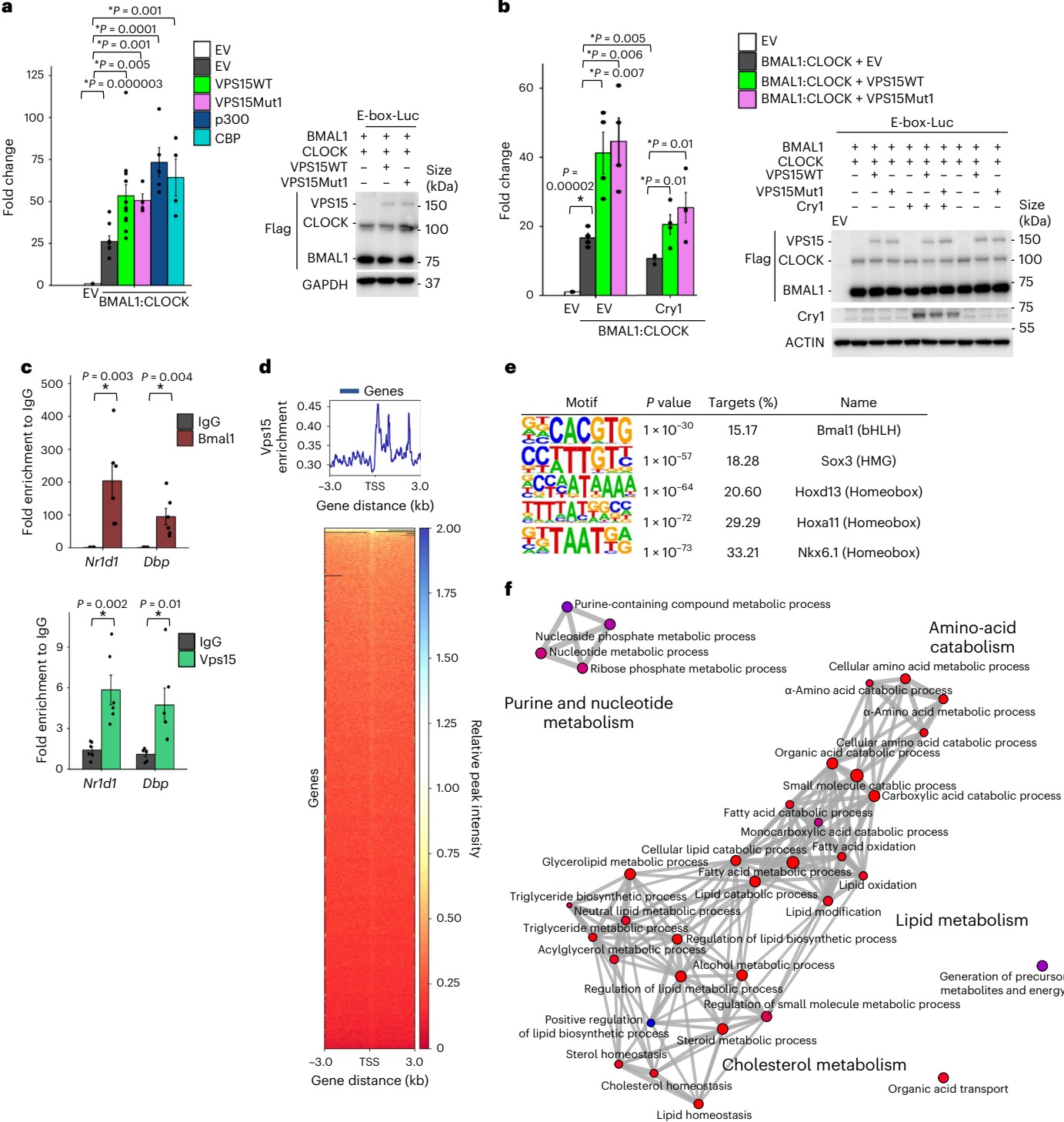

**Fig. 5 | Vps15 transcriptionally coactivates Bmal1. a**, Left: luciferase assay in HEK293T cells co-transfected with the E-box-Luc reporter and EV or BMAL1–CLOCK with or without VPS15WT, VPS15Mut1, CBP or p300. Relative luminescence presented as fold difference compared with cells transfected with E-box-Luc + EV. Data are the mean ± s.e.m. (independent experiments, $n = 4$ for BMAL1–CLOCK + VPS15Mut1/CBP, $n = 5$ for BMAL1–CLOCK + p300, $n = 8$ for BMAL1–CLOCK + EV and $n = 12$ for BMAL1–CLOCK + VPS15WT). Right: representative immunoblot with anti-Flag showing the expression levels of BMAL1, CLOCK, VPS15 and GAPDH as the loading control (right). **b**, Left: luciferase assay in HEK293T cells co-transfected with E-box-Luc reporter construct and EV or plasmids expressing BMAL1 and CLOCK with or without Cry1, VPS15WT or VPS15Mut1. Relative luminescence is presented as the fold difference compared with E-box-Luc-transfected cells. Data are the mean ± s.e.m. ($n = 4$ independent experiments). Right: representative immunoblot showing

the expression levels of Cry1, BMAL1, CLOCK, VPS15 and GAPDH as the loading control. **c**, ChIP–qPCR analyses of Bmal1 and Vps15 enrichment at the promoters of the indicated genes in the liver tissue of 5-week-old male mice (ZT6). Data are the mean ± s.e.m. fold enrichment ($n = 6$ mice). **a**–**c**, *$P < 0.05$; two-tailed unpaired Student's $t$-test. **d**, Vps15 binding peak profile and heatmap for the livers ($n = 2$ mice) of WT male mice (ZT6; 5 weeks old). Peaks are ordered by their signal strength and each row shows the promoter region from −3 kb to +3 kb from the TSS. **e**, HOMER motif analysis of Vps15 peaks. Identified consensus motifs are shown with their respective significance calculated with HOMER and the percentage of target coverage in all ChIP peaks. **f**, Analysis of GO Biological process using enrichGO showing significantly enriched genes for which chromatin binding of Vps15 was detected in WT liver and for which Bmal1 chromatin enrichment and transcript levels were downregulated in the livers of Vps15LKO mice. Source data and unprocessed blots are provided.

Bmal1. Although it might be due to differences in antibody affinity, it is also consistent with a coactivator role of Vps15 as it does not contain a DNA-binding domain and is likely to be localized further away from chromatin (more labile and presents constraints for fixation). Moreover, consistent with the findings of transcriptionally active Bmal1 in Vps34-null MEFs, chromatin enrichment of Vps15 in the E-box region of the *Nr1d1* and *Dbp* promoters bound by Bmal1 did not differ between control and Vps34-null MEFs (Extended Data Fig. 7g).

To provide a broader view of gene networks co-regulated by Vps15 and Bmal1, we performed ChIP–Seq of Vps15. Vps15 ChIP–Seq of the livers of mice collected at ZT6, the peak of Bmal1 transcriptional activity, showed that Vps15 binding is enriched at the TSS, with a Bmal1-binding motif present among the precipitated chromatin regions (Fig. 5d,e and Supplementary Table 3). To identify the biological processes that are potentially transcriptionally co-regulated by Vps15 and Bmal1, we selected the genes for which Vps15 was bound in the promoters (ChIP–Seq) as well as for which both transcript levels (RNA sequencing (RNA-Seq); Supplementary Table 4) and Bmal1 enrichment (ChIP–Seq; Fig. 1c) were decreased in the livers of Vps15LKO mice. Gene ontology pathway analyses of the 371 shared genes pointed to metabolic processes under potential transcriptional co-control of Vps15 and Bmal1 including lipid, amino-acid and nucleotide metabolism (Fig. 5f and Supplementary Table 5).

### Vps15 controls metabolic rhythmicity in liver

To investigate the input of Vps15–Bmal1 on metabolic homeostasis, we performed targeted metabolomics analyses of the liver tissue of WT and Vps15LKO mice collected every 6 h over a period of 24 h. We reasoned that collection at physiologically relevant low-feeding activity (ZT0 and ZT6) and active-feeding (ZT12 and ZT18) states would be stringent enough to detect the metabolic processes that were most impacted. The metabolomics analyses yielded a total of 144 annotated metabolites with nearly half (63 of 144) showing circadian rhythmicity in the WT mouse liver (Fig. 6a, Extended Data Fig. 8a,b and Supplementary Tables 1,6). In line with previous reports, the detected rhythmic metabolites belonged to diverse chemical classes (Extended Data Fig. 8b)[52,53]. Only 38 metabolites were rhythmic in the Vps15LKO livers (Fig. 6a, Extended Data Fig. 8a and Supplementary Tables 1,6), including 17 metabolites that gained rhythmicity, in coherence with reports of Bmal1-independent rhythms[54,55]. Among oscillating metabolites in the liver of WT mice, 42 (66.7% of all rhythmic metabolites) were detected as arrhythmic in Vps15LKO livers (Fig. 6a). These metabolites belong to classes of amino acids (50%), nucleotides (23.8%), lipids (11.9%), carbohydrates (11.9%) and ketone bodies (2.4%; Extended Data Fig. 8b). Pathway analyses pointed to purine metabolism being the most affected, followed by glutamate, arginine and fatty-acid metabolism (Fig. 6b). These same pathways were suggested to be under transcriptional control by Vps15 and Bmal1 in ChIP–Seq analyses (Figs. 1f and 5f). The circadian clock was proposed to act upstream of de novo

purine synthesis in liver but the mechanisms were not elucidated[56]. The requirement of Vps15 for purine metabolism was evidenced by dysregulated levels and rhythmicity of metabolites with three types of responses (Extended Data Fig. 8c). The first were metabolites that lost rhythmicity in Vps15LKO livers (increased or decreased levels in Vps15LKO—that is, inosine monophosphate (IMP), inosine, hypoxanthine, xanthine, guanosine and guanine). The second group were metabolites that were not rhythmic in WT livers but whose levels were affected by Vps15 deletion (phosphoribosyl pyrophosphate (PRPP), GDP, AMP and ADP). Third were metabolites that did not differ between two genotypes (adenine and adenosine; Extended Data Fig. 8c). In de novo purine synthesis, IMP represents a central metabolite production, which relies on PRPP, glutamine, aspartate, glycine and ATP (Fig. 6c and Extended Data Fig. 8c)[57]. In addition to IMP, glutamine and aspartic acid lost rhythmicity in the livers of Vps15LKO mice, whereas the levels of PRPP differed significantly from the WT (Extended Data Fig. 8c). The decreased ratio of IMP/PRPP further supported the defect in IMP synthesis in the livers of Vps15LKO mice (Fig. 6d). This was backed by a lower incorporation of two labelled $^{15}$N-glutamine tracer atoms in IMP (mass ($M$) + 2) in the livers of Vps15LKO mice compared with controls (Fig. 6e). In line with the in vivo findings, shRNA knockdown of Vps15 or Bmal1 in AML12 cells resulted in significant inhibition of $^{15}$N-glutamine tracer incorporation into IMP ($M$ + 2), AMP ($M$ + 2) and ADP ($M$ + 2) compared with control cells (Fig. 6f). Thus, rhythmic de novo purine synthesis in liver cells depends on Vps15 expression.

### Vps15 coactivates Bmal1 for *Ppat* transcription

Ppat is a key enzyme in de novo purine synthesis that drives the production of IMP. It was suggested, but not formally demonstrated, to be a Bmal1 gene target[56]. Consistent with the metabolomics findings and Bmal1 inhibition, the levels of *Ppat* transcripts were decreased and lost rhythmicity in the livers of Vps15LKO mice (Fig. 6g and Supplementary Table 1). In line with the coactivating role of Vps15 on Bmal1, overexpression of either VPS15WT or VPS15Mut1 protein in primary mouse hepatocytes resulted in the upregulation of *Ppat* transcripts, comparable to Bmal1 overexpression (Extended Data Fig. 8d). Inversely, acute deletion of Vps15 in primary mouse hepatocytes resulted in decreased levels of *Ppat* transcripts (Extended Data Fig. 8e). Providing support for Bmal1–Vps15 acting upstream of *Ppat* transcription, an enrichment of Bmal1 and Vps15 was detected on the *Ppat* promoter in the liver of WT mice (Fig. 6h). Moreover, consistent with decreased nuclear levels of Bmal1 in the livers of Vps15LKO mice, its enrichment on the *Ppat* promoter was abrogated (Extended Data Fig. 8f). Together, these findings in liver and primary hepatocytes suggest that Vps15 and Bmal1 cooperate for the transcriptional control of de novo purine synthesis.

### Acute Vps15 depletion in liver inhibits Bmal1

To demonstrate the in vivo impact of acute Vps15 deletion on Bmal1 activity, we developed hepatocyte-specific tamoxifen-inducible

---

**Fig. 6 | Vps15 controls de novo purine synthesis in the liver. a**, Number of cycling liver metabolites in 5-week-old Vps15LKO and WT male mice (*n* = 5). The percentage of metabolites oscillating in the WT are indicated. **b**, Top six oscillating metabolic pathways in WT liver. All metabolites are listed in Supplementary Table 6. **c**, Representation of de novo purine synthesis. $^{15}$N-glutamine-amide entry is shown. **d**, Ratio of liver IMP to PRPP in the samples in **a** (*n* = 4 for Vps15LKO$_{ZT12}$ and *n* = 5 for all other groups). The mice were fed ad libitum and kept under a 12-h light–dark regimen (grey shading). **e**, $^{15}$N-glutamine-amide incorporation in IMP at ZT12 in the livers of 5-week-old WT and Vps15LKO male mice (*n* = 6 for WT and *n* = 12 for Vps15LKO). **f**, Left: $^{15}$N-glutamine-amide incorporation in AML12 cells expressing GFP, small hairpin RNA (shRNA) targeting *Bmal1* (shBmal1) or *Vps15* (shVps15; *n* = 9 independent repeats). Right: depletion controlled in representative immunoblot. **g**, Relative liver *Ppat* expression (*n* = 3 for Vps15LKO$_{ZT0}$, *n* = 5 for WT$_{ZT12}$ and *n* = 4 for all other groups). The mice were fed ad libitum and kept under a 12-h light–dark

regimen (grey shading). **h**, ChIP–qPCR of Bmal1 and Vps15 enrichment on the *Ppat* promoter in the liver of 5-week-old WT male mice (ZT6; *n* = 6). **i**, Top: bioluminescence recordings of *Nr1d1-Luc* in the liver of a representative four-month-old male *TtrCre*[+];*Vps15*[f/f] mouse kept in constant darkness (pre-tamoxifen, WT) for 9 days, treated with tamoxifen for 5 days and monitored for 14 days following tamoxifen-induced Vps15iLKO (post-tamoxifen; top). Bottom: locomotor activity is shown (bottom). **j**, Periodograms (FFT analysis) of the data in **i**. **k**, ChIP–qPCR of Bmal1 and Vps15 promoter enrichment in the liver of 5-week-old male mice (ZT6) that were fed ad libitum or fasted for 24 h (*n* = 3 for fast, *n* = 4 for fed and *n* = 6 for IgG). **d–h,k**, Data are the mean ± s.e.m. *\*P* < 0.05; two-tailed unpaired Student's *t*-test. **l**, Functions of class 3 PI3K in vesicular trafficking to lysosomes and as coactivator of the circadian clock for de novo purine synthesis (created with BioRender.com). **a,g**, Rhythmicity was determined using JTK_CYCLE (Supplementary Table 1). Source data and unprocessed blots are provided.

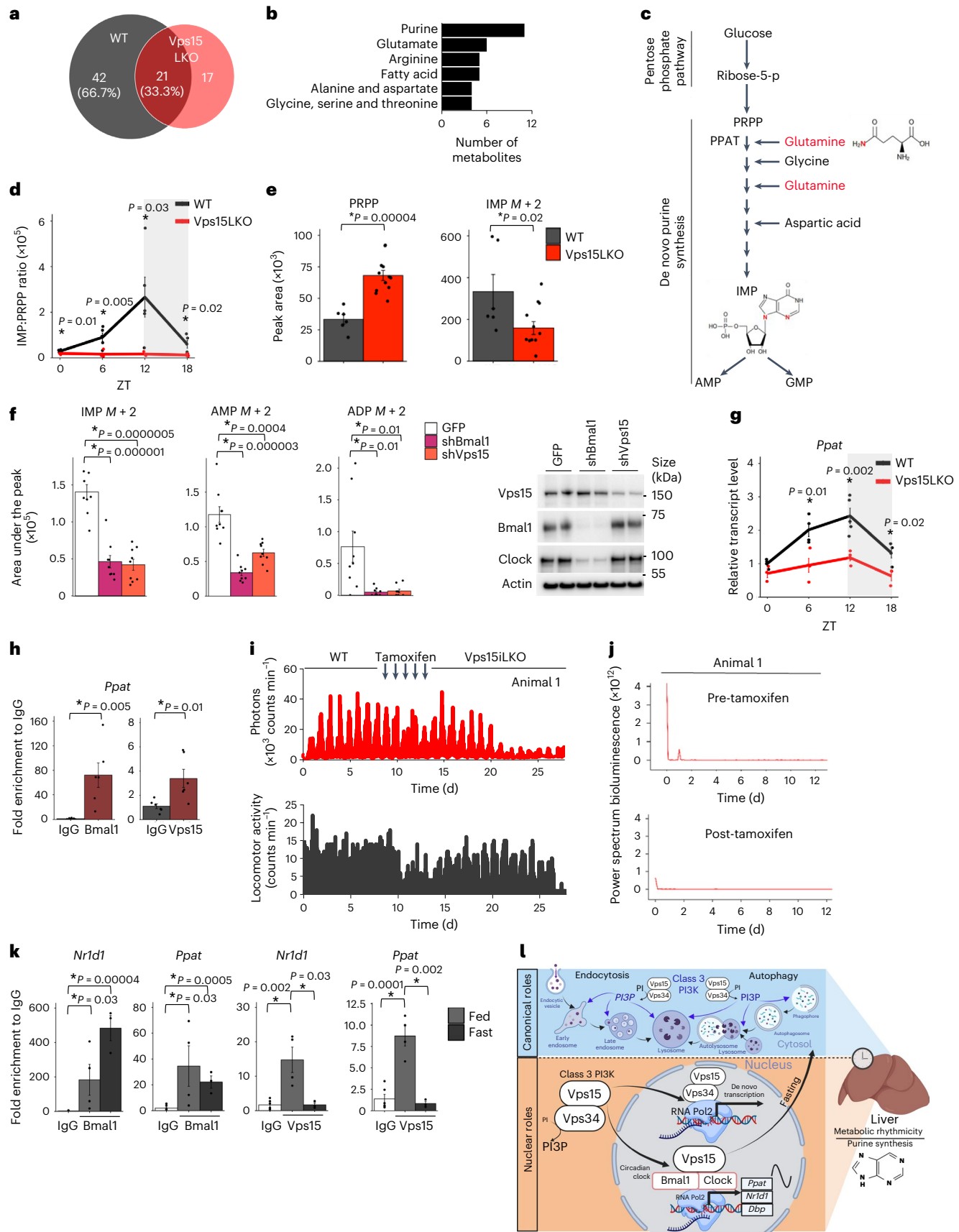

Vps15-mutant mice (*TtrCre*[+];*Vps15*[f/f]mice; hereafter referred as Vps15iLKO). Acute deletion of Vps15 in the hepatocytes of these mice resulted in decreased levels of Vps15 and Vps34 and manifested in autophagy block, seen as p62 and LC3 accumulation (Extended Data Fig. 9a). Immunoblot analyses in total liver extracts of Vps15iLKO mice showed that the Bmal1 and Rev-Erbα proteins were still rhythmic, albeit with decreased amplitude (Extended Data Fig. 9b). Moreover, similar to findings in MEFs, the nuclear levels of Bmal1 and Clock proteins were largely unmodified in Vps15iLKO mouse livers (Extended Data Fig. 9c,d). The expression pattern of Rev-Erbα in nuclear and total extracts was similar and showed decreased levels in the livers of Vps15iLKO mice (Extended Data Fig. 9c). Consistent with these data, in vivo biolumi-nescence analyses in freely moving mice one week after tamoxifen injection showed loss of *Nr1d1-Luc* rhythmicity in the livers of Vps15iLKO mice without changes in their locomotor activity (Fig. 6i, Extended Data Fig. 9e and Supplementary Table 7). Fast Fourier transform (FFT) analysis of bioluminescence profiles showed a 24 h FFT peak in the control pre-tamoxifen animals, which was abrogated following Vps15 deletion (Fig. 6j and Extended Data Fig. 9f). This transcriptional arrhyth-micity in Vps15iLKO mice was accompanied by decreased expression of the Bmal1 targets, most of which lost rhythmicity (Extended Data Fig. 9g and Supplementary Table 1). Moreover, similar to MEFs, acute hepatic depletion of Vps15 did not modify the enrichment of Bmal1 on the promoters of circadian clock-related genes (*Nr1d1* and *Dbp*) or *Ppat* (Extended Data Fig. 9h).

### Fasting inhibits Vps15 recruitment to Bmal1-bound regions

Given that the zenith of IMP levels (ZT12) corresponds to the onset of feeding, we hypothesized that feeding-fasting might regulate *Ppat* expression via the Bmal1–Vps15 axis. Consistent with previous findings that fasting repressed Bmal1 activity[58], the transcript levels of its targets were decreased in the livers of fasted control mice (Extended Data Fig. 10a). Notably, they were lower and not responsive to fasting in Vps15LKO mice (Extended Data Fig. 10a). However, neither the levels of nuclear Bmal1 protein nor its chromatin recruitment to the pro-moters of *Nr1d1* and *Ppat* genes were modified by fasting (Fig. 6k and Extended Data Fig. 10b). At the same time the nuclear levels of Vps15 were decreased in the livers of fasted mice and its recruitment to pro-moters of Bmal1 target genes (*Nr1d1* and *Ppat*) was abrogated with fasting (Fig. 6k and Extended Data Fig. 10b). Together with the transcrip-tional and metabolomic findings, these data back the role of Vps15 as a transcriptional coactivator of Bmal1 for the expression of circadian clock genes and the key enzyme in de novo purine synthesis (Fig. 6l).

## Discussion

Our results establish a model in which the essential subunit of class 3 PI3K, Vps15, functionally interacts with the circadian clock in tran-scriptional control of metabolic rhythmicity in liver and specifically for de novo purine synthesis (Fig. 6l). These findings suggest that class 3 PI3K acts upstream of the clock at multiple levels. First, as recently reported, class 3 PI3K might be engaged in the selective autophagy of Cry repressors of the Bmal1–Clock complex[13]. Second, our findings in Vps34-mutant cells and use of Vps34-lipid-kinase inhibitors point to a potential role of PI3P metabolism in the control of *Per2* and *Dbp* transcription. A previous report suggested nuclear metabolism of different phosphoinositides, including PI3P[59]. Yet the role of PI3P in transcription remains largely unexplored. Finally, we report that Vps15 interacts and coactivates Bmal1 on its early target genes (*Nr1d1* and *Dbp*) and drives diurnal transcription of a key metabolic enzyme in de novo purine synthesis, Ppat. Thus, our study proposes a paradigm that, in addition to its cytosolic roles in vesicular trafficking and autophagy, class 3 PI3K may directly couple cellular energy status with metabolic anticipation through transcriptional coactivation of the circadian clock. Beyond this, our surprising findings of class 3 PI3K interaction with RNA Pol2 and its co-localization with de novo transcription sites

opens to future work on mechanisms of its active nuclear transport and chromatin recruitment, its nuclear interactome and its specific gene targets under different physiological and disease conditions. Furthermore, in addition to liver, recent circadian metabolomics analy-ses of brown adipose tissue and skeletal muscle have suggested that a high caloric diet and Bmal1 deletion impact purine metabolism[53,60]. Thus, given that class 3 PI3K is ubiquitously expressed, its functional interaction with Bmal1 for purine metabolism might expand beyond the liver into other organs and could be relevant in metabolic diseases. Finally, recent findings in a liver-specific reconstitution model of Bmal1 in whole-body *Bmal1*-mutant mice challenged the liver clock autonomy upstream of de novo purine synthesis. This suggests that purine metab-olism is controlled by other clocks in the body (central or peripheral clocks in other organs)[54]. Given that defects of purine rhythmicity in mice bearing liver-specific *Vps15* deletion resembled the phenotype of whole-body *Bmal1* mutants, a plausible scenario is that hepatic class 3 PI3K might act upstream of systemic cues for clock communication between different organs to achieve whole-body metabolic synchrony.

## Online content

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

[1]Institut Necker-Enfants Malades (INEM), Paris, France. [2]INSERM U1151/CNRS UMR 8253, Paris, France. [3]Université Paris Cité, Paris, France. [4]The Thoracic and Endocrine Surgery Division, Department of Surgery, University Hospital of Geneva, Geneva, Switzerland. [5]Department of Cell Physiology and Metabolism, University of Geneva, Geneva, Switzerland. [6]Diabetes Center, Faculty of Medicine, University of Geneva, Geneva, Switzerland. [7]Institute of Genetics and Genomics in Geneva (iGE3), Geneva, Switzerland. [8]Platform for Metabolic Analyses, Structure Fédérative de Recherche Necker, INSERM US24/CNRS, UAR 3633, Paris, France. [9]Computational Discovery Research, Institute for Diabetes and Obesity (IDO), Helmholtz Diabetes Center (HDC), Helmholtz Zentrum München—German Research Center for Environmental Health, Neuherberg, Germany. [10]Division of Metabolic Diseases, Department of Medicine, Technische Universität München (TUM), Munich, Germany. [11]German Center for Diabetes Research (DZD), Neuherberg, Germany. [12]Department of Molecular and Integrative Physiology, Caswell Diabetes Institute, University of Michigan Medical School, Ann Arbor, MI, USA. [13]Department of Physiology, Toho University School of Medicine, Tokyo, Japan. [14]Department of Chemistry, School of Science, The University of Tokyo, Tokyo, Japan. [15]Institute of Cardiovascular and Metabolic Diseases (I2MC), INSERM-UMR 1297, University Paul Sabatier, Toulouse, France. [16]These authors contributed equally: Chantal Alkhoury, Nathaniel F. Henneman. ✉e-mail: ganna.panasyuk@inserm.fr

## Methods

The conducted research complies with all relevant ethical regulations; all animal studies were performed by authorized users in compliance with ethical regulations for animal testing and research. The study was approved by the ethical committee of the Université Paris Cité.

### Reagents

The following primary antibodies were used: anti-Vps15 (1:1,000 (Abnova, H00030849-M02) and 1:200 for the proximity ligation assay (Genetex, GTX108953)), anti-p62 (1:1,000; Abnova, H00008878-MO1), anti-β-actin (1:5,000; Sigma, A5316), anti-tubulin (1:1,000; Sigma, T9026), anti-β-catenin (1:500; BD Biosciences, 610153), anti-lamin A/C (1:1,000; Cell Signaling Technology, 2032), anti-LC3 (1:1,000; NanoTools, 0231-100/LC3-3-5-5F10), anti-GAPDH (1:1,000; Santa Cruz Biotechnology, sc-25778), anti-Rev-Erbα (1:1,000; Cell Signaling Technology, 13418S), anti-Bmal1 (1:250 for immunohistochemistry and 4 µg per immunoprecipitation for ChIP (Abcam, ab3350); 1:1,000 for western blots and 1:200 for immunofluorescence imaging (Cell Signaling Technology, 14020S)), anti-Clock (1:1,000; Cell Signaling Technology, 5157), anti-HIS-Tag (1:1,000; Proteintech, 66005-12-Ig), anti-IPOA5 (1:1,000; Proteintech, 18137-1-AP), anti-Cry1 (1:500; Origene, TA342728), anti-RNA Pol2 total (1:1,000; Active Motif, 39097), anti-RNA Pol2 phospho-S5 (1:1,000; Chromotek, 3E8-1), anti-Vps34 (1:1,000; Cell Signaling Technology, 4263), anti-histone H3 (1:3,000; Cell Signaling Technology, 4499), anti-Flag (1:1,000; Sigma, F1804), anti-Hsp90 (1:1,000; Proteintech, 13171-1-AP), normal IgG rabbit isotype control (4 µg per immunocippitation for ChIP, Cell Signaling Technology, 3900), normal IgG mouse (1:5,000; Santa Cruz Biotechnology, sc-2025), anti-mouse-IgG horseradish peroxidase (HRP)-linked antibody (1:5,000; Cell Signaling Technology, 7076) and anti-rabbit-IgG HRP-linked antibody (1:5,000; Cell Signaling Technology, 7074). Plasmids expressing VPS15WT–Flag, VPS15-D165R and VPS15-E200R-Flag were purchased from MRC PPU Reagents and Services. The pG4-E6s-luc (E-box-Luc) plasmid was a gift from S. Brown (Addgene, plasmid 46324)[61]. Flag–mCry1ER-pBABEpuro was a gift from A. Sancar (Addgene, plasmid 61429)[62]. The adenoviral vectors expressing GFP, GFP-Cre, VPS15WT-6HIS-V5 and shRNAVps15 were described previously[32,37]. The adenoviral vector expressing VPS15-E200R–Flag was custom-made by Vector Biolabs. The following Vps34 inhibitors were used: SAR405 (Selleckchem, S7682), Vps34-IN1 (Selleckchem, S7980) and PIK-III (Cayman, 17002). The adenoviral vectors expressing shRNA targeting Bmal1 and Bmal1 were described previously[63]. The plasmids expressing HIS-tagged Bmal1 fragments were described previously[46]. The plasmids expressing 3×Flag–VPS15 full-length and VPS15 domains were custom-made by Vectorbuilder. The constructs were designed corresponding to the domains in the VPS15 protein defined in[26], namely the kinase domain spanning amino acids 20–275, the HEAT domain spanning amino acids 300–797 and the WD40 domain spanning 975–1358. A 3×Flag (DYKDHDGDYKDHDIDYKDDDDK) tag was added at the amino-terminal region of each deletion construct.

### Animals

The mouse line AlbCre⁺;Vps15f/f with liver-specific Vps15 knockout was obtained by crossing Vps15f/f mice (strain 022624, Jackson Laboratory) with AlbCre⁺ mice as reported[32]. Male mice (5 weeks old) were used for the experiments. For the generation of inducible hepatocyte-specific Vps15 knockout mice, Vps15f/f mice were crossed with the mouse line expressing Cre recombinase under the transthyretin promoter[64]. The resulting TtrCre⁺;VPS15f/f (named Vps15iLKO, for inducible liver knockout) and TtrCre⁻;Vps15f/f (named WT) mice were used in this study. To knockout the Vps15 gene, 12–16-week-old male and female mice were administered an intraperitoneal tamoxifen injection (1.5 mg per mouse) over five consecutive days. Efficient deletion of Vps15 in hepatocytes was observed 10 d post injection. The mice were randomly allocated to experimental groups and at least three animals were used for each

condition. All animals used in the study were fed standard chow diet (Teklad Global 2918, 18% protein, irradiated) ad libitum and kept under a 12 h–12 h (8:00–20:00) light on–off cycle under ambient temperature (21–22 °C) and humidity of 50–60%. Animals were euthanized at the ZT time indicated in each experiment. The study was approved by the ethical committee of Université Paris Cité (authorization numbers APAFIS#32312 and APAFIS#14968).

### Cell culture

Mouse embryonic fibroblast cells were immortalized by passaging[65]. Vps15f/f MEFs obtained from a pool of embryos from two different females were passaged every 3 d before spontaneous transformation[37]. The Vps34f/f MEFs were prepared following the same protocol. All cell cultures were tested bi-weekly and validated as mycoplasma-free. To obtain the Vps15−/− and Vps34−/− MEFs, cells were plated (10 × 10³ cells cm−²) in High-glucose DMEM medium (Gibco) containing 100 units (U) ml−¹ penicillin, 100 mg ml−¹ streptomycin, 10% fetal bovine serum (FBS; Dutscher, S-1810) and transduced 12 h later with CRE-GFP or GFP (as a control) adenovirus at a multiplicity of infection of 500. The cells were incubated for 3 d before plating for the synchronization or rescue experiments. For the rescue experiments, Vps15-depleted MEFs were transduced with VPS15WT-6HIS-V5- or VPS15-E200R–1×Flag-expressing adenoviral vectors and synchronized 48 h after transduction. The MEFs were treated with the inhibitors SAR405, PIK-III and Vps34-IN1 at a dose of 5 µM after synchronization; the cells were exposed to the inhibitor until collection. When indicated, fully confluent cells were synchronized by treatment with 100 nM dexamethasone (Sigma) in serum-free medium for 1 h. The dexamethasone was removed after synchronization and the cells were kept in serum-free medium for the indicated times before collection. Primary hepatocytes from male WT mice (6–8 weeks old) were isolated by liver perfusion[32]. The hepatocytes were plated (12 × 10⁴ cells cm−²) in Williams medium (Life Technologies) supplemented with 20% FBS (Dutscher, S-1810-500), penicillin (100 U ml−¹), streptomycin (100 µg ml−¹) and amphotericin B (Fungizone; 250 ng ml−¹). The cells were transduced with adenoviral vectors 12 h after plating and samples were collected for analyses 36 h post infection. AML12 cells were purchased from the ATCC (CRL-2254) and cultured in DMEM:F12 (Gibco) medium supplemented with 100 U ml−¹ penicillin, 100 mg ml−¹ streptomycin, 10% FBS, 10 µg ml−¹ insulin, 5.5 µg ml−¹ transferrin, 5 ng ml−¹ selenium and 40 ng ml−¹ dexamethasone. HEK293T cells were purchased from the ATCC (CRL-3216) and cultured in High-glucose DMEM supplemented with 100 U ml−¹ penicillin, 100 mg ml−¹ streptomycin and 10% FBS.

### Subcellular fractionation

Cytosolic and soluble nuclear fractions were prepared from 50 mg frozen tissue[66]. Briefly, tissue powder was homogenized in 1 ml hypotonic buffer (10 mM HEPES pH 7.9 and 0.5 mM dithiothreitol) and incubated on ice for 30 min with vortexing for 10 s every 5 min. NP-40 was added to the samples to a final concentration of 0.4% and the samples were incubated on ice for 2 min, followed by vortexing for 10 s. Nuclei were pelleted by centrifugation at 800g for 1 min at 4 °C. The supernatant was collected as the cytosolic fraction. Before nuclei extraction, the pellet was washed as follows: three times with 1 ml hypotonic buffer, once with 1 ml hypotonic buffer complemented with 14% NP-40 and three times with 1 ml hypotonic buffer. The nuclear pellet was resuspended in 300 µl hypertonic buffer (20 mM HEPES pH 7.9, 0.5 mM dithiothreitol, 0.42 M NaCl, 25% glycerol and 0.2 mM EDTA pH 8) and incubated on ice for 40 min with vortexing for 10 s every 10 min. The soluble nuclear fraction was recovered by centrifugation at 10,000g for 1 min at 4 °C. For euchromatin extraction, nuclei were extracted with isotonic buffer (10 mM Tris–HCl pH 8.0, 15 mM NaCl, 60 mM KCl and 1.5 mM EDTA) before euchromatin extraction (10 mM Tris–HCl pH 8.0, 250 mM NaCl and 1 mM EDTA)[67].

## Metabolic flux and targeted metabolomics analysis

For metabolite tracing, mice were intraperitoneally injected with 0.75 mg g$^{-1}$ body weight of $^{15}$N-glutamine-amide (Cambridge Isotope Laboratories, NLM-557-1) at ZT12 and their livers were harvested 24 h later. For metabolite tracing in AML12 cells, the cells were transduced with adenoviral vectors expressing GFP, or shRNA to *Bmal1* or *Vps15* for 48 h before labelling. Labelling was performed by washing cells once with warm D-PBS, followed by the addition of DMEM-F12 no-glutamine medium (Gibco) complemented with 10% FBS (Dutscher, S-1810-500). Either L-glutamine or $^{15}$N-glutamine (584 mg l$^{-1}$) was added to the respective experimental conditions and the cells were incubated for 30 min before being flash frozen in liquid nitrogen for analysis through liquid chromatography with mass spectrometry. Targeted metabolomics analyses were performed following a previously published protocol[68] using the extracts prepared with 50% methanol, 30% acetonitrile and 20% water. The volume of extraction solution added was calculated taking into account the weight of powdered tissue (60 mg ml$^{-1}$). After the addition of the extraction solution, the samples were vortexed for 5 min at 4 °C and then centrifuged at 16,000$g$ for 15 min at 4 °C. The supernatants were collected and analysed by liquid chromatography with mass spectrometry using a SeQuant ZIC-pHilic column (Merck) for the liquid-chromatography separation. Mobile phase A consisted of 20 mM ammonium carbonate plus 0.1% ammonia hydroxide in water. Mobile phase B consisted of acetonitrile. The flow rate was kept at 100 ml min$^{-1}$ and the gradient was: 0 min, 80% phase B; 30 min, 20% of phase B; 31 min, 80% of phase B and 45 min, 80% of phase B. The mass spectrometer (QExactive Orbitrap, Thermo Fisher Scientific) was operated in a polarity switching mode and metabolites were identified using TraceFinder Software (Thermo Fisher Scientific). For the analyses, the metabolomics data were normalized using the median normalization method. The MetaboAnalyst 3.0 software was used to conduct statistical analyses and generate heatmaps, and an unpaired two-sample Student's $t$-test was chosen to perform the comparisons. The algorithm for heatmap clustering was based on the Pearson distance measure for similarity and the Ward linkage method for biotype clustering.

## ChIP

Chromatin immunoprecipitation was performed with the following antibodies: anti-BMAL1 (Abcam, ab3350), anti-Vps15 (Abnova, H00030849-M02), rabbit IgG (Santa Cruz Biotechnology, sc-3888), anti-RNA Pol2 CTD phospho-S5 (Abcam, ab5408) and mouse IgG (Santa Cruz Biotechnology, sc-2025). Chromatin immunoprecipitation from liver was performed as reported[54]. Briefly, liver tissue was powdered and fixed in 1% formaldehyde (Sigma) for 15 min, after which it was quenched in glycine (125 mM). The chromatin was sonicated to achieve fragments of 200–500 bp. Antibodies were added to the chromatin, followed by overnight incubation at 4 °C, after which Dynabeads protein A (Thermo Fisher Scientific, 10001D) were added to the immune complexes for 4 h and then washed once with Low Salt RIPA buffer (150 mM NaCl, 50 mM Tris–HCl pH 8.0, 1 mM EDTA pH 8.0, 1% Triton X-100, 0.1% SDS and 0.1% sodium deoxycholate (NaDOC)), once with High Salt RIPA buffer (500 mM NaCl, 50 mM Tris–HCl pH 8.0, 1 mM EDTA pH 8.0, 1% Triton X-100, 0.1% SDS and 0.1% NaDOC), once with liver-LiCl Wash Buffer (250 mM LiCl, 10 mM Tris–HCl pH 8.0, 1 mM EDTA, 0.5% NP-40 and 0.5% NaDOC) and twice with TE Buffer (10 mM Tris–HCl pH 8.0 and 1 mM EDTA pH 8.0). The immune complexes were eluted with Elution Buffer (10 mM Tris–HCl pH 8.0, 300 mM NaCl, 5 mM EDTA pH 8.0 and 0.5% SDS) and de-crosslinked overnight at 65 °C. Following RNase and Proteinase K treatment, the DNA fragments were purified using phenol chloroform:isoamyl alcohol (Thermo Fisher Scientific, 15593049) extraction and Phase lock gels (VWR 10847-802). For cells, fixation was done with 1% formaldehyde (Sigma) for 10 min and then quenched in glycine (125 mM). The chromatin sonication and immune-complex pulldown using Dynabeads protein A (Thermo Fisher Scientific, 10001D)

steps were performed as for liver. The immune complexes were washed once with Isotonic Buffer (10 mM Tris–HCl pH 8.0, 150 mM NaCl, 1% Triton X-100 and 0.1% NaDOC), once with Isotonic Buffer KCl (10 mM Tris–HCl pH 8.0, 150 mM KCl, 1% NP-40 and 1% NaDOC), once with High Salt Buffer (10 mM Tris–HCl pH 8.0, 500 mM NaCl, 0.5% Triton X-100 and 0.1% NaDOC), once with cells-LiCl Wash Buffer (20 mM Tris–HCl pH 8.0, 250 mM LiCl, 1 mM EDTA pH 8.0, 0.5% NP-40 and 0.5% NaDOC), once with NaCl Wash Buffer (20 mM Tris–HCl pH 8.0, 150 mM NaCl, 1 mM EDTA pH 8.0 and 0.1% NP-40) and twice with TE Buffer, after which the immune complexes were eluted with Elution Buffer (100 mM NaCl, 100 mM NaHCO$_3$ and 1% SDS) and de-crosslinked overnight at 65 °C (ref. 69). Subsequent steps for DNA purification were the same as for the liver ChIP. The relative immunoprecipitated DNA was determined by quantitative PCR with reverse transcription using the $2^{-\Delta\Delta C_T}$ method, with IgG samples as the enrichment controls. The primer sequences are listed in Supplementary Table 8.

## ChIP DNA library preparation and sequencing

Sample testing included concentration, sample integrity and purity. Concentration was determined using a fluorometer (Qubit Fluorometer, Invitrogen). Sample integrity and purity were determined using an Agilent Technologies 2100 Bioanalyzer. ChIP DNA was subjected to end-repair and then 3′ adenylation. Adaptors were ligated to the ends of the 3′-adenylated fragments. The PCR products were purified and selected using a Agencourt AMPure XP-medium kit. Double-stranded PCR products were heat denatured and circularized by the splint oligonucleotide sequence. The single-stranded circle DNA was formatted as the final library. The library was quantified using a Qubit ssDNA kit. The library was amplified to make DNA nanoballs. The DNA nanoballs were loaded into the patterned nanoarray and single-end base reads were generated by sequenced combinatorial Probe-Anchor Synthesis. Sequencing was performed with the DNBSEQ-400 sequencer (Beijing Genomics Institute).

## ChIP–seq data analysis

Data filtering was performed with SOAPnuke to remove adaptor sequences and low-quality reads. The data filtering parameter was: SOAPnuke filter -l 5 -q 0.5 -n 0.1 -Q 2 -c 40. Clean reads were aligned to the mm10 mouse genome using Bowtie2 version 2.4.4 and SAMtools version 1.13 was used to index and sort binary alignment map (BAM) files[70,71]. Heatmaps were generated using the deepTools2 plotHeatmap function[72]. Peak calling was performed for all samples using MACS2 version 2.2.7.1 (ref. 73) for ChIP-BMAL1 with the following parameter: callpeak -t -c -f BAM -g mm -p 0.01; H3K27Ac peak calling was performed using: callpeak -t -c -f BAM -g mm –broad and ChIP-Vps15 was performed using: callpeak -t -c -f BAM -g mm -p 0.001 –broad –nomodel. Differential peak calling analysis was performed using DiffBind 3.10.0, an R package[74]. Differentially bound peaks were annotated and pathway analysis performed using ChIPseeker and clusterProfiler[75,76]. Motif analysis and $P$ value ranked motif list was performed using HOMER with the following parameter: findMotifsGenome.pl -mask -size 200 (ref. 77; Supplementary Table 3).

## RNA-seq and analysis

Total RNA extracted from the livers of 5-week-old Vps15LKO and control male mice (fed ad libitum) collected at ZT6 was sequenced using the DNBSEQ-400 system. To obtain clean reads, the raw reads were filtered following sequencing using the SOAPnuke parameters: -n 0.03 -l 20 -q 0.4 -A 0.28. Roughly 41 million clean reads were obtained per sample (Beijing Genomics Institute, China). FASTQ files were mapped to the ENSEMBL (mouse GRCm38/mm10) reference using HISAT2 and counted using featureCounts from the Subread R package. Read-count normalizations and group comparisons were performed by three independent and complementary statistical methods: Deseq2, edgeR and LimmaVoom. Flags were computed from

counts normalized to the mean coverage. All normalized counts <20 were considered as background (flag 0) and ≥20 as signal (flag = 1). P50 lists used for the statistical analysis regroup the genes showing flag = 1 for at least half of the compared samples. Direct comparisons were made between WT and Vps15LKO animals to generate lists of differentially expressed genes (Supplementary Table 4). Pathway enrichment was performed using Gene Set Enrichment Analysis version 4.2.2 (refs. [78],[79]). Deep-sequencing (ChIP–seq and RNA-seq) data have been deposited in the Gene Expression Omnibus under accession code GSE229551.

## Quantitative PCR with reverse transcription

Total RNA was isolated from liver tissue using an RNAeasy lipid tissue mini kit (Qiagen) and from cells using an RNeasy mini kit (Qiagen). Single-strand complementary DNA was synthesized from 1 µg total RNA using 125 ng random hexamer primers and SuperScript II (Life Technologies). Quantitative PCR with reverse transcription was performed on a QuantStudio 1 instrument (Thermo Fisher Scientific) using iTaq Universal SYBR Green Supermix (BioRad). The relative amounts of the messenger RNAs studied were determined by means of the $2^{-\Delta\Delta C_T}$ method, with *meEf1a1*, *Gapdh*, *Pinin* and *18S* as reference genes (geometric mean) and the control treatment or control genotype as the invariant control. The primer sequences are listed in Supplementary Table 8.

## Protein extraction, immunoblotting and immunoprecipitation

To prepare protein extracts for immunoblot analysis, cells were washed twice with cold PBS, scraped from the dishes in lysis buffer containing 20 mM Tris–HCl pH 8.0, 5% glycerol, 138 mM NaCl, 2.7 mM KCl, 1% NP-40, 20 mM NaF, 5 mM EDTA, 1×protease inhibitors (Roche) and 1×PhosphoStop inhibitors (Roche). The same buffer was used to prepare protein extracts from liver tissue. Homogenates were centrifuged at 12,000g and 4 °C for 10 min. For immunoprecipitation, 500 µg of cleared protein extract was incubated with 2 µg antibody for 3 h at +4 °C. The immune complexes were then pulled down using Protein G Sepharose beads (GE) for 2 h, followed by four washes with the extraction buffer. The protein complexes were eluted by boiling the beads in 1×SDS sample buffer for 10 min. The protein extracts or immunoprecipitated eluates were resolved by SDS–PAGE before transfer onto polyvinylidene fluoride membrane, followed by incubation with primary antibodies and HRP-linked secondary antibodies. TGX stain-free gels were used following the recommendations of the manufacturer (Bio-Rad, 1610182). Immobilon western chemiluminescent HRP substrate (Millipore) was used for the detection. The images were acquired on a ChemiDocTM imager (BioRad).

## Immunohistochemistry and immunofluorescence microscopy of liver tissue

For immunohistochemical analysis or immunofluorescence microscopy, liver tissue was fixed overnight in phosphate-buffered 10% formalin and embedded in paraffin. Sections (4 µm) of the fixed tissue were cut and after citrate retrieval of the antigen processed for immunohistochemical analyses with anti-BMAL1 (Abcam). Permeabilization was achieved with 0.1% Triton X-100 for 20 min, followed by blocking with a 3% solution of goat pre-immune serum in Emerald solution. Slides were treated with anti-BMAL1 (Abcam) overnight and the secondary antibodies used were anti-rabbit IgG Alexa Fluor 568 (Life Technologies) or biotinylated anti-rabbit IgG (Vector), followed by VECTASTAIN Elite ABC-kit peroxidase (Vector). The sections were counterstained with haematoxylin for immunohistochemical analyses and the slides were digitalized with the NanoZoomer S210 (Hamatsu) and visualized using the NDP.view2 software. Fluorescence microscopy was performed using an inverted microscope (Zeiss Apotome 2) with a ×40 oil-immersion objective.

## Immunofluorescence microscopy

For the fluorescence microscopy analyses, MEF cells were cultured on four-well Millipore EZ glass slides. After 24 h the cells were washed once with PBS and fixed with 4% paraformaldehyde for 10 min. Permeabilization was achieved with 0.1% Triton X-100. For the CSK treatment[80], the cells were washed twice with PBS before incubation in CSK buffer (10 mM PIPES pH 7.0, 100 mM NaCl, 300 mM sucrose, 3 mM MgCl$_2$ and 0.7% Triton X-100) for 2 min at room temperature. The cells were then washed twice with PBS and fixed with 4% formol for 15 min. The control cells were permeabilized with 0.7% Triton X-100 in PBS following fixation. Blocking was done with a 3% solution of the appropriate pre-immune serum. The slides were treated with primary antibodies overnight and the secondary antibodies used were: donkey anti-rabbit IgG (H + L) highly cross-absorbed secondary antibody, Alexa Fluor Plus 488 (1:200; Thermo Fisher Scientific. A32790); donkey anti-mouse IgG (H + L) highly cross-absorbed secondary antibody, Alexa Fluor Plus 488 (1:200; Thermo Fisher Scientific, A32766); donkey anti-rabbit IgG (H + L) highly cross-absorbed secondary antibody, Alexa Fluor Plus 568 (1:200; Thermo Fisher Scientific, A10042); donkey anti-mouse IgG (H + L) highly cross-absorbed secondary antibody, Alexa Fluor Plus 568 (1:200; Thermo Fisher Scientific, A10037) and goat anti-rat IgG (H + L) highly cross-absorbed secondary antibody, Alexa Fluor Plus 647 (1:200; Thermo Fisher Scientific, A-21247). Fluorescence microscopy was performed using an inverted microscope (Zeiss Apotome 2 or Zeiss Axio Observer Z1 with Yokogawa CSU-X1 Spinning Disk) using ×40 or ×63 oil-immersion objectives.

## PI3P detection

The pGEX-2TK-FYVE/HRS-WT plasmid was a gift from M. Lemmon (Yale, USA). The FYVE/HRSMut (R24A/K25A/R29A) was obtained by directed mutagenesis and both constructs were first subcloned into pmCherry-C1 (Clontech; Xho1 and BamH1 sites) and then subcloned into pGEX-4T-1 (Pharmacia Biotech; BamH1 and Sal1). The GST-mCherry-FYVE-domains (HRS) were expressed in *Escherichia coli* BL21(DE3) bacteria incubated overnight at 18 °C with 0.5 mM isopropylthiogalactoside and purified by affinity chromatography using Glutathione Sepharose 4B beads (GE Healthcare) according to the manufacturer's instructions. The GST-mCherry-FYVE-domains (HRS) were cleaved by thrombin, dialysed against 50 mM Tris pH 8.0, 100 mM NaCl solution and concentrated (Vivaspin Colum 5–10 kDa molecular-weight cut). Glycerol (10%) was added to the final recombinant protein before rapid snap-freezing in liquid nitrogen and storage at −80 °C until use. For the PI3P detection, cells were fixed with 3.7% formaldehyde, quenched with NH$_4$Cl for 10 min and permeabilized with 20 µM digitonin in PIPES-BS (20 mM PIPES pH 6.8, 137 mM NaCl and 2.7 mM KCl) for 5 min. After saturation with 10% pre-immune goat serum in PIPES-BS, the cells were incubated with the GST-mCherry-FYVE probes (50 µg ml$^{-1}$) for 2 h at +4 °C. After three washes with PIPES-BS, the cells were then fixed a second time with 3.7% formaldehyde and the nuclei were stained with DAPI present in mounting media. Slides were imaged using an inverted microscope (Zeiss Apotome 2 or Zeiss Axio Observer Z1 with Yokogawa CSU-X1 Spinning Disk) using ×40 or ×63 oil-immersion objectives.

## PLA assays

The PLA assays were performed in HEK293T cells (acquired from the ATCC, tested bi-weekly and validated as mycoplasma-free). The assays were carried out according to the manufacturer's instructions (Duolink in situ red starter kit mouse/rabbit; Sigma, DUO92101). Briefly, where indicated, the cells were transfected with the respective plasmids (Flag–VPS34 or Flag–BMAL1 together with GFP-expressing vector to visualize the transfected cells) 48 h before fixation with 4% paraformaldehyde for 20 min, followed by permeabilization with 0.1% Triton X-100 in PBS for 20 min at room temperature and blocking for 30 min at 37 °C. The slides were incubated with a pair of respective primary

antibodies: anti-Flag (1:200; Sigma, F1804), anti-Vps15 (1:200; Genetex, GTX108953) or anti-BMAL1 (1:200; Cell Signaling Technologies, 14020S) for 1 h at 37 °C. The incubation with PLA probe PLUS and MINUS conjugated with oligonucleotides was performed for 1 h at 37 °C. The terminal steps of ligation and amplification were performed at 37 °C for 30 min and 90 min, respectively. Samples treated with PLA probes without primary antibodies served as controls for background binding (negative control). Images were acquired using an inverted microscope (Zeiss Axio Observer Z1 with Yokogawa CSU-X1 spinning disk) with a ×40 oil-immersion objective and Z-stack acquisition. Quantification of the PLA signal was performed on independent fields of cells (at least 300 cells per condition) either in all cells (proximity between endogenous proteins) or in GFP+ cells (when at least one protein was transiently expressed). The data are presented as the number of puncta per field with the number of cells analysed specified in the figure legend.

### BrUTP labelling and detection
Primary hepatocytes and MEFs were plated on four-well EZ glass slides (Millipore). After 24 h, the MEFs were transfected with the respective plasmids of VPS15WT–Flag and VPS34WT–Flag. BrUTP labelling was performed 48 h following transfection. First, the cells were washed twice with warm PBS. The cells were then incubated with permeabilization buffer (20 mM Tris–HCl pH 7.4, 5 mM $MgCl_2$, 0.5 mM EGTA, 25% glycerol, 0.1% Triton X-100 (Sigma) and 1 mM phenylmethylsulfonyl fluoride) for 3 min at room temperature. The permeabilization buffer was gently aspirated off and Transcription buffer (100 mM KCl, 50 mM Tris–HCl pH 7.4, 10 mM $MgCl_2$, 0.5 mM EGTA, 25% glycerol, 2 mM ATP (Roche), 0.5 mM CTP (Roche), 0.5 mM GTP (Roche), 0.5 mM BrUTP (Sigma) and 1 mM phenylmethylsulfonyl fluoride) was added to the cells for 5 min at 37 °C. The cells were then washed once with warm PBS and fixed with 4% paraformaldehyde for 10 min. Antibodies to BrUTP (Roche, clone BMC9318) and RNA Pol2 phospho-S5 (1:1,000; Chromotek, 3E8-1) were used in immunofluorescence.

### Luciferase assay
The luciferase assay was performed in HEK293T cells. The cells were plated in 24-well plates and PEI-transfected with a mix of luciferase reporter construct (E-box-Luc; Addgene, 46324; 0.15 μg) and control plasmid expressing β-galactosidase (0.05 μg) together with BMAL1–Flag (0.15 μg) and CLOCK–Flag (0.15 μg) or as a control empty vector (0.3 μg); when indicated, the cells were co-transfected with 0.4 μg plasmid expressing VPS15WT–Flag, VPS15-E200R–Flag, VPS34WT–Flag, p300 or CBP. In the assay with co-expression of Cry1, 0.1 μg Flag–mCry1ER-pBABEpuro (Addgene, 61429) or empty vector was co-expressed. After transfection (22–24 h), the cells were collected, the extract was prepared with 1×Passive lysis buffer (Promega), and the luciferase reporter activity was measured and normalized to β-galactosidase activity as reported previously[81].

### In vivo and in vitro bioluminescence recording
For in vivo bioluminescence recording, *Nr1d1*-luciferase-expressing adenoviral construct (*Nr1d1-Luc*) was used[82]. Luciferin administration (in drinking water), luciferase reporter delivery, bioluminescence recording as well as data analysis were performed as in ref. 83; briefly, 48 h following the administration of *Nr1d1-Luc* adenoviral vectors, *TtrCre*+;*Vps15*f/f mice were transferred to a RT-Bioluminicorder and the bioluminescence monitoring started. After 9 days of recording, the mice were injected with tamoxifen for five consecutive days under red light and the recording was continued until days 20–25. For the bioluminescence analyses of cells, *Vps15*f/f MEFs were transduced with *Nr1d1-Luc* adenoviral vectors. After 48 h, the cells were plated for the synchronization at full confluent density. The cells were synchronized by treatment with 100 nM dexamethasone in serum-free high-glucose DMEM (Gibco) medium for 1 h. After synchronization, the medium was changed to serum-free high-glucose DMEM (Gibco) complemented

with the inhibitors SAR405, PIK-III and Vps34-IN1 at a dose of 2.5 μM and 5 μM as well as 100 μM luciferin. The continuous bioluminescence recordings were performed using a LumiCycler 96 luminometer (Actimetrics), allowing the recording of 24-well plates simultaneously. To analyse the circadian parameters of time series without the variability of magnitudes, raw data were processed in parallel graphs by moving average with a window of 24 h (ref. 84).

### Analyses of circadian rhythmicity, plots and schema
For datasets of time series, rhythmic metabolites, transcripts and proteins were detected by implementing JTK_CYCLE[85]. The period length, phase and amplitude were detected with the non-parametric 'MetaCycle' (JTK_CYCLE) algorithm implemented in R[86]. No modifications to the code were made for analysis. The selected parameters were evaluated as rhythmic with $P < 0.05$. Metabolite classes and pathways were identified using the R package 'MetaboAnalystR3.0' and crossed with published lists[87]. For data presentation, all data plots were made with ggplot2 using R studio. Where indicated, the transcript analysis around the clock was plotted using a LOESS regression curve fitted to the $\log_2$-normalized data. The summary schemas were created with BioRender.com.

### Statistics and reproducibility
Data are shown are the mean ± s.e.m. The numbers of distinct samples are presented in the figure legends. One- or two-way ANOVA with post-hoc Benjamini–Hochberg correction, or unpaired two-tailed Student's *t*-tests were applied for statistical analysis, as specified in the figure legends. For all experiments, results were considered significant for $P < 0.05$. All in vitro and in vivo assays were performed three times, unless specified otherwise in the legends. The accompanying quantification and statistics were derived from $n = 3$ independent replicates, unless specified otherwise in the legends. No data or animals were excluded from the analyses. In studies with animals, the '*n*' number corresponds to an individual mouse. Power analysis was used to determine the animal numbers to take into account previously observed magnitudes of response to the in vivo deletion of a gene of interest as well as previously reported observations of circadian behaviour for the B6/C57 strain of mice (expression and metabolomics analyses). We applied >80% power and error rate 5% (two-sided type 1) to detect >1.5 effect. Experiments in cells were carried out as biological replicates and each also included technical replicates. For the biological replicates, independent preparation of depleted cells (adenoviral infection) and independent treatments were used to ensure reproducibility. For the in vivo experiments, three mice per condition were analysed unless specified otherwise in the figure legends. The animals, including those treated with tamoxifen for gene deletion, were randomly allocated into experimental groups based on genotype. The breeding was set up to obtain both genotypes in the same litter. For the studies in cells, the treatment groups were randomly assigned between plates and wells. For the quantification of all image-based analyses, the investigators were blinded to allocation during experiments.

### Reporting summary
Further information on research design is available in the Nature Portfolio Reporting Summary linked to this article.

## Data availability
All data are available in the main text or the supplementary materials. Deep-sequencing (ChIP–seq and RNA-seq) data that support the findings of this study have been deposited in the Gene Expression Omnibus under the accession code GSE229551. Mass spectrometry metabolomics data are available as Supplementary Table 6. Source data are provided within this paper. All other data supporting the findings of this study are available from the corresponding author on reasonable request.

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

## Acknowledgements

We thank all the members of the Panasyuk laboratory, M. Pende and members of his laboratory, F. Terzi, J.-C. Weil, G. Asher and D. Jacobi as well as the members of INSERM-U1151 for their support and insightful discussions. We thank B. Lemesre and E. Chalouhi for their technical help with cell culture. We thank M. Pontoglio, A. Bagattin, S. Garbay and K. Shostak for technical help with the set-up of the ChIP method. We thank S. Berissi (SFR Small Animal Histology and Morphology Platform) and S. Fabrega (SFR Viral Vector and Gene Transfer Platform) for technical support. We thank L. Fajas for the gift of plasmid vectors (β-Gal, p300 and CBP) and Y. Liu for the gift of Flag–BMAL1-expressing plasmid; M. Vasseur-Cognet and A.-F. Burnol for TtrCre deleter line. This work was supported by grants from: Agence National de la Recherche (grant no. ANR-JCJC-NUTRISENSPIK-16-CE14-0029), the European Research Council (grant no. ERC-CoG-MetaboSENS-819543), an INSERM trempling grant (PINUT), Campus France PHC GERMAINE DE STAEL and Campus France PHC PROCOPE to G.P.; C.A. was supported by the Boulos Foundation and Fondation pour la Recherche Médicale; N.F.H. was supported by the French Ministry for Education and the Fondation pour la Recherche Médicale; Y.S. was supported by JSPS and the Uehara Foundation; C.D. was supported by the Swiss National Science Foundation (grant no. 310030_184708/1), Vontobel Foundation, Novartis Consumer Health Foundation, EFSD/Novo Nordisk Programme for Diabetes Research in Europe, Swiss Life Foundation, Olga Mayenfisch Foundation, Velux Foundation, Swiss Cancer League, Ligue pulmonaire Genevoise and Fondation pour l'innovation sur le cancer et la biologie; V.P. was supported by the Bo and Kerstin Hjelt Foundation for diabetes type 2, a Young Independent Investigator Grant SGED/SSED and the Gertrude von Meissner Foundation; D.L. and A.W. were supported by a German Center for Diabetes Research (DZD) Cell free DNA biomarkers grant (grant no. 82DZD07C1G) and the DAAD/PROCOPE grant (57658289); and K.H. was supported by the AFM (association Française contre les myopathies, grant no. 23101), ANR-PRC (grant no. ANR-21-CE14-0056-02-MuscLY to K.H. and G.P.) and ANR France Relance program (grant no. ANR_21_PRRD-0002-01).

## Author contributions

C.A. and N.F.H. conducted most of the experiments, analysed the data, prepared the figures and contributed to the original paper preparation, review and editing. C.A. made the original observation of the dysfunctional clock in *Vps15*-null models, demonstrated

that it is independent of the Vps34 role and showed its binding on chromatin. N.F.H. characterized the nuclear pool of Vps15 and Vps34, and performed the luciferase assays, cell transcription assays, RNA Pol2 and Bmal1 interaction studies with Vps15 and Vps34, ChIP–qPCR with Bmal1 and RNA Pol2, and ChIP–Seq/RNA-Seq data bioinformatic analyses. Y.S., A.S. and A.G. contributed to the animal experiments, performed analyses in cells and molecular analyses in liver. K.A. and M.G. contributed to the molecular analyses. A.G., E.L.G. and K.H. performed the immunofluorescence microscopy, immunohistology and PI3P analyses in cells and liver. I.N. performed the metabolomics analyses. D.L. and A.D.W. performed the bioinformatic analyses of metabolomics and gene expression data. T.T. and T.O. provided the Bmal1 constructs. X.T. provided the adenoviral vectors for Bmal1 overexpression and validated the shRNA to *Bmal1*. V.P. and C.D. performed bioluminescence analyses in vivo and in cells, and also contributed to the preparation, review and editing of the original paper. G.P. conceived and conceptualized the study, contributed to molecular analyses in cells and in vivo, analysed the data, acquired funding, supervised the work, was responsible for the project administration and wrote the paper. All authors discussed the results and commented on the paper.

## Competing interests

The authors declare no competing interests.

## Additional information

**Extended data** is available for this paper at https://doi.org/10.1038/s41556-023-01171-3.

**Correspondence and requests for materials** should be addressed to Ganna Panasyuk.

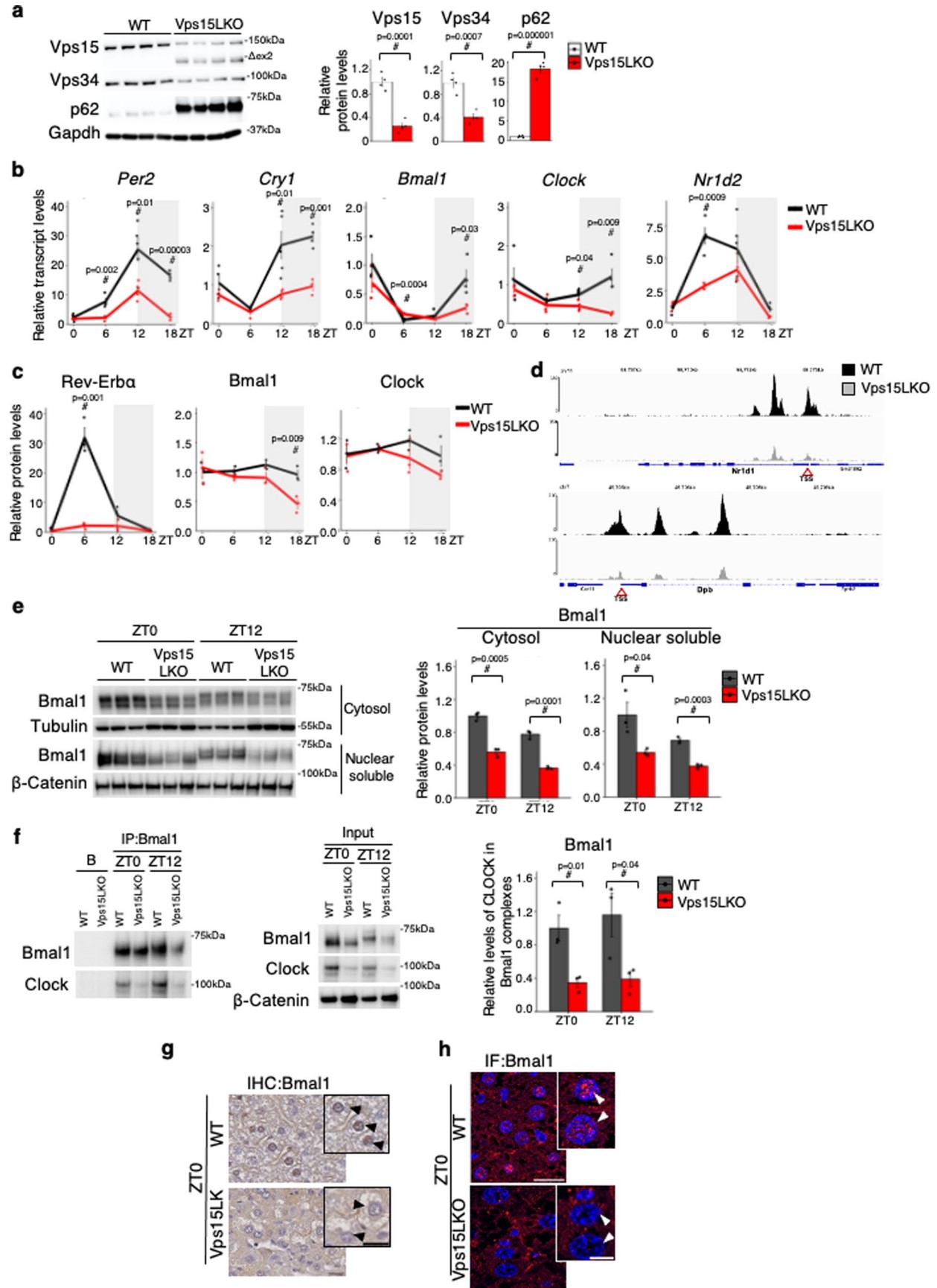

**Extended Data Fig. 1 | See next page for caption.**

**Article** https://doi.org/10.1038/s41556-023-01171-3

**Extended Data Fig. 1 | Bmal1 dysfunction in the livers of Vps15 hepatic mutants. a**, Immunoblot analysis of total protein extracts from the liver of five-week-old ad libitum-fed male Vps15LKO and control mice (ZT6). Densitometric analyses of proteins normalized to the Gapdh levels are presented. Data are the mean ± s.e.m., ($n = 4$), #$P < 0.05$, two-tailed unpaired Student's $t$-test. **b**, Relative transcript levels of clock genes in the livers of Vps15LKO and control five-week-old ad libitum-fed male mice collected at the indicated ZT times under a 12 h light–dark regimen. Data are presented as the mean ± s.e.m. ($n = 3$ Vps15LKO$_{ZT0,ZT18}$, $n = 5$ WT$_{ZT12}$ and $n = 4$ for all other groups). #$P < 0.05$ versus WT, two-tailed unpaired Student's $t$-test. Rhythmicity pattern tested using JTK_CYCLE (Supplementary Table 1). **c**, Densitometric analyses of proteins normalized to Gapdh. Data are the mean ± s.e.m. ($n = 3$). #$P < 0.05$ versus WT, two-tailed unpaired Student's $t$-test. Rhythmicity pattern tested using JTK_CYCLE (Supplementary Table 1). **d**, Bmal1 ChIP–Seq peak signals on the *Nr1d1* and *Dbp* promoters. The TSS is indicated by a triangle. **e**, Immunoblot analysis of the cytosolic and soluble nuclear fractions of WT and Vps15LKO mouse livers collected at the indicated ZT times. Densitometric analyses of Bmal1 protein normalized to tubulin (cytosolic fraction) or β-catenin (soluble nuclear fraction) levels presented as the fold change over WT$_{ZT0}$. Data are the mean ± s.e.m. ($n = 3$). #$P < 0.05$ versus WT, two-tailed, unpaired Student's $t$-test. **f**, Immunoblot analyses of Bmal1-containing complexes immunoprecipitated from the soluble nuclear fraction of the livers of WT and Vps15LKO mice. Beads alone (marked as B) served as non-specific binding control. Densitometric analysis of the relative Clock levels in Bmal1 immunoprecipitates presented. Data are the mean ± s.e.m. ($n = 3$). #$P < 0.05$ versus WT, two-tailed unpaired Student's $t$-test. **g,h**, Immunohistochemistry (**g**) and immunofluorescent microscopy (**h**) analyses of Bmal1 in liver sections of Vps15LKO and WT mice (ZT0). Analyses performed on three WT and four Vps15LKO mice, conditions and representative images are shown. Scale bars, 20 μm and 10 μm (inset). Source numerical data and unprocessed blots are provided.

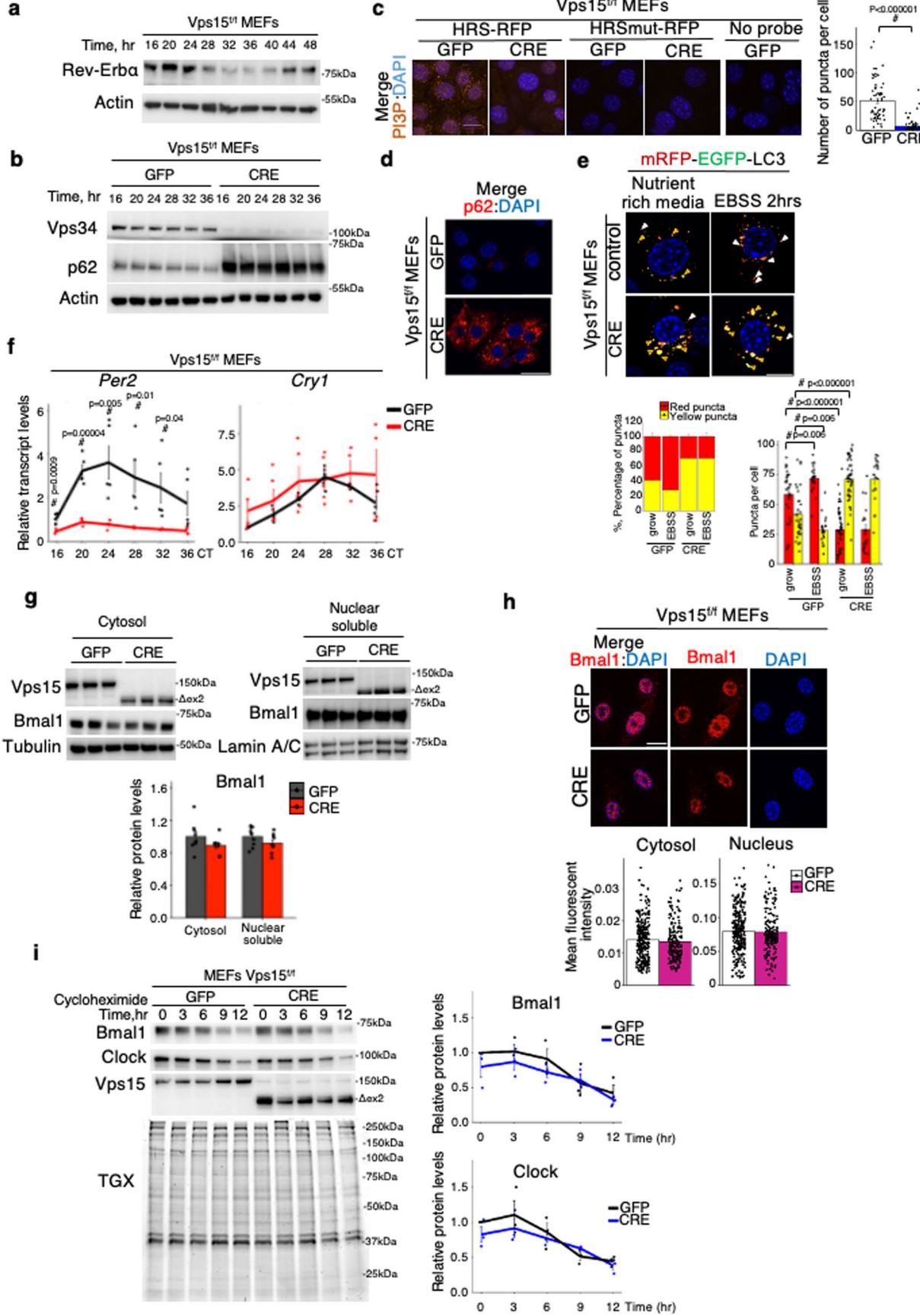

**Extended Data Fig. 2 | See next page for caption.**

**Extended Data Fig. 2 | Acute Vps15 depletion does not affect Bmal1 turnover.**
**a**,**b**, Immunoblots of total extracts of dexamethasone-synchronized control
(**a**) and Vps15-depleted MEFs (**b**) performed three times; representative repeat
shown. **c**, PI3P detection with HRS–RFP probe in Vps15-depleted (CRE) and
control (GFP) MEFs. HRSmut–RFP probe and cells treated without the probe
served as non-specific-binding controls. Quantification of HRS–RFP puncta
per cell presented the mean ± s.e.m. ($n = 66$ GFP and $n = 73$ CRE cells taken from
three experiments). $\#P < 0.000001$ versus GFP, two-tailed unpaired Student's
$t$-test. Scale bar, 10 μm. **d**, Immunofluorescence microscopy analyses of p62.
Experiment repeated three times, representative field shown. Scale bar, 20 μm.
**e**, Fluorescence microscopy of mRFP–EGFP-LC3 in MEFs that were either kept
in nutrient-rich media or EBSS for 2 h to induce autophagic flux. Red and yellow
puncta were quantified for each cell and data are presented as the percentage
of puncta (left) or number of puncta per cell (right; $n = 32$ GFP$_{grow}$, $n = 23$ GFP$_{EBSS}$,
$n = 47$ CRE$_{grow}$, $n = 20$ CRE$_{EBSS}$ cells taken from three experiments). $\#P < 0.05$ grow
versus EBSS or CRE versus GFP, two-way ANOVA with Benjamini–Hochberg
correction. Scale bar, 5 μm. **f**, Transcript levels in dexamethasone-synchronized

MEFs. Data collected in three independent experiments presented as the fold
change ± s.e.m. over GFP-treated cells ($n = 4$ CRE$_{CT32,CT36}$ and $n = 5$ other groups).
$\#P < 0.05$ versus GFP, two-tailed unpaired Student's $t$-test. Rhythmicity was
determined using JTK_CYCLE (Supplementary Table 1). **g**, Immunoblot of
cytosolic and soluble nuclear protein extracts in dexamethasone-synchronized
MEFs. Densitometric analyses of Bmal1 normalized to tubulin (cytosolic
fraction) and Lamin A/C (soluble nuclear fraction) presented as the fold change
over GFP-treated cells. Data collected in three independent experiments
presented as the mean ± s.e.m. ($n = 8$ CRE$_{Cytosol}$ and $n = 9$ for all other groups).
**h**, Immunofluorescence microscopy of Bmal1 in MEFs. Quantification of
Bmal1-positive signal in the nucleus and cytosol presented. Data are the mean
fluorescence intensity in $n = 278$ (GFP) and $n = 180$ (CRE) cells taken from three
experiments. Scale bar, 10 μm. **i**, Immunoblot of soluble nuclear fractions of
MEFs treated with cycloheximide. Densitometric analyses of Bmal1 and Clock
normalized to total protein (TGX gels) presented as the fold change ± s.e.m. over
the vehicle-treated group ($n = 3$ independent experiments). Source numerical
data and unprocessed blots are provided.

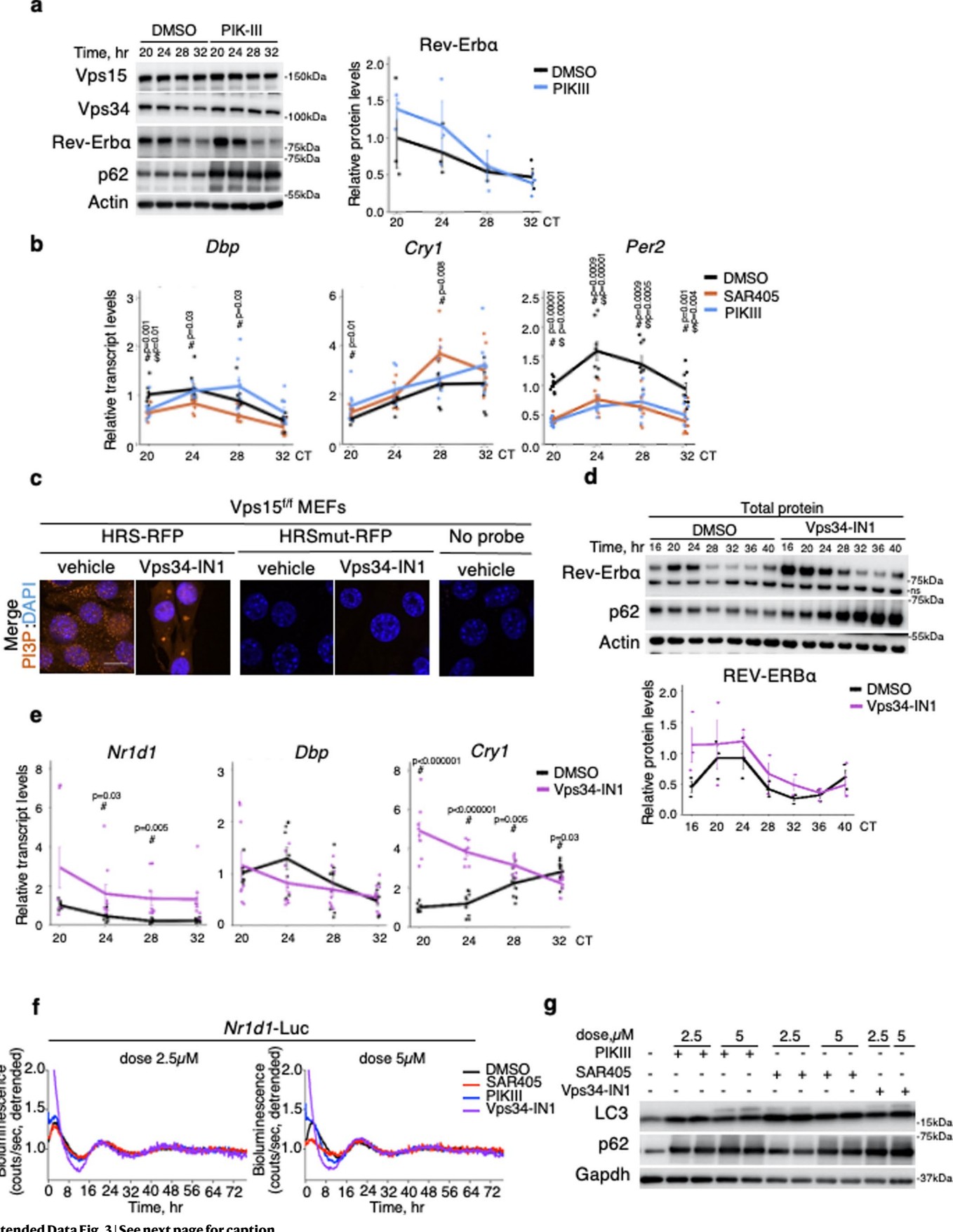

**Extended Data Fig. 3 | See next page for caption.**

**Extended Data Fig. 3 | Lipid kinase-independent role of class 3 PI3K in control of the circadian clock. a**,**b**, Protein (*n* = 3; **a**) and relative transcript levels (*n* = 8; **b**) in MEFs synchronized with dexamethasone and treated with DMSO or PIK-III. Densitometry analyses of Rev-Erbα protein normalized to the Actin levels presented as the fold change over DMSO-treated cells. Data are the mean ± s.e.m. (*n* = 3 independent experiments). #*P* < 0.05 versus DMSO (SAR405) and $*P* < 0.05 versus DMSO (PIK-III); two-way ANOVA with Benjamini−Hochberg correction. **c**, PI3P detection with HRS−RFP probe in *Vps15*^f/f^ MEFs treated with Vps34-IN1 (5 μM) for 18 h. Incubation with and without HRSmut−RFP probe served as a control for non-specific binding. Experiment repeated three times; representative fields shown. Scale bar, 10 μm. **d**,**e**, Protein (*n* = 3; **d**) and relative transcript levels (*n* = 9; **e**) in MEFs synchronized with dexamethasone and treated with DMSO

or Vps34-IN1. Densitometry analyses of Rev-Erbα protein normalized to Actin presented as the fold change over DMSO-treated cells (ns, non-specific band). Data collected in three independent experiments presented as the mean ± s.e.m. #*P* < 0.05 versus DMSO; two-tailed unpaired Student's *t*-test. **f**, Bioluminescence recordings of *Nr1d1-Luc* oscillations in MEFs synchronized with dexamethasone and treated with DMSO or Vps34 lipid kinase inhibitors (PIK-III, SAR405 and Vps34-IN1) at the indicated doses during the duration of the recording. Data are expressed as average detrended values from *n* = 4 independent experiments. **g**, Representative immunoblot analysis in MEF cells treated with DMSO, SAR405, PIK-III or Vps34-IN1 collected at the end of the recording in **f**. Source data and unprocessed blots are provided.

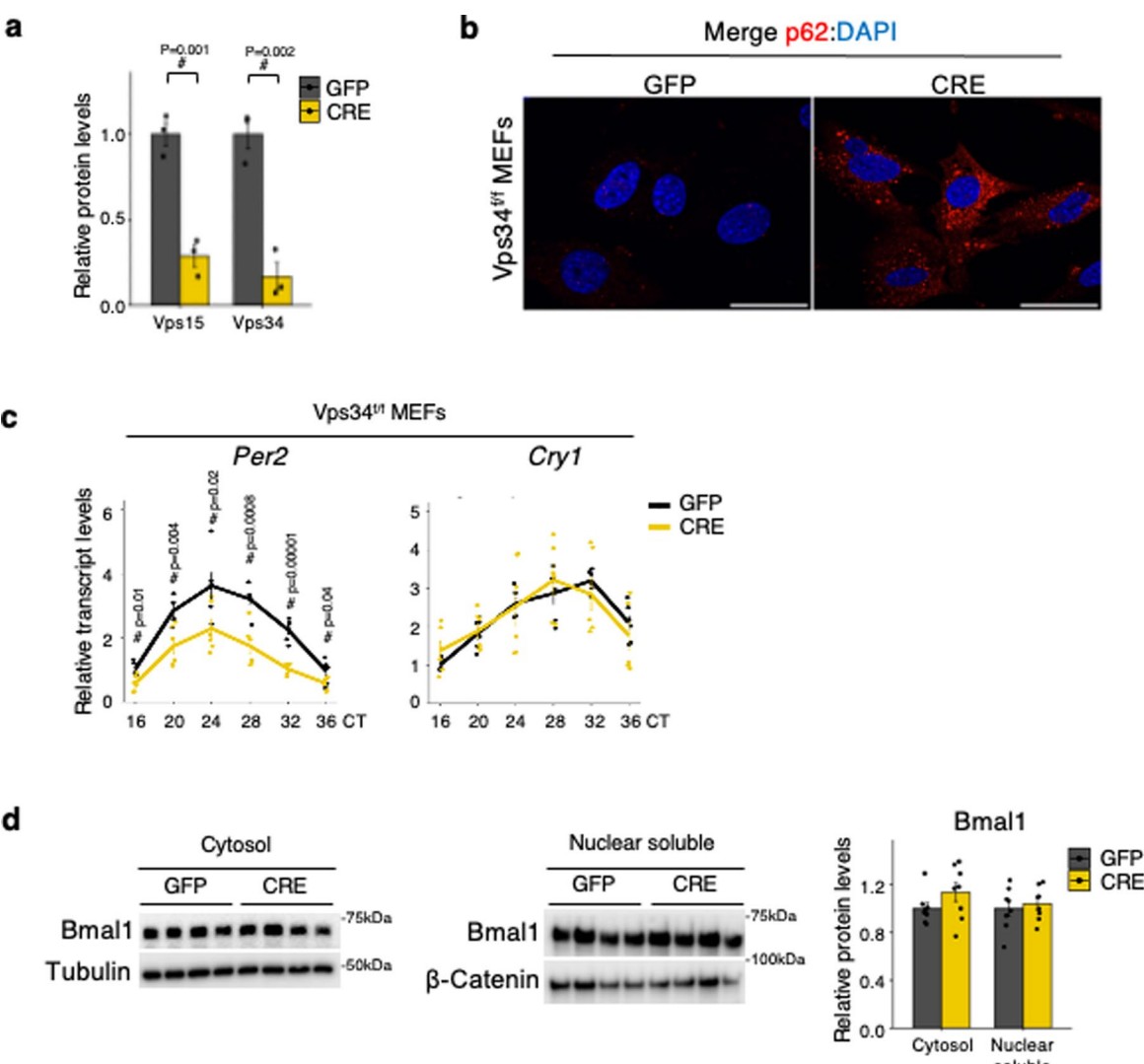

**Extended Data Fig. 4 | Vps34 inactivation does not impact Bmal1 protein levels. a**, Densitometric analysis of Vps15 and Vps34 levels normalized to Actin in total protein extracts of synchronized control (GFP) and Vps34-depleted (CRE) MEFs collected 16 h after synchronization with dexamethasone. Data are the mean ± s.e.m. ($n = 3$). #$P < 0.05$ versus GFP; two-tailed unpaired Student's $t$-test. **b**, Immunofluorescence microscopy analyses of p62 in Vps34-depleted (CRE) and GFP-transduced MEFs 5 d post infection. Experiment repeated three times, representative field shown. Scale bar, 20 µm. **c**, Relative transcript levels of the indicated genes in dexamethasone-synchronized control (GFP) and Vps34-depleted (CRE) MEFs. Data collected in three independent experiments presented as the fold change ± s.e.m. over GFP-treated cells ($n = 5$ GFP$_{CT16}$ and $n = 6$ for all other groups). #$P < 0.05$ versus GFP; two-tailed unpaired Student's $t$-test. Rhythmicity was determined using JTK_CYCLE (Supplementary Table 1). **d**, Immunoblot analysis, using the indicated antibodies, of cytosolic and soluble nuclear protein extracts in dexamethasone-synchronized control and Vps34-depleted MEFs. Densitometric analyses of Bmal1 protein normalized to tubulin (cytosolic fraction) and β-catenin (soluble nuclear fraction) presented as the fold change over GFP-treated cells. Data collected in three independent experiments are the mean ± s.e.m. ($n = 8$). Source data and unprocessed blots are provided.

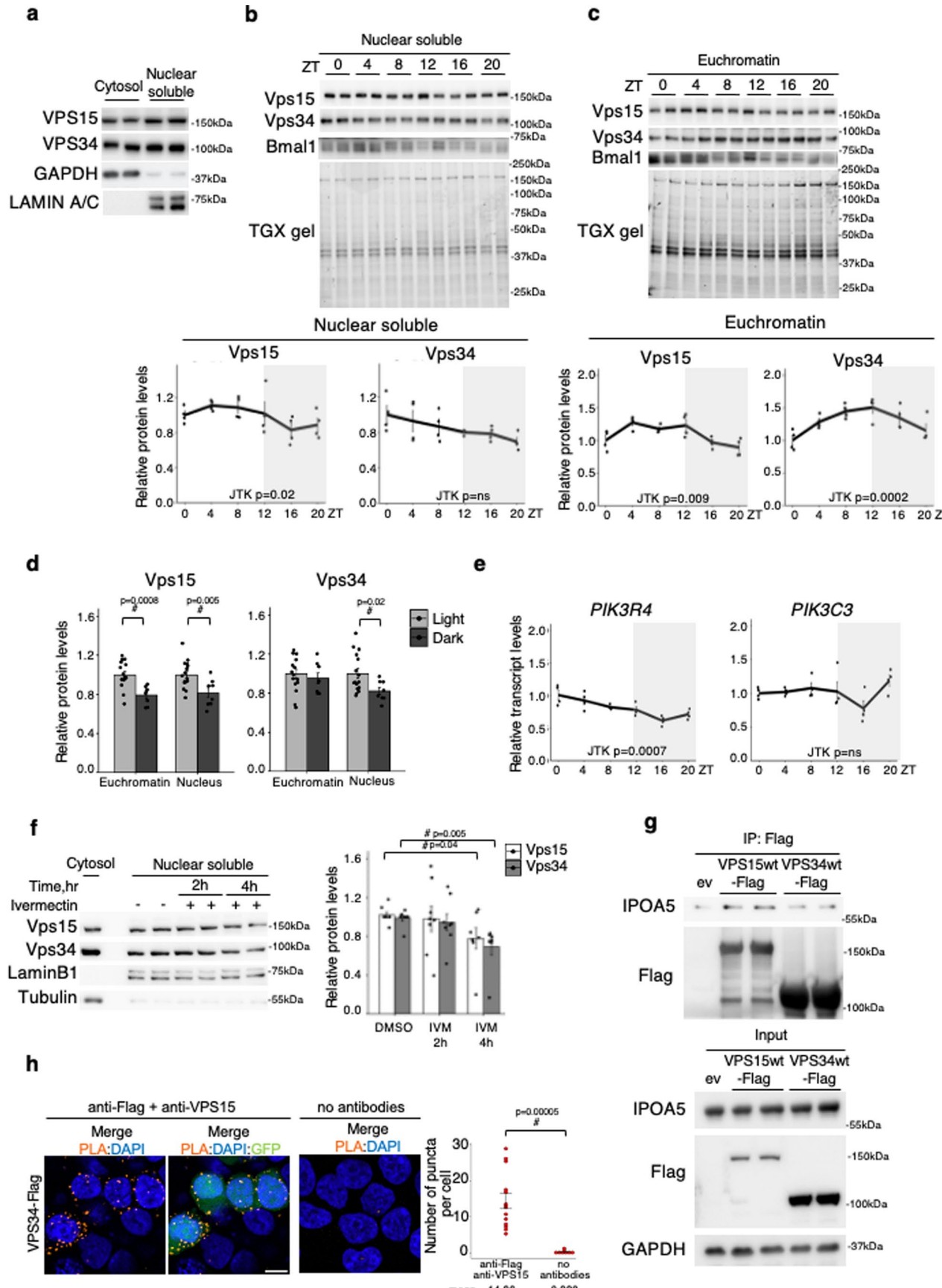

**Extended Data Fig. 5 | See next page for caption.**

**Extended Data Fig. 5 | Rhythmic nuclear expression of class 3 PI3K subunits.**
**a**, Immunoblot analysis, using the indicated antibodies, of the soluble nuclear and cytosolic fractions of HEK293T cells. GAPDH and LAMIN A/C served as purity controls for the cytosolic and soluble nuclear fractions. Experiment repeated five times; representative repeat shown. **b**,**c**, Immunoblot analysis of soluble nuclear (**b**) and euchromatin (**c**) fractions from the livers of eight-week-old male WT mice. Densitometric analysis of Vps15 and Vps34 normalized to total protein (TGX gels) are presented as the fold change over ZT0. Data are the mean ± s.e.m. ($n$ = 4 mice). Rhythmicity pattern was tested using JTK_CYCLE (Supplementary Table 1). **d**, Mean normalized Vps15 and Vps34 protein levels in soluble nuclear (as in **b**) and euchromatin (as in **c**) fractions during the light (ZT0–ZT8) and dark (ZT12–ZT20) phase. Data are the mean ± s.e.m. ($n$ = 8 mice in dark and $n$ = 16 mice in light). #$P$ < 0.05 versus light; two-tailed unpaired Student's $t$-test. **e**, Relative transcript levels of the indicated genes in the livers of mice treated as in **b**. Data are presented as the mean ± s.e.m. ($n$ = 4 mice). Rhythmicity was determined using JTK_CYCLE (Supplementary Table 1). **f**, Immunoblot analysis, using the indicated antibodies, of the soluble nuclear protein extracts of MEF cells treated with ivermectin for the indicated times. Densitometric analyses of Vps15 and Vps34 normalized to lamin B1 presented as the fold change over vehicle-treated cells. Data are the mean ± s.e.m. (independent repeats: $n$ = 7 ivermectin (4 h) and $n$ = 8 for all other groups). #$P$ < 0.05 versus DMSO; two-tailed unpaired Student's $t$-test. **g**, Immunoblot analyses of IPOA5 co-immunoprecipitated with ectopically expressed VPS15 and VPS34 from HEK293T cells using anti-Flag (ev, empty vector). Immunoprecipitation was performed three times; representative blots shown. **h**, Proximity ligation assay between endogenous VPS15 and ectopic Flag–VPS34 protein in HEK293T cells (co-transfection with GFP-expressing vector visualized transfected cells). The 'no antibody' condition served as a control for the non-specific signal. Scale bar, 10 μm. Data are the mean ± s.e.m. of proximity puncta per cell ($n$ = 8 fields (no antibody) and $n$ = 14 fields (Flag + VPS15) with over 300 cells collected in three independent experiments for each condition is presented; no antibody condition shared with Fig. 4e). #$P$ = 0.00005 versus no antibody; two-tailed unpaired Student's $t$-test. Source numerical data and unprocessed blots are provided.

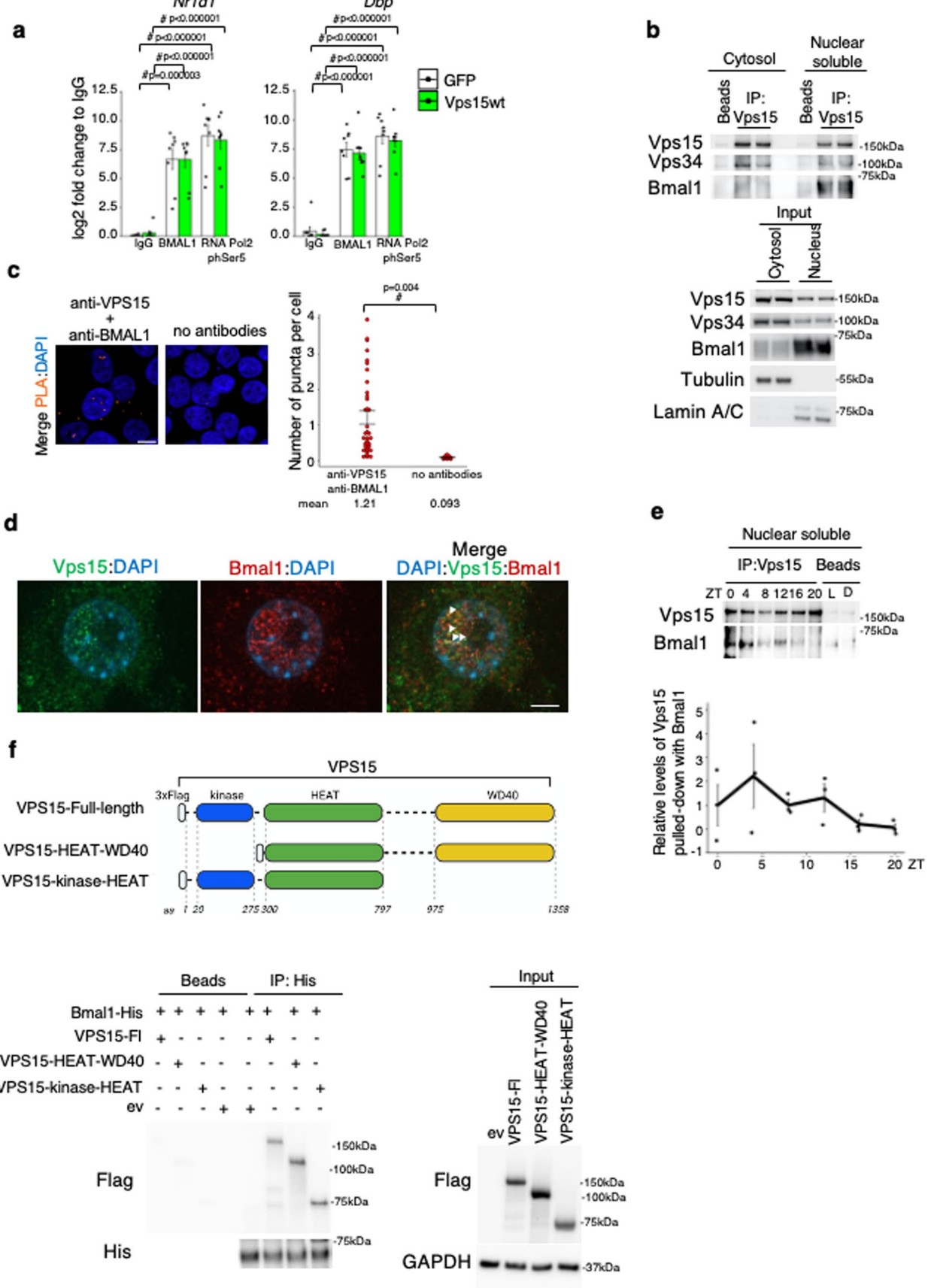

**Extended Data Fig. 6 | See next page for caption.**

**Extended Data Fig. 6 | Vps15 interacts with Bmal1. a**, Bmal1 and RNA Pol2 pS5 recruitment to the promoters of the indicated genes in MEFs transduced with adenoviral vectors expressing VPS15WT or GFP as a control (determined by ChIP–qPCR). The cells were dexamethasone-synchronized and collected 24 h after synchronization. Data are presented as the $\log_2$-transformed fold change enrichment over IgG ± s.e.m. ($n = 8$ independent experiments). #$P < 0.05$ versus IgG; two-tailed unpaired Student's $t$-test. **b**, Immunoblot analyses of Vps15-containing immunoprecipitates from the soluble nuclear and cytosol fractions of MEF cells. Immunoprecipitation was performed three times; representative blots shown. **c**, Proximity ligation assay between endogenous VPS15 and BMAL1 proteins in HEK293T cells. The no antibody condition served as a control for non-specific binding. Scale bar, 10 μm. Data are the mean ± s.e.m. proximity puncta per cell ($n = 9$ fields (no antibody) or $n = 36$ fields (VPS15 + BMAL1) with over 1,000 cells collected in three independent experiments for each condition). #$P = 0.004$ versus no antibody; two-tailed unpaired Student's $t$-test.

**d**, Immunofluorescence microscopy analyses of endogenous Vps15 and Bmal1 in MEF cells. The white triangles point to the co-localization. Experiment repeated three times, representative field shown. Scale bar, 5 μm. **e**, Immunoblot analyses, using anti-Vps15, of Vps15-containing immunoprecipitates from the soluble nuclear fraction of WT livers of five-week-old ad libitum-fed male mice collected at the indicated ZT times. Quantification of Bmal1 in Vps15 immunoprecipitates normalized to the non-specific binding to beads is presented as the fold change over ZT0. Data are the mean ± s.e.m. ($n = 3$ independent experiments on different mice). **f**, Schematic representation of Flag-tagged VPS15-deletion mutants. Immunoblot analyses, using anti-Flag, of Bmal1-His-containing immunoprecipitates from total protein extracts of HEK293T cells expressing deletion mutants of VPS15 (Flag tagged) together with Bmal1-His full-length (Fl) protein. Experiment performed three times and representative blot shown. Source numerical data and unprocessed blots are provided.

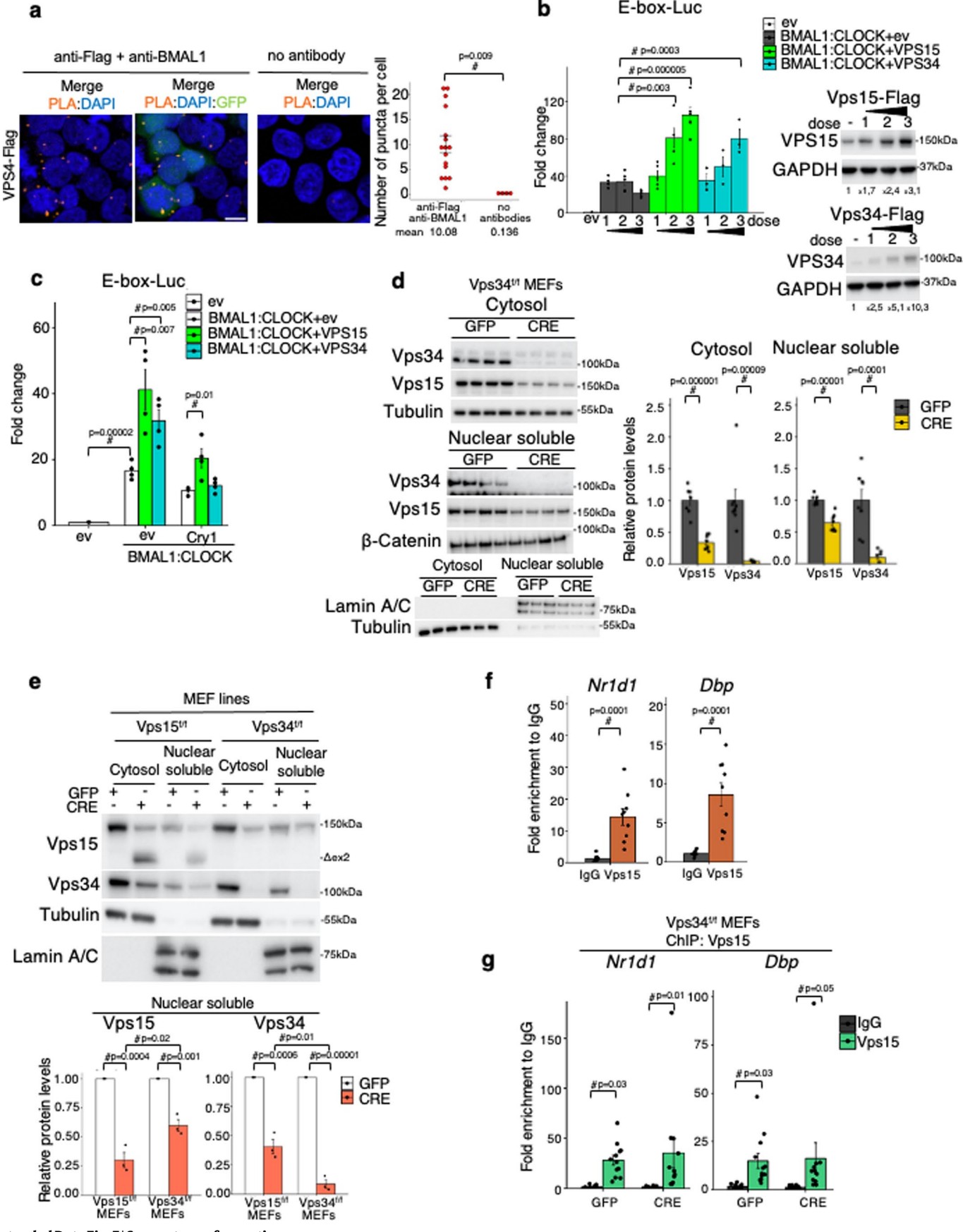

**Extended Data Fig. 7 | See next page for caption.**

**Extended Data Fig. 7 | Vps34-independent function of Vps15 in Bmal1−Clock coactivation. a**, Proximity ligation assay of BMAL1 and Flag–VPS34 in HEK293T cells (GFP co-transfection shows transfected cells). No antibody served as a non-specific control. Scale bar, 10 μm. Data are the mean ± s.e.m. proximity puncta per cell ($n = 4$ fields (no antibody) and $n = 16$ fields (Flag+BMAL1) with over 290 cells collected in three independent experiments). #$P = 0.009$ versus no antibody; two-tailed unpaired Student's $t$-test. **b**, Luciferase assay in HEK293T cells co-transfected with E-box-Luc reporter, empty vector or BMAL1−CLOCK, with or without increasing doses of VPS15WT or VPS34WT. Relative luminescence presented as the fold difference over E-box-Luc. Data are the mean ± s.e.m. (independent experiments, $n = 3$ BMAL1−CLOCK + VPS34 and $n = 5$ for all other groups). #$P < 0.05$; two-tailed unpaired Student's $t$-test. Representative immunoblot of VPS15 and VPS34 shows overexpression. **c**, Luciferase assay in HEK293T cells co-transfected with E-box-Luc reporter, empty vector or BMAL1−CLOCK, with or without Cry1 or VPS34 (control and VPS15WT conditions shared with Fig. 5b). Relative luminescence presented as the fold difference over the E-box-Luc only condition. Data are the mean ± s.e.m. ($n = 4$ independent experiments). #$P < 0.05$; two-tailed unpaired Student's $t$-test. **d**, Immunoblot of the cytosolic and soluble nuclear extracts of dexamethasone-synchronized MEFs. Densitometric analyses of proteins normalized to tubulin (cytosol) and β-catenin (nucleus) presented as the fold change over the GFP condition. Data are the mean ± s.e.m. ($n = 8$ independent repeats). #$P < 0.05$ versus GFP; two-tailed unpaired Student's $t$-test. Bottom panel shows the fraction purity. **e**, Immunoblot of the cytosolic and nuclear extracts of dexamethasone-synchronized MEFs. Densitometric analyses of proteins normalized to tubulin (cytosol) and lamin A/C (nucleus) presented as the fold change over the GFP condition. Data are the mean ± s.e.m. ($n = 3$ independent experiments). #$P < 0.05$; two-tailed unpaired Student's $t$-test. **f**, ChIP–qPCR of Vps15 in dexamethasone-synchronized MEFs 24 h post synchronization. Data collected in three independent experiments presented as the mean ± s.e.m. fold enrichment over IgG ($n = 9$). #$P < 0.05$ versus IgG; two-tailed unpaired Student's $t$-test. **g**, ChIP–qPCR of Vps15 in dexamethasone-synchronized control and Vps34-depleted MEFs 24 h post synchronization. Data collected in three independent experiments presented as the mean ± s.e.m. fold enrichment over IgG ($n = 10$ ChIP-IgG-CRE, $n = 11$ ChIP-Vps15-CRE and $n = 12$ for all other groups). #$P < 0.05$ versus IgG, two-way ANOVA with Benjamini–Hochberg correction. Source numerical data and unprocessed blots are provided.

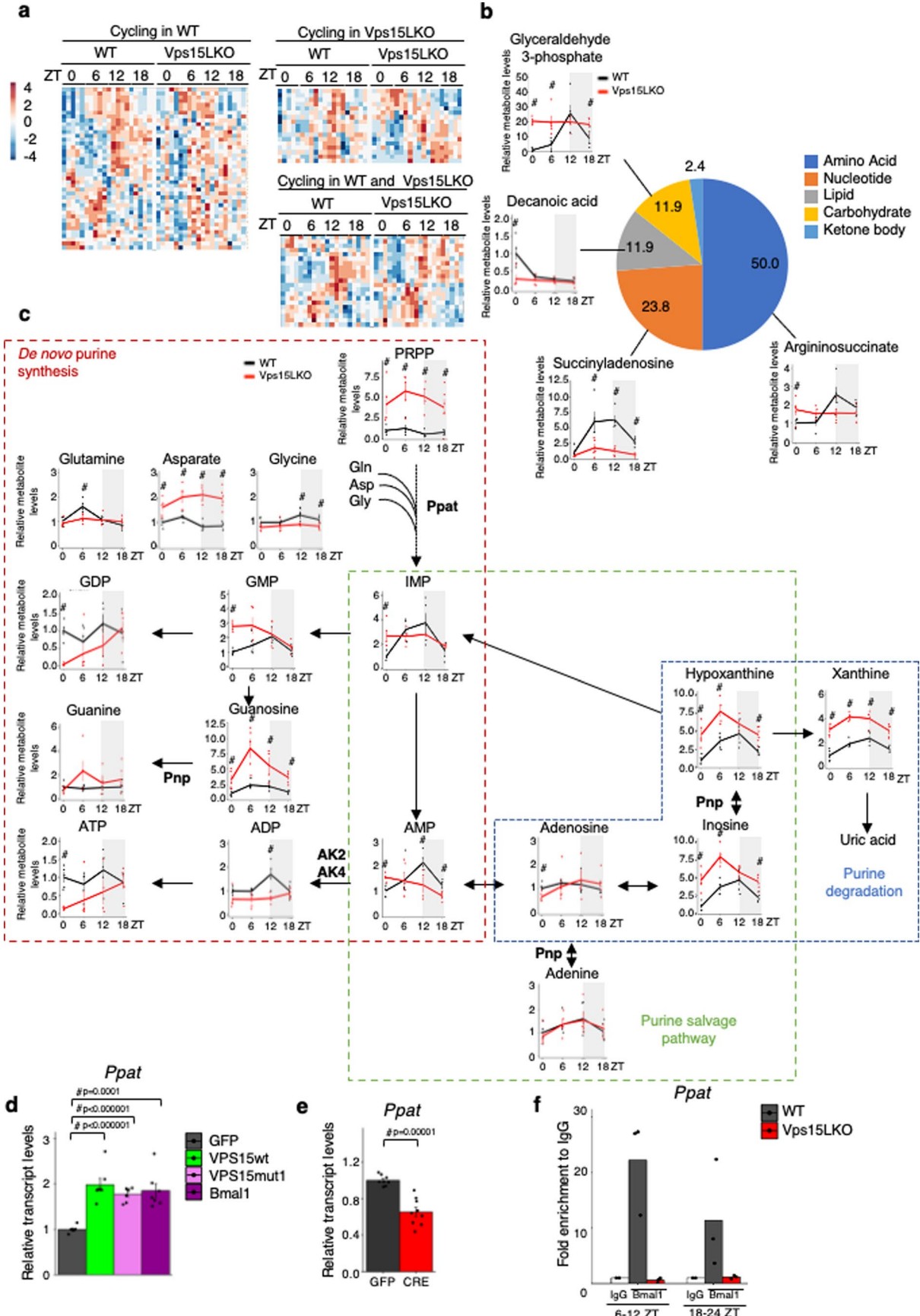

**Extended Data Fig. 8 | See next page for caption.**

**Extended Data Fig. 8 | De novo purine synthesis in liver depends on Vps15. a**, Heatmap showing cycling metabolites in the livers of five-week-old ad libitum-fed Vps15LKO and control male mice. Rhythmicity pattern determined with JTK_CYCLE (Supplementary Table 1). Columns correspond to different mice and cell colour shows the relative metabolite level. **b**, Pie chart of chemical classification of metabolites arrhythmic in Vps15LKO; percentage of each class of metabolites cycling in WT liver presented. Metabolite examples shown as graphs. Data are the mean ± s.e.m. ($n = 5$). #$P < 0.05$ versus WT (glyceraldehyde 3-phosphate: $ZT_0 P < 0.000001$, $ZT_6 P = 0.008$, $ZT_{18} = 0.01$; decanoic acid: $ZT_0 P = 0.007$; succinyladenosine: $ZT_6 P = 0.03$, $ZT_{12} P = 0.0001$, $ZT_{18} = 0.0001$; arginossuccinate: $ZT_0 P = 0.04$); two-tailed unpaired Student's $t$-test. **c**, Schematic representation of the purine metabolism. Data are the mean ± s.e.m. ($n = 5$). #$P < 0.05$ versus WT (PRPP: $ZT_0 P = 0.02$, $ZT_6 P = 0.002$, $ZT_{12} P = 0.0003$, $ZT_{18} P = 0.03$; glutamine: $ZT_6 P = 0.007$; aspartate: $ZT_0 P = 0.009$, $ZT_6 P = 0.009$, $ZT_{12} P = 0.0001$, $ZT_{18} P = 0.0006$; glycine: $ZT_{12} P = 0.02$, $ZT_{18} P = 0.01$; IMP: $ZT_0 P = 0.008$; GMP: $ZT_0 P < 0.000001$; GDP: $ZT_0 P < 0.000001$; guanosine: $ZT_0$

$P = 0.01$, $ZT_6 P = 0.004$, $ZT_{12} P = 0.04$, $ZT_{18} P = 0.0002$; AMP: $ZT_0 P = 0.002$, $ZT_{12} P = 0.01$, $ZT_{18} P = 0.007$; ADP: $ZT_{12} P = 0.006$; ATP: $ZT_0 P = 0.0001$; adenosine: $ZT_0 P = 0.04$; inosine: $ZT_0 P = 0.0007$, $ZT_6 P = 0.0004$, $ZT_{18} P = 0.002$; hypoxanthine: $ZT_0 P = 0.001$, $ZT_6 P = 0.001$, $ZT_{18} P = 0.001$; xanthine: $ZT_0 P < 0.000001$, $ZT_6 P < 0.000001$, $ZT_{12} P = 0.02$, $ZT_{18} P = 0.0001$); two-tailed unpaired Student's $t$-test. **d**, *Ppat* transcript levels in primary hepatocytes transduced with the indicated adenoviral vectors. Data are the mean ± s.e.m. ($n = 7$ independent experiments). #$P < 0.05$ versus GFP; two-tailed unpaired Student's $t$-test. **e**, *Ppat* transcript levels in *Vps15*$^{f/f}$ primary hepatocytes transduced with GFP or CRE adenoviral vectors. Data are the mean ± s.e.m. (independent experiments, $n = 8$ GFP and $n = 9$ CRE). #$P = 0.00001$ versus GFP; two-tailed unpaired Student's $t$-test. **f**, Bmal1 recruitment, determined by ChIP–qPCR, to the promoter of *Ppat* in pooled liver tissue samples collected at $ZT_{6-12}$ and $ZT_{18-24}$ of five-week-old ad libitum-fed Vps15LKO and control male mice. Data are the mean fold enrichment over IgG (independent experiments, $n = 2$ IgG and $n = 3$ ChIP-Bmal1). Source data are provided.

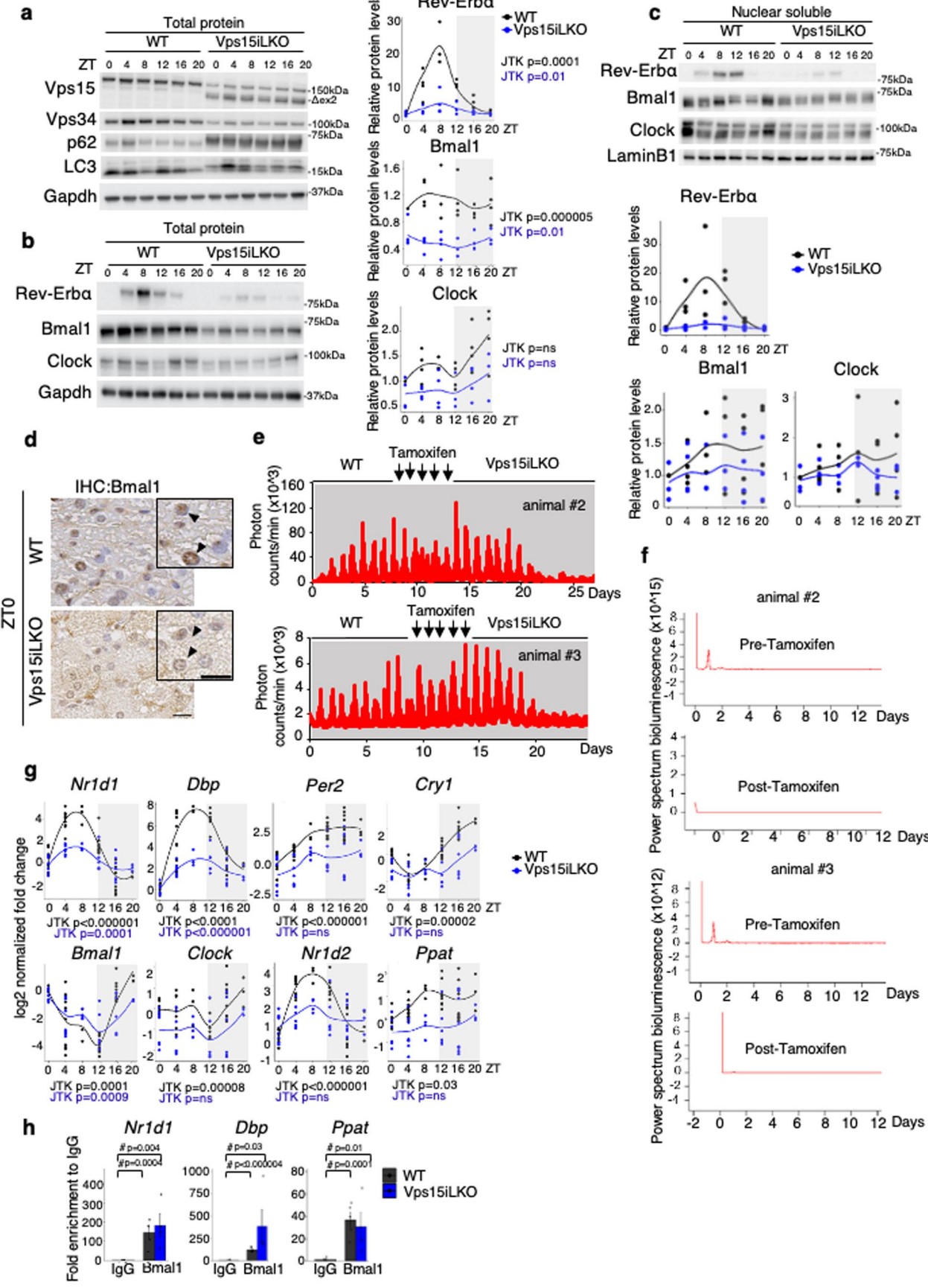

**Extended Data Fig. 9 | See next page for caption.**

**Extended Data Fig. 9 | Bmal1 transcriptional activity is inhibited following acute depletion of Vps15 in liver. a,b,** Immunoblot analysis, using the indicated antibodies, of total protein extracts from the liver of 3–4-month-old male and female ad libitum-fed Vps15iLKO and control mice. The samples were collected every 4 h under a 12 h light–dark regimen. Densitometric analyses of protein levels normalized to Gapdh presented as a Loess best-fit curve ($n$ = 3 WT$_{ZT8}$ and Vps15iLKO$_{ZT12,ZT20}$ and $n$ = 4 for all other groups). Rhythmicity was determined using JTK_CYCLE (Supplementary Table 1). **c,** Immunoblot analysis, using the specified antibodies, of the soluble nuclear fractions of the livers of WT and Vps15iLKO male and female mice collected at the indicated time points. Densitometric analyses of proteins normalized to Lamin B1 (soluble nuclear fraction) presented as a Loess best-fit curve ($n$ = 3 WT$_{ZT0,ZT8}$ and Vps15iLKO$_{ZT0,ZT8,ZT12,ZT20}$, and $n$ = 4 for all other groups). **d,** Immunohistochemistry analyses of Bmal1 in liver sections of Vps15iLKO and control male mice (ZT0) treated as in **a.** Scale bars, 20 μm and 10 μm (inset). Analyses performed on three mice; representative images are shown. **e,** Recordings of circadian *Nr1d1-Luc* expression in the livers of control (pre-tamoxifen) and Vps15iLKO

(post-tamoxifen) male mice receiving luciferin in their drinking water (animals 2 and 3). The mice were injected with Ad-*Nr1d1-Luc* vectors via their tail vein. Bioluminescence started 48 h after injection and was monitored for 9 d (WT) in constant darkness. Tamoxifen was injected for five consecutive days, after which the Vps15iLKO phase was recorded. Simultaneously monitored spontaneous locomotor activity profiles are shown (bottom). **f,** Periodograms (FFT analysis) of the data shown in **e. g,** Relative transcript levels of the indicated genes in the livers of mice treated as in **a.** Data are log$_2$-transformed normalized fold change as a Loess best-fit curve ($n$ = 3 WT$_{ZT8}$, Vps15iLKO$_{ZT20}$; $n$ = 4 Vps15iLKO$_{ZT8}$, WT$_{ZT20}$; $n$ = 6 Vps15iLKO$_{ZT12}$; $n$ = 7 Vps15iLKO$_{ZT4}$; $n$ = 8 WT$_{ZT0}$, Vps15iLKO$_{ZT0}$, Vps15iLKO$_{ZT16}$; $n$ = 9 WT$_{ZT12}$, WT$_{ZT16}$; and $n$ = 10 for all other groups). Rhythmicity was determined using JTK_CYCLE (Supplementary Table 1). **h,** Bmal1 recruitment to the promoter of the indicated genes (determined by ChIP–qPCR) in pooled liver tissue (collected at ZT6) of male mice treated as in **a.** Data are the mean ± s.e.m. fold enrichment over IgG ($n$ = 4 Vps15iLKO, $n$ = 5 W and $n$ = 6 IgG). #$P$ < 0.05 versus IgG; two-tailed unpaired Student's $t$-test. Source data and unprocessed blots are provided.

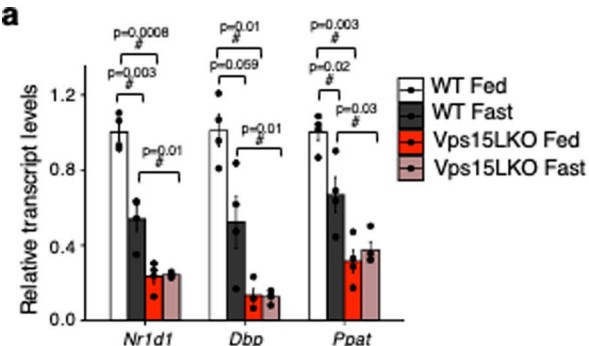

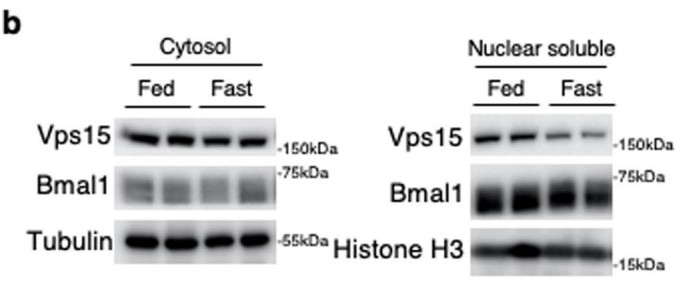

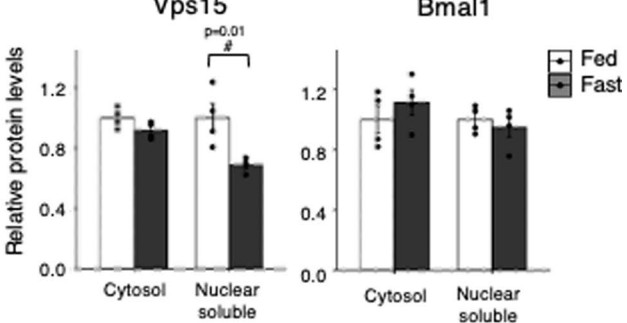

**Extended Data Fig. 10 | Dampened response to fasting in hepatic Vps15 mutants. a**, Relative transcript levels of the indicated genes in liver tissue collected at ZT6 from WT and Vps15LKO male mice that were fed ad libitum or fasted for 24 h. Data are the mean ± s.e.m. fold change over the ad libitum-fed WT (*n* = 4). #*P* < 0.05 versus WT or Fed; two-way ANOVA with Benjamini–Hochberg correction. **b**, Immunoblot analysis, using the indicated antibodies, of the cytosolic and soluble nuclear fractions of the liver tissue of WT mice treated as in **a**. Densitometric analyses of Vps15 or Bmal1 proteins normalized to Tubulin (cytosol) or Histone H3 (nucleus) levels presented as the fold change ± s.e.m. over WT fed (*n* = 4). #*P* = 0.01 versus Fast; two-tailed unpaired Student's *t*-test. Source data and unprocessed blots are provided.

# Reporting Summary

## Statistics

For all statistical analyses, confirm that the following items are present in the figure legend, table legend, main text, or Methods section.

| n/a | Confirmed | |
|---|---|---|
| ☐ | ☒ | The exact sample size (*n*) for each experimental group/condition, given as a discrete number and unit of measurement |
| ☐ | ☒ | A statement on whether measurements were taken from distinct samples or whether the same sample was measured repeatedly |
| ☐ | ☒ | The statistical test(s) used AND whether they are one- or two-sided *Only common tests should be described solely by name; describe more complex techniques in the Methods section.* |
| ☒ | ☐ | A description of all covariates tested |
| ☐ | ☒ | A description of any assumptions or corrections, such as tests of normality and adjustment for multiple comparisons |
| ☐ | ☒ | A full description of the statistical parameters including central tendency (e.g. means) or other basic estimates (e.g. regression coefficient) AND variation (e.g. standard deviation) or associated estimates of uncertainty (e.g. confidence intervals) |
| ☐ | ☒ | For null hypothesis testing, the test statistic (e.g. *F*, *t*, *r*) with confidence intervals, effect sizes, degrees of freedom and *P* value noted *Give P values as exact values whenever suitable.* |
| ☒ | ☐ | For Bayesian analysis, information on the choice of priors and Markov chain Monte Carlo settings |
| ☒ | ☐ | For hierarchical and complex designs, identification of the appropriate level for tests and full reporting of outcomes |
| ☒ | ☐ | Estimates of effect sizes (e.g. Cohen's *d*, Pearson's *r*), indicating how they were calculated |

*Our web collection on statistics for biologists contains articles on many of the points above.*

## Software and code

Policy information about availability of computer code

| Data collection | Softwares for data collection are listed in Methods, including: ChemiDocTM Imager (BioRad) for Western Blots revelation; Zeiss Zen Blue 3.7 Microscopy Software was used for image processing following microscopy. For in vivo bioluminescence measurements, custom made lumicycler was used (Dr. Charna Dibner Laboratory as described in PMID: 33602874) and bioluminescence pattern was monitored by a LumiCycle 96 (Actimetrics). |
|---|---|
| Data analysis | The software packages used in this study are listed in the Methods, including: Western blot quantifications were done using ImageJ software (v. 2.1.0) and calculations in Microsoft Excel 365 (v. 2016) and GraphPad Prizm9. q-PCR data analyses were done using QuantStudio 1 (ThermoFisher Scientific) and calculations in Microsoft Excel 365 (v. 2016) and GraphPad Prizm9. Chip-seq and RNA-seq data were analyzed with SOAPnuke, Bowtie2 v2.4.4, SAMtools v 1.13, deepTools2, MACS2 v2.2.7.1, and R v4.1.2 (http://www.r-project.org/foundation/). Metabolite classes and pathways were identified using R package "MetaboAnalystR3.0". Rhythmicity was assessed with non-parametric "MetaCycle" (JTK_CYCLE) algorithm implemented in R. For data presentation, all data plots made with ggplot2 using R-studio. IGV_2.8.13 used for bigwig/bam file ChIP-seq visualization. |

For manuscripts utilizing custom algorithms or software that are central to the research but not yet described in published literature, software must be made available to editors and reviewers. We strongly encourage code deposition in a community repository (e.g. GitHub). See the Nature Portfolio guidelines for submitting code & software for further information.

## Data

Policy information about <u>availability of data</u>

All manuscripts must include a <u>data availability statement</u>. This statement should provide the following information, where applicable:

- Accession codes, unique identifiers, or web links for publicly available datasets
- A description of any restrictions on data availability
- For clinical datasets or third party data, please ensure that the statement adheres to our <u>policy</u>

All data are available in the main text or the supplementary materials. Deep-sequencing (ChIP-seq and RNAseq) data that support the findings of this study have been deposited in the Gene Expression Omnibus (GEO) under accession code GSE229551. Mass spectrometry metabolomics data are available as Supplementary data 6. Source data have been provided in Source Data file. All other data supporting the findings of this study are available from the corresponding author on reasonable request.

## Human research participants

Policy information about <u>studies involving human research participants and Sex and Gender in Research.</u>

| | |
|---|---|
| Reporting on sex and gender | NA |
| Population characteristics | NA |
| Recruitment | NA |
| Ethics oversight | NA |

Note that full information on the approval of the study protocol must also be provided in the manuscript.

# Field-specific reporting

Please select the one below that is the best fit for your research. If you are not sure, read the appropriate sections before making your selection.

☒ Life sciences  ☐ Behavioural & social sciences  ☐ Ecological, evolutionary & environmental sciences

For a reference copy of the document with all sections, see <u>nature.com/documents/nr-reporting-summary-flat.pdf</u>

# Life sciences study design

All studies must disclose on these points even when the disclosure is negative.

| | |
|---|---|
| Sample size | For the studies in cell lines we determined the number of the experimental repetitions based on our previous studies using these lines (pubmid: 26387534; 23630012). All experiments were independently three times unless specified in the legends, and mean and the standard error from the mean were calculated.<br>In studies with animals, the "n" number corresponds to an individual mouse. Power analysis was used to determine the animal numbers to take into account previously observed magnitude of response to deletion of gene of interest in vivo (e.g. metabolic parameters, autophagy block for Vps15 deletion) and observations of circadian behavior for C57/B6 strain of mice previously reported (expression and metabolomics analyses).  We applied >80% power and error rate 5% (two sided type 1) to detect >1.5 effect. |
| Data exclusions | No data or animals were excluded from the analyses. |
| Replication | All in vitro and in vivo assays have been performed  3 times unless specified in the legends. The accompanying quantification and statistics were derived from n=3 independent replicates unless specified in the legends. For biological replicates, independent preparation of depleted cells (adenoviral infection) were used to ensure reproducibility. For in vivo experiments, 3 mice per condition were analyzed. The numbers of replicates and mice used for each analysis is indicated in figure legends. |
| Randomization | For studies in cells, the treatment groups were attributed randomly between plates and wells. For studies in vivo, animals, including those treated with tamoxifen for gene deletion, were randomly allocated into experimental groups based on genotype. The breeding was set up in order of obtain both genotypes in the same litter. |
| Blinding | The blinding was applied for quantifications of all image-based analyses. In all treatments in vitro and in vivo, the samples were number-coded and the investigators were blinded during extract preparations for molecular analyses (e.g. transcript and protein expression, luciferase assays). Depending on the analysis, the groups were revealed either at the stage of sample quantifications and statistical analyses (e.g. qPCR, luciferase assay) or when it was essential for analysis (e.g. immunoblot of time series to include all time points/genotype/treatments in the same gel panel, immunoprecipitation or ChIP to set-up control and experimental groups with antibodies). |

# Reporting for specific materials, systems and methods

We require information from authors about some types of materials, experimental systems and methods used in many studies. Here, indicate whether each material, system or method listed is relevant to your study. If you are not sure if a list item applies to your research, read the appropriate section before selecting a response.

## Materials & experimental systems

| n/a | Involved in the study |
|-----|----------------------|
| ☐ | ☒ Antibodies |
| ☐ | ☒ Eukaryotic cell lines |
| ☒ | ☐ Palaeontology and archaeology |
| ☐ | ☒ Animals and other organisms |
| ☒ | ☐ Clinical data |
| ☒ | ☐ Dual use research of concern |

## Methods

| n/a | Involved in the study |
|-----|----------------------|
| ☐ | ☒ ChIP-seq |
| ☒ | ☐ Flow cytometry |
| ☒ | ☐ MRI-based neuroimaging |

## Antibodies

| Antibodies used | All antibodies are listed in Methods section: Vps15 (1:1000, Abnova, H00030849-M02; For Proximity ligation assay 1:200, Genetex, GTX108953), p62 (1:1000, Abnova, H00008878-MO1), β-actin (1:5000, Sigma, A5316), Tubulin (1:1000, Sigma, T9026), β-catenin (1:500, BD Biosciences, 610153), Lamin A/C (1:1000, Cell Signaling, 2032), LC3 (1:1000, NanoTools, 0231-100/LC3-3-5-5F10), GAPDH (1:1000, Santa Cruz, SC-25778), REV-ERBα (1:1000, Cell Signaling, 13418S), BMAL1 (for IHC 1:250 and for Chip 4µg/IP, Abcam, ab3350; for WB 1:1000 and IF 1:200, Cell Signaling, 14020S), CLOCK (1:1000, Cell Signaling, 5157), HIS-Tag (1:1000, Proteintech, 66005-12-Ig), IPOA5 (1:1000, Proteintech, 18137-1-AP), Cry1 (1:500, Origene, TA342728), RNA Pol II total (1:1000 Active Motif, 39097), RNA Pol II phospho-Serine5 (1:1000, Chromotek, 3E8-1), RNA Pol II CTD phospho-Serine5 (1:1000, Abcam, ab5408), Vps34 (1:1000, Cell Signaling, 4263), Flag (1:1000, Sigma, F1804), Hsp90 (1:1000, Proteintech, 13171-1-AP), Histone H3 (1:3000, Cell Signaling 4499), normal-IgG Rabbit (Cell Signaling, 3900), normal-IgG Mouse (Santa Cruz, sc-2025), anti-Rabbit IgG HRP-linked Antibody (1:5000, Cell Signaling 7074), Anti-Mouse IgG HRP-linked Antibody (1:5000, Cell Signaling 7076), Donkey anti-Rabbit IgG (H+L) Highly Cross-Absorbed Secondary Antibody, Alexa Fluor Plus 488 (1:200, ThermoFisher, A32790), Donkey anti-Mouse IgG (H+L) Highly Cross-Absorbed Secondary Antibody, Alexa Fluor Plus 488 (1:200, ThermoFisher, A32766), Donkey anti-Rabbit IgG (H+L) Highly Cross-Absorbed Secondary Antibody, Alexa Fluor Plus 568 (1:200, ThermoFisher, A10042), Donkey anti-Mouse IgG (H+L) Highly Cross-Absorbed Secondary Antibody, Alexa Fluor Plus 568 (1:200, ThermoFisher, A10037), Goat anti-Rat IgG (H+L) Highly Cross-Absorbed Secondary Antibody, Alexa Fluor Plus 647 (1:200, ThermoFisher, A-21247). |
|---|---|
| Validation | The validation information for the antibodies used in this study are available on the websites of the respective commercial providers as well as in published studies. No custom-made antibodies were used in this study. In addition, for anti-Vps15, anti-Vps34 and anti-Bmal1 antibodies, the specificity in WB, IP and IF assays was validated in our laboratory in human and mouse cell models upon knock-down (siRNA or shRNA)/KO and overexpression of respective cDNA constructs.<br>The molecular weight markers were used to identify proteins of the expected molecular weight for the target proteins.<br><br>The links to the manufacturer's website for used antibodies could be found below:<br>Vps15 Abnova - Validated in WB, IF, IHC and ELISA: https://www.abnova.com/products/products_detail.asp?catalog_id=H00030849-M02<br><br>Vps15 Genetex - Validated in WB, KD/KO, Protein overexpression: https://www.genetex.com/Product/Detail/PI3-kinase-p150-antibody-C1C3/GTX108953<br><br>p62 Abnova - Validated in WB, ELISA, IF, IHC, IP: https://www.novusbio.com/products/p62-sqstm1-antibody-2c11_h00008878-m01<br><br>B-Actin Sigma - Validated in IHC, ELISA, IF, WB: https://www.sigmaaldrich.com/FR/en/product/sigma/a5441<br><br>Tubulin Sigma - Validated in IF and WB: https://www.sigmaaldrich.com/FR/en/product/sigma/t9026<br><br>B-Catenin BD Biosciences - Validated in WB, IF, IHC, IP: https://www.bdbiosciences.com/en-fr/products/reagents/microscopy-imaging-reagents/immunofluorescence-reagents/purified-mouse-anti-catenin.610153<br><br>Lamin A/C Cell Signaling - Validated for WB and IHC: https://www.cellsignal.com/products/primary-antibodies/lamin-a-c-antibody/2032<br><br>Histone H3 Cell Signaling - Validated for WB, IHC, IF, Flow Cyt: https://www.cellsignal.com/products/primary-antibodies/histone-h3-d1h2-xp-rabbit-mab/4499<br><br>LC3 Nanotools - Validated for WB, ICC, IHC: https://www.labome.com/product/Nanotools/0231-100-LC3-5F10.html<br><br>GAPDH Santa Cruz- Validated for WB, IHC, IF: https://www.scbt.com/p/gapdh-antibody-fl-335<br><br>Rev-Erba Cell Signaling - Validated for WB, IP, ChIP: https://www.cellsignal.com/products/primary-antibodies/rev-erba-e1y6d-rabbit-mab/13418 |

Bmal1 Abcam - Validated for WB, ICC: https://www.abcam.com/products/primary-antibodies/bmal1-antibody-ab3350.html

Bmal1 Cell Signaling - Validated for WB, IP, ChIP: https://www.cellsignal.com/products/primary-antibodies/bmal1-d2l7g-rabbit-mab/14020

Clock Cell Signaling - Validated for WB and IP: https://www.cellsignal.com/products/primary-antibodies/clock-d45b10-rabbit-mab/5157

His-Tag Proteintech - Validated for WB, IP, IF: https://www.ptglab.com/products/His-Tag-Antibody-66005-1-Ig.htm

IPOA5 Proteintech - Validated for WB, ELISA, IP, IHC, IF: https://www.ptglab.com/products/KPNA1-Antibody-18137-1-AP.htm

Cry1 Origene - Validated for WB and IHC: https://www.origene.com/catalog/antibodies/primary-antibodies/ta342728/cryptochrome-i-cry1-rabbit-polyclonal-antibody

RNA Pol II Active Motif - Validated for ChIP, WB, IF, ICC: https://www.activemotif.com/catalog/details/39097/rna-pol-ii-antibody-mab

RNA Pol II pSer5 Chromotek - Validated for WB and ChIP: https://www.citeab.com/antibodies/2452247-3e8-rna-pol-ii-ser5-p-antibody-3e8

RNA Pol II pSer5 Abcam - Validated for WB, ChIP, ELISA, ICC, Dot Blot, Flow Cyt. IF: https://www.abcam.com/products/primary-antibodies/rna-polymerase-ii-ctd-repeat-ysptsps-phospho-s5-antibody-4h8-chip-grade-ab5408.html

Vps34 Cell Signaling - Validated for WB and IP: https://www.cellsignal.com/products/primary-antibodies/pi3-kinase-class-iii-d9a5-rabbit-mab/4263

Flag Sigma - Validated for WB, IP, IF, ICC, IHC: https://www.sigmaaldrich.com/FR/en/product/sigma/f1804?gclid=Cj0KCQjw9deiBhC1ARIsAHLjR2AI28S1O1XODfRmBg7nS1dpYv_1hvWPg695SGJdQtx_7Kbk3iCCnkMaAmzhEALw_wcB&gclsrc=aw.ds

HSP90 Proteintech - Validated for WB, IP, IHC, IF, Flow Cyt, ELISA: https://www.ptglab.com/products/HSP90-Antibody-13171-1-AP.htm

IgG Rabbit Cell Signaling - Validated for WB, IF, IHC: https://www.cellsignal.com/products/primary-antibodies/rabbit-da1e-mab-igg-xp-isotype-control/3900

IgG Mouse Santa Cruz - Validated for WB, IHC, IF: https://www.scbt.com/p/normal-mouse-igg

# Eukaryotic cell lines

Policy information about cell lines and Sex and Gender in Research

| Cell line source(s) | HEK293T (CRL-3216) and AML12 (CRL-2254) cells were acquired from ATCC. Mouse embyonic fibroblasts (MEFs) were generated by our laboratory from Vps15f/f and Vps34 f/f mice. MEFs were obtained from a pool of embryos from two different females and were passaged every three days before spontaneous transformation. The MEF line proliferating after p25 was considered spontaneously immortalized. |
| --- | --- |
| Authentication | All members of our laboratory are trained on the Best Laboratory Practices and safe experimenting including cell line culture in order to prevent contamination with different cell types (ICLAC guidelines).  HEK293T and AML12 cell lines were obtained from ATCC and were not further authenticated. For MEFs Vps15f/f or Vps34f/f knock-out was verified by western blots following Cre infections in every experiment. |
| Mycoplasma contamination | All cell lines used were bi-weekly tested for  contaminations with mycoplasma using commercial PCR Mycoplasma Detection Kit (ABM, #G-238). All tests were negative. |
| Commonly misidentified lines (See ICLAC register) | No misidentified lines were used in this study |

# Animals and other research organisms

Policy information about studies involving animals; ARRIVE guidelines recommended for reporting animal research, and Sex and Gender in Research

| Laboratory animals | Liver specific Vps15 knockout mouse line AlbCre+;Vps15f/f was derived from the strain #022624 (Jackson Laboratory, USA) crossed with AlbCre+ mice as reported (Nemazanyy et al., Nature Comm, 2015). Inducible hepatocyte-specific Vps15 knockout TtrCre+;Vps15 f/f line was generated by our laboratory as described in the methods sections.<br>Animals of 5 week old for AlbCre+;Vps15f/f and 12-16 week old for TtrCre+;Vps15f/f were used for the experimentation. All animals used in the study were fed ad libitum standard chow diet (Teklad Global 2918, 18% protein, irradiated) and kept under 12h/12h (8am/8pm) light on/off cycle under an ambient temperature (+21-22C) and humidity ranging between 50-60%. Animals were sacrificed at indicated time points in text, figures and figure legends. All animal studies were performed by authorized users in compliance with ethical regulations for animal testing and research. |
| --- | --- |

| Wild animals | No wild animals were used. |
|---|---|
| Reporting on sex | Male and female mice were used in this study as it is specified in figure legends. |
| Field-collected samples | The study did not involve samples collected in the field. |
| Ethics oversight | The study was approved by the ethical committee of University Paris Cité (authorization number APAFIS#32312 and APAFIS#14968). |

Note that full information on the approval of the study protocol must also be provided in the manuscript.

# ChIP-seq

## Data deposition

☒ Confirm that both raw and final processed data have been deposited in a public database such as GEO.

☒ Confirm that you have deposited or provided access to graph files (e.g. BED files) for the called peaks.

| Data access links<br>*May remain private before publication.* | The data are deposited in the Gene Expression Omnibus (GEO) under accession code GSE229551 |
|---|---|
| Files in database submission | ChIP-Seq files:<br><br>ChIP_Bmal1_Vps15LKO_rep1.bed<br>ChIP_Bmal1_Vps15LKO_rep2.bed<br>ChIP_Bmal1_WT_rep1.bed<br>ChIP_Bmal1_WT_rep2.bed<br>ChIP_H3K27ac_Vps15LKO_rep1.bed<br>ChIP_H3K27ac_Vps15LKO_rep2.bed<br>ChIP_H3K27ac_WT_rep1.bed<br>ChIP_H3K27ac_WT_rep2.bed<br>ChIP_Vps15_rep1.bed<br>ChIP_Vps15_rep2.bed<br>Input_WT.bed<br>Input_Vps15LKO.bed<br><br>ChIP_Bmal1_Vps15LKO_rep1.fq.gz<br>ChIP_Bmal1_Vps15LKO_rep2.fq.gz<br>ChIP_Bmal1_WT_rep1.fq.gz<br>ChIP_Bmal1_WT_rep2.fq.gz<br>ChIP_H3K27ac_Vps15LKO_rep1.fq.gz<br>ChIP_H3K27ac_Vps15LKO_rep2.fq.gz<br>ChIP_H3K27ac_WT_rep1.fq.gz<br>ChIP_H3K27ac_WT_rep2.fq.gz<br>ChIP_Vps15_rep1.fq.gz<br>ChIP_Vps15_rep2.fq.gz<br>Input_WT.fq.gz<br>Input_Vps15LKO.fq.gz |
| Genome browser session<br>(e.g. UCSC) | Not applicable. |

## Methodology

| Replicates | n=2 mice per condition were sequenced and analyzed. |
|---|---|
| Sequencing depth | Average of 30million (100bp-Paired End) clean reads following filtering low quality reads, N reads, and adapter sequences. |
| Antibodies | Vps15 Abnova, H00030849-M02. Bmal1  Abcam, ab3350. H3K27ac Cell Signaling, #8173. |
| Peak calling parameters | ChIP-Bmal1 with the following parameter: callpeak -t -c -f BAM -g mm -p 0.01<br>H3K27ac peak calling was performed using: callpeak -t -c -f BAM -g mm –broad<br>Vps15 was performed using: callpeak -t -c -f BAM -g mm -p 0.001 --broad –nomodel |
| Data quality | Data filtering was performed with SOAPnuke to remove adaptor sequences and low-quality reads. Data filtering parameter was: SOAPnuke filter -l 5 -q 0.5 -n 0.1 -Q 2 -c 40. |
| Software | Sequencing was performed with the DNBSEQ-400 sequencer (Beijing Genomics Institute, China). All data were analyzed with the pipeline: Bowtie2 for alignment to mm10, SAMtools for indexing and sorting SAM files, MACS2 for peak calling, R package Diffbind (V3.10.0) for differential peak calling, R package ChIPseeker and clusterProfiler for peak annotation and GO/KEGG pathway enrichment. Motif analysis was performed using HOMER, (findMotifsGenome.pl -mask -size 200). IGV_2.8.13 used for bigwig/bam file visualization. |

