## [Peer Review File · Nature Cell Biology]

Peer Review Information

Journal: Nature Cell Biology

Manuscript Title: Class 3 PI3K transcriptionally co-activates circadian clock to drive purine synthesis

Corresponding author name(s): Dr Ganna Panasyuk

Editorial Notes:

**Redactions –
unpublished data**

Reviewer Comments & Decisions:

Decision Letter, initial version:
--

Dear Dr Panasyuk,

Thank you for submitting your manuscript "Class 3 PI3K transcriptionally co-activates circadian clock to drive purine synthesis", to Nature Cell Biology and thank you very much for your patience while the manuscript was undergoing external peer review. The manuscript has now been seen by 4 referees, who are experts in PI3K, phosphoinositide signaling (Referee #1); circadian clock biology (Referee #2); PI3K, phosphoinositide signaling (Referee #3), and circadian clock biology/liver (Referee #4), and whose comments are pasted below. In light of their advice, we regret that we cannot offer to publish the study in Nature Cell Biology.

As you will see, although the reviewers found this work interesting, they raised serious concerns that question the strength of the data and of the novel conclusions that can be drawn at this stage. In particular, they requested more cell biological insight into the mechanism by which VPS15 operates to control BMAL1-dependent transcription, they questioned the rigor of the experimental design as well as inconsistencies in the impact of VPS15 loss, undermining the current conclusions. The referees additionally asked for more evidence to address the impact of Vps15 on BMAL1 transcriptional output and to strengthen the independence of VPS34.

We discussed the referee feedback in depth within the NCB editorial team and found it significant and valid. Upon careful evaluation of the points raised by the reviewers, we find the present dataset too preliminary to further consider the work at the journal. We are also concerned that the experiments that would be required to sufficiently bolster the study would entail significant revisions that would extend well beyond six months, and we thus cannot invite a revision that would be unlikely to be completed within a reasonable timeframe.

We are very sorry that we could not be more positive on this occasion, but we thank you for the opportunity to consider this work. We hope that you find the reviewers' comments below helpful as you decide how to move forward with this work.

With kind regards,
Melina

Melina Casadio, PhD
Senior Editor, Nature Cell Biology
ORCID ID: <https://orcid.org/0000-0003-2389-2243>

Reviewers' comments:

Reviewer #1 (Remarks to the Author):

The manuscript by Alkhoury et al has define the co-transcriptional function of class III PI 3-kinase

(VPS34) in regulation of clock-controlled gene expression. Authors have shown VPS34 kinase-independent role of VPS15 adaptor subunit in co-transcriptional activation of the clock-controlled genes and their regulators. Authors have provided compelling evidence and high quality of works and should be interesting for broader readership of Nat Cell Biol audiences. We have raised following few points that would help authors to strengthen the manuscript further.

1. As authors agree that class III PI 3-kinase function has been largely attributed to phosphoinositide-dependent function in endocytosis and endosomal vesicle trafficking in the cytoplasm, though there are paucities of the studies showing the nuclear transcriptional activation or repression function of class III PI 3-kinase. Thus, authors have uncovered a novel function of class III PI 3-kinase in transcriptional co-activation of clock-controlled gene expression. However, authors seem to have ignored to provide the molecular details of how VPS34 or adaptor subunit Vps15 can translocate into the nucleus by its own or via other interactomes. Authors need to discuss this important point in the manuscript.
2. The authors need to provide the rationale to start their study using the Vps15 subunit KO mice but not the lipid kinase Vps34, as the introduction shows a link between circadian lysosomal activity to class 3 PI3K, which is attributed to the PI-kinase function. Why do they check the regulatory subunit instead of the kinase in the first figure? The work could be improved by comparing to Vps34 KO mice.
3. What is the impact of PI3P generation upon Vps15 KO? Would impact on PI3P generation lead to a change in the circadian clock?
4. Figure 1b, what is the expression level of Vps34 upon Vps15 KO in the WB? Also need to probe for Vps15 to validate the KO.
5. In the later part of the manuscript, the authors have shown Vps15 interaction with BMAL1-CLOCK complex. Does this interaction provide the mechanism for Vps15 to translocate into the nucleus?
6. The authors need to provide IF staining of PI3P, Vps34, Vps15, BMAL1 in the wild-type cells and Vps15 KO cells with nuclear staining to demonstrate their subcellular localization, especially nuclear localization upon Vps15 KO. Their model shows subcellular translocation of Vps15 and nuclear complex of it with BMAL1, but they have only done the subcellular fractionation and performed WB, where imaging data would be essential for supporting those claims but are largely missing.
7. Figure 3b is at poor quality, no scale bar, only 2~3 cells are shown and without quantification. Also, is there PLA between endogenous Vps15 and BMAL1?
8. The key finding of the authors in this manuscript is the co-transcriptional activator function of the VPS15 with BMAL1-CLOCK complex in the transcriptional regulation of the clock-controlled genes/regulators. However, the Vps15 knockout seems to have noticeable impact in the expression level of BMAL1, especially in the liver and authors seem ignoring this important point based on results in MEFs. In this context, VPS15 seems to have role in the stability of BMAL1 and BMAL1-CLOCK complex impacting on transcriptional co-activation of BMAL1-CLOCK complex target genes/regulators.

Reviewer #2 (Remarks to the Author):

A. Summary of the key results

Here, the authors show that targeting the subunits of the Class 3 PI3K influences the activity of CLOCK-BMAL1 in mouse liver, including classic clock-controlled genes like *Per2*, *Dbp*, and *Nr1d1*, as well as those involved in purine biosynthesis. There is quite a lot of data here, from genome-wide studies of transcript abundance and chromatin occupancy to cellular biochemical studies looking at protein abundance, localization, interactions, and transcriptional activation. Overall, the major conclusion is that the regulatory subunit VPS15 binds to BMAL1 to coactivate transcriptional activation in the nucleus independent of the catalytic subunit VPS34. This conclusion seems largely supported by

the data, although most of it is indirect (see below). In addition, some issues with the rigor of experimental design and discrepancies between the phenotypes (in constitutive liver knockout (LKO) of the regulatory subunit VPS15 in mice vs. the Cre-inducible deletion of VPS15 in MEFs or in the liver vs. chemical inhibition or Cre-inducible deletion of the catalytic subunit VPS34) confound a clear and compelling mechanistic interpretation here.

B. Originality and significance: if not novel, please include reference

To my knowledge, this work is the first to look at the effects of Class 3 PI3K subunits on CLOCK-BMAL1 activity. If the major conclusion was better supported by the data, it could provide a nice basis to understand the clock-dependent regulation of purine biosynthesis first discovered by Fustin and colleagues (ref. 43). Other recent work shows that CLOCK-BMAL1 activity is regulated directly by 'unexpected' proteins like the RelA subunit of NF- κ B (PMID 34807912), so this could add to a growing body of work on regulatory mechanisms of this central clock transcription factor.

C. Data & methodology: validity of approach, quality of data, quality of presentation

1. The 6-hr sampling frequency is not ideal for studying circadian rhythms. According to the Guidelines for Genome-Scale Analysis of Circadian Rhythms (PMID 29098954), sampling should ideally occur every 2 hours (or every 4 hr at a minimum).
2. The immunohistological staining of BMAL1 in mouse liver presented in Ext. Data Fig. 1f shows an apparently high level of non-nuclear protein staining, but this does not seem to match the western blot data from panel d of the same figure, where cytosolic levels of BMAL1 are significantly reduced in VPS15LKO livers. Can the authors explain this discrepancy?
3. The different effects on BMAL1 expression and chromatin binding in VPS15LKO mice and the acutely driven deletion by Cre in MEFs make it hard to conceptually integrate the whole animal and cell-based data, diminishing confidence in the conclusions. Inhibition of lipid kinase activity with chemical inhibitors altered gene expression but with gene-specific effects that varied quite widely (see below). By contrast, depletion of the VPS34 protein by Cre had minimal effect on the expression of clock-controlled genes, and it also reduced VPS15 levels, although apparently not enough to see the anticipated VPS15-dependent effects. Neither Cre-treated line reduced BMAL1 binding in MEFs as assessed by ChIP, although it was significantly decreased in the VPS15LKO livers. Therefore, it's unclear how the comparison of transcripts in WT and VPS15LKO livers (Fig. 3 and 4) adds to the clarity of the conclusions here. Data from the Cre-inducible deletion of VPS15 in the liver are compelling, as are data in primary hepatocytes, but there are enough confounding results here that were not clarified or explained amongst the different experimental conditions that the strength and clarity of the conclusions are diminished.
4. Variable effects on the abundance of clock-controlled genes with Class 3 PI3K inhibitors do not rigorously support the conclusion/header of that paragraph as stated, that there is a "lipid kinase independent role of class 3 PI3K in the clock control." The range of effects on the several transcript levels range from decreased to increased abundance, or no effect at all, suggesting that something more complicated is going on.
5. The statement that VPS15 protein displays diurnal rhythmicity based on an approximate change of ± 10 -20% in protein abundance (Ext. Data Fig. 4) should be modified to say "...showed a modest diurnal rhythmicity..."

D. Appropriate use of statistics and treatment of uncertainties

Seems appropriate.

E. Conclusions: robustness, validity, reliability

Please see the points made in sections C and F. The presence of key data needed to support the conclusions in both main and extended figures made it hard to read the manuscript, as it required constantly flipping back and forth to look at both sets of figures to critically evaluate the authors' conclusions. This could be improved by better streamlining data presentation in the manuscript.

F. Suggested improvements: experiments, data for possible revision

1. Most of the data generally support a role for VPS15 in coactivation of CLOCK-BMAL1-dependent genes, but they only provide indirect evidence for this conclusion (e.g., genome occupancy, gene expression, nuclear proximity, and the co-IP of VPS15 and BMAL1). The luciferase reporter data in Fig. 3c showing enhancement of BMAL1-dependent E-box-driven luc reporter activity upon coexpression with VPS15 is one way to interrogate this activity in a relatively direct manner. The data look interesting, but the gold standard in the field is to co-express BMAL1 with its partner CLOCK to properly assess activity of the transcription factor in HEK293T cells. With the data displayed as "relative activity", it is not clear that this represents robust activity that is physiologically relevant. For example, how does coactivation by VPS15 activity compare to the activity of known CLOCK-BMAL1 coactivators p300 or CBP? These assays should be repeated looking at CLOCK-BMAL1 activity to support the statement that VPS15 coactivates transcriptional activity of the heterodimer. Alternatively, showing that VPS15 can coactivate BMAL1-dependent activation in a heterologous one-hybrid assay would also provide strong evidence for coactivation of the BMAL1 transcriptional activation domain (TAD) by VPS15, unless it works by binding a completely different region of BMAL1. Along those lines, does VPS15 require the BMAL1 TAD for binding, as other reported coactivators do?

G. References: appropriate credit to previous work?

Seems appropriate.

H. Clarity and context: lucidity of abstract/summary, appropriateness of abstract, introduction and conclusions.

The abstract is reasonably clear, although some editing could increase readability. Breaking the introduction and other sections down into paragraphs less than a page in length would help a lot with clarity. Technically, CLOCK and BMAL1 are subunits of the heterodimeric transcription factor and should not be referred to individually as transcription factors (line 55).

Reviewer #3 (Remarks to the Author):

Summary: In this manuscript, Alkhoury et al. investigated the mechanistic connection between circadian clock-driven metabolic rhythmicity and a critical component in nutrient sensing pathways, class III PI3K, which consists of a kinase subunit Vps34 and a regulatory subunit Vps15. The mammalian molecular clock comprises the heterodimer of two transcription factors, CLOCK and BMAL1, along with their target genes, many of which are negative transcriptional regulators of the CLOCK-BMAL1 complex to form transcriptional/translational feedback loops. Circadian rhythm of this molecular clock can be regulated by multiple nutrient sensing pathways, including mTOR/Class I PI3K, AMPK signaling and autophagy. Class III PI3K are functionally connected to all these pathways, yet its direct functional interaction with the molecular clock remains poorly understood. Using a liver Vps15 knockout mouse model (Vps15LKO), the authors found that multiple gene targets of BMAL1, including Nr1d1, Dbp, and Per2, showed abnormal transcript levels and deregulated circadian rhythmicity in livers of Vps15LKO mice. Similar results were observed in Vps15-null MEF cells. Surprisingly,

treatment of Vps34 kinase inhibitors and acute depletion of Vps34 in MEFs showed little effects in BMAL1 transcriptional activity. Therefore, the authors proposed that Vps15 functions independently from Vps34 as a transcriptional coactivator of BMAL1. They found that both Vps15 and Vps34 could localize to the nucleus, but only Vps15 interacted with BMAL1 in the nucleus and was capable of activating *in vitro* transcription from a promoter with BMAL1 binding motif E-box. Further, the authors showed that Vps15 was enriched at the promoter regions of *Nr1d1*, *Dbp* in wild-type (WT) mouse liver extracts, WT MEFs and Vps34-null MEFs. GO pathway analyses for data acquired from BMAL1 and Vps15 ChIP-Seq and bulk transcriptomics in mouse livers pointed out multiple metabolic processes under co-regulation of Vps15 and BMAL1. The authors then performed targeted metabolomics in livers harvested from wild-type or Vps15LKO mice, and they found 63 metabolites from diverse chemical classes showed circadian rhythmicity in WT livers, with 42 of them losing the rhythmicity in Vps15LKO livers. A number of these deregulated metabolites were involved in the *de novo* synthesis of purines, and they showed that a key enzyme in this biosynthetic pathway, PPAT, was under transcriptional control of Vps15/BMAL1, demonstrating rhythmic *de novo* purine synthesis in liver depends on Vps15. At last, to complement studies done with the chronic Vps15LKO mouse model, the authors developed a tamoxifen-inducible Vps15 liver knockout mouse model (Vps15iLKO), and again found that induced Vps15 depletion caused decreased transcript and protein levels and aberrant rhythmicity of BMAL1 target genes, including the newly confirmed Vps15/BMAL1 target *Ppat*. In conclusion, the authors reported that Vps15 has novel function in the nucleus, independent of Vps34, to serve as a transcriptional coactivator of BMAL1, a core molecular component, to regulate the transcriptions of circadian genes and couple the circadian clock to maintain energy homeostasis and metabolic rhythmicity. Overall, the findings here present an unexpected mechanism by which expression of BMAL1 target genes can be regulated. The physiological relevance of the findings remains underexplored, including what factors upstream of Vps15/34 are responsible for modulating the Vps15-BMAL1 interaction and the physiological consequences of the effects of the observed changes to purine metabolism.

Other major comments:

1. One important claim the authors made in the manuscript is that the lipid kinase subunit of the Class III PI3K, Vps34, is dispensable for the clock regulation. To support this claim, the authors used two structurally unrelated Vps34 inhibitors to block lipid kinase activity and generate Vps34f/f MEFs to semi-acutely deplete Vps34. The results of these experiments are a little perplexing. For example, different inhibitors showed mixed effects in *Dbp* levels, and *Per2* was sensitive to Vps34 modulation but not *Nr1d1*. Do the two inhibitors use different mechanisms to suppress Vps34 activity? Why are some transcript levels strongly affected by Vps34 depletion/inhibition, but others were not (yet all of them are reported as gene targets of BMAL1)? Further, in Vps34f/f MEFs, the protein levels of REV-ERBa were shown not to be reduced by Vps34 depletion (Fig. 2d), but if we compare the levels of REV-ERBa in the GFP control of Vps34f/f and Vps15f/f cell lines (western blot in Fig. 1g), we can see much higher REV-ERBa level at earlier timepoints (16-24 h) compared to later timepoints (28 – 36 h) in Vps15f/f cells, but not as a big difference in Vps34f/f cells. Does this indicate that REV-ERBa levels were lower in Vps34f/f GFP control cells during those time points to start with, and thus upon Vps34 depletion no reduction was observed? This part of the study would be more convincing if the authors could provide more detailed explanations for the discrepancy in the responses to different treatments instead of glossing over the observations. Also, the authors should conduct the rhythm analysis on these inhibitor experiments like in the Vps15LKO livers and Vps15-null MEFs, if possible. Currently, only the differences in the target gene transcript level are reported.
2. The transcript levels of BMAL1 target genes in Vps34-null MEFs were similar to the GFP control (Fig. 2e, Extended Data Fig. 3b), which was drastically different when compared to Vps15-null cells (Fig. 1h, Extended Data 2c). However, in Vps34f/f MEFs, the overall protein level of Vps15 was also reduced to ~30% (Fig. 2d, Extended Data Fig. 3a). The authors explained that this observation was due to higher retention of Vps15 in the nucleus of Vps34-null cells (70% reduction in cytosol vs 30% reduction in the nucleus). However, since no cytosolic protein markers were included in the representative western

blots, it is hard to tell the purity of the nuclear soluble fractions and thus fully trust the quantification.

3. Also, the authors should include the ratio of cytosolic vs. nuclear pools of Vps15 in control GFP MEFs and in Vps34-null MEFs? These data would support the authors' explanation if they can show the remaining level of Vps15 in the nucleus of Vps34-null cells is comparable to the Vps15 level in the nucleus of the control GFP cells, so the remaining Vps15 in the Vps34-null nucleus is sufficient to support its function as a transcriptional coactivator.

4. For subcellular fractionation experiments done in the manuscript, only western blots shown in Extended Data Fig. 4a and b included a nuclear protein marker for the cytosolic fraction, and a cytosolic protein for the nuclear fraction to demonstrate the purity of these fractions. This practice should be extended to at least one representative western blot for subcellular fractionation performed in different cell lines (e.g. Vps15f/f MEF, Vps34f/f MEF, etc) and from mouse liver extracts to show the robustness of the procedure.

5. The authors performed co-IP, PLA and E-box-Luciferase reporter experiments to demonstrate that Vps15 interacts with BMAL1 in the nucleus, is able to initiate *in vitro* transcription, and that Vps34 was not needed for all these processes. The following experiments are necessary to support the proposed mechanism.

(a) For the co-IP experiment shown in Extended Data Fig. 5a and b, the authors should probe for Vps34 in the co-IP from the different fractions to assess if Vps34 and Vps15 complex formation is different in the cytosol and in the nucleus. Supposedly, Vps34 might show less interaction with Vps15 in the nucleus.

(b) Alternatively, the authors could consider repeating the FLAG IP shown in Fig.3a in cytosol and nuclear fractions, and similarly probe for Vps34 in the co-IP in addition to BMAL1.

(c) For the PLA experiment shown in Fig. 3b, the authors should add a panel for FLAG-BMAL1 and Vps34 to show negative interaction for these two proteins.

(d) Also, using these same antibodies for FLAG (BMAL1), Vps15 and potentially Vps34 to perform IF in parallel with the PLA procedures would allow assessment of the subcellular localizations of these proteins (BMAL1, Vps15 and Vps34 were all reported to show both cytosolic and nuclear localization), and would further support the authors about the Vps15-BMAL1 interaction claim if the PLA signal is only observed in the nucleus for Vps15 and FLAG-BMAL1.

(e) Showing no Vps34 enrichment at the promoters of *Nr1d1* and *Dbp* would also strengthen the statement that Vps34 was dispensable for the transcription of clock related genes.

6. The authors used the JTK-CYCLE algorithm from R for circadian rhythmicity analysis of proteins, mRNAs and metabolites levels in the manuscript. The authors should provide brief explanation of the analyses in the method section to help the audiences better understand how the calculations were done and interpret the data correctly. For example, in Extended Data Fig.4, the authors demonstrated that the protein and transcript levels of Vps15 but not Vps34 show rhythmic pattern according to the analysis by the JTK-CYCLE algorithm. However, just from looking at the representative western blots and the quantification plots for the protein and mRNA levels of these two proteins, sometimes both Vps15 and Vps34 seemed to show oscillating expression pattern (Extended Data Fig.4d), or neither seemed to show a pattern (Extended Data Fig.4f), but the p-values from the rhythmic analysis for the two proteins are very different.

7. ChIP-Seq and ChIP-qPCR analyses for Vps15 showed much lower enrichment at transcription start sites and at promoter regions of *Nrdr1*, *Dbp* and *Ppat* than BMAL1, indicating that the Vps15 and BMAL1 interaction is sub-stoichiometric (many molecules of BMAL1 not bound to Vps15). The authors should discuss how a sub-stoichiometric Vps15 interaction affects the overall BMAL1 transcriptional activity so drastically (as seen by tens and hundreds fold differences in some BMAL1 target genes transcript levels in WT vs Vps15LKO or Vps15iLKO livers or Vps15-null cells).

8. Based on data shown in the manuscript, Vps15 did not seem to facilitate the recruitment of BMAL1 to chromatin (Fig. 2g, Extended Fig. 7f, line 202-206; reduced BMAL1 occupancy at target gene promoters observed in the chronic Vps15LKO liver was probably due to decreased BMAL1 protein level), and there was not enough data showing Vps15 depletion negatively regulates the BMAL1-CLOCK heterodimer formation. Also, the nuclear pool of Vps15 seemed to show rather stable levels

despite a minor reduction during the dark phase (Extended Data Fig. 4c-e). Therefore, for the proposed working model that Vps15 couples transcriptional activity of the circadian clock with regulation of energy homeostasis, the authors should provide some more evidence of how Vps15 may facilitate the activation of BMAL1 as its transcription coactivator, and how could this activated transcriptional machinery for clock genes and purine de novo synthesis be terminated/paused when necessary. Does Vps15 constitutively interact with BMAL1 in the nucleus to bolster its transcriptional activity, and thus the rhythm of the system only comes from the regulation of BMAL1 expression from other clock components, or the Vps15-BMAL1 interaction itself is also rhythmic and subject to some regulations upon receiving signaling cues?

Minor comments for specific figures:

1. Fig. 1a: For the time course plots like the two shown in Fig.1a, if there are no data points between adjacent time points, the data point at each time point should not be connected by smoothed/curve lines, which may show visually deceiving trends, but rather by straight lines.
2. Fig. 1b (and for the rest): Representative Western blots should include molecular weight markers (The authors included MW in source data, but not in figures).
3. For Fig. 1b and Extended Data Fig. 1b and 1d: the protein level of BMAL1 in Vps15LKO liver at ZT0 and ZT12 for these two experiments were inconsistent. There was no obvious difference in BMAL1 level at ZT0 or ZT12 in WT vs Vps15LKO livers from data shown in Fig. 1b/Extended 1b, but then in Extended 1d, BMAL1 level in Vps15LKO liver was lower in both cytosol and in nucleus at both time points. This discrepancy should be explained/resolved.
4. Fig. 1e (and for the rest): In accordance with changing standards in the field toward better demonstration of all data points, the authors should replace bar graphs with scatter plots to show individual data points for quantifications.
5. Extended Data Fig. 1e: the authors stated that less CLOCK was in complex with BMAL1 in Vps15LKO livers because of lower level of CLOCK the co-IP with BMAL1. However, there was less CLOCK in the lysate input as well. Did the author take this into account when performing quantification?
6. Extended Data Fig. 1e: it would be more helpful if the authors could add arrow heads pointing to the staining of BMAL1 in the nucleus.
7. Fig. 3b: Images shown here are missing scale bars, and it would be convenient for the readers if the authors could indicate what color is corresponding to which signal (likewriting DAPI in blue font and PLA in red font, for example). In this particular experiment, the GFP fluorescence was only observed in the nucleus of the cell, which is strange, as it should be in cytosol as well. Also, co-expression of GFP from another plasmid is actually not helpful to visualize cells successfully transfected with the FLAG-BMAL1, since some cells may only receive one of the plasmids.
8. Fig. 3e: the Vps15mut is a point mutant of E200R, but does that single mutation explain its different migration from the WT form in this blot? The WT and mutant proteins showed similar migration in Fig. 3d (FLAG blots).
9. For numbers with decimal on the y-axis of many plots, periods should be used instead of commas.

Reviewer #4 (Remarks to the Author):

In this manuscript (NCB-A48597), "Class3 PI3K transcriptionally co-activates circadian clock (CC) to drive purine synthesis" Alkhoury et al., have utilized various approaches in several model systems to provide evidence for VPS15 to be a co-activator of BMAL1 and regulate metabolism, specifically purine synthesis. Experiments are well controlled, and presentation of results are clear. The statistics is appropriate.

As a general comment, I previous investigations have provided evidence for links between other classes of PI3K signaling (but not class 3) and 'Clock' system, thus somewhat diluting the

Additionally, the present study raises several concerns/questions (see below).

Major Points:

1.The manuscript initiates (Fig.1a, Extended Data Fig. 1a) and finishes (Extended Data Fig.7e) with analyzing the expression (transcripts) of various CC-components. It is clear from the results with VPS15iLKO mice that: mutation of VPS15 kills the CC-oscillator, with significant reduction in the expression of all the CC genes including BMAL1 and CLOCK.

This is an extremely crucial observation and raises the question- Why in absence of VPS15 the RORE-dependent transcription of BMAL1 and CLOCK is reduced?

2.Continuing with VPS15iLKO system: How does the authors explain the reduced nuclear accumulation of BMAL1. Is this reduction in nuclear accumulation also happens for other CC-components too? Could the authors speculate on the mechanism for this?

3.The study demonstrates a protein -protein interaction between the BMAL1 and VPS15. It will be a great help if the region or the residues in both BMAL1 and VPS15 which helps in the interaction could be determined. Is it a direct interaction or indirect?

4.The study claims VPS15 as a co-activator of BMAL1-dependent transcription for a certain class of genes. Considering BMAL1/CLOCK are pioneer transcription factors (in mouse liver), how VPS15 co-activates this transcriptional activity? This is significant given the fold enrichment for VPS15 in ChIP assays seem to be manyfold less than that of BMAL1? Can authors determine the stoichiometry?

5.The ChIP assays were performed at ZT6, which is considered to be the peak of BMAL1 binding to DNA. But, it is well known that enhancer remodeling starts earlier with production of eRNAs. Does VPS15 plays any role in this process for its target genes?

6.Fig. 3h: Analyses of VPS15 genomic recruitment indicates that its recruitment seems to be far more downstream of the TSS rather than upstream. Is VPS15 a part of the elongating RNA PolII complex rather than initiation complex? This distinction is necessary to provide possible mechanism of its suggested role as a co-activator.

7.Fig.3c: This experiment describes the result of luciferase assay from the transfected cells. This is not an actual classically described 'in vitro transcription assay' as mentioned in the text.

8.At least some microscopic evidence is necessary (along with presented western blots) to demonstrate that autophagy is indeed affected by VPS15 mutation.

9.If possible a demonstration of reduced purine incorporation in nucleotides in VPS15 and BMAL1 mutant cells will be extremely convincing.

Minor Point:

The scheme and its legend could be improved.

Author Rebuttal to Initial comments

Point-by point response

Reviewer #1

The manuscript by Alkhoury et al has define the co-transcriptional function of class III PI 3-kinase (VPS34) in regulation of clock-controlled gene expression. Authors have shown VPS34 kinase-independent role of VPS15 adaptor subunit in co-transcriptional activation of the clock-controlled genes and their regulators. Authors have provided compelling evidence and high quality of works and should be interesting for broader readership of Nat Cell Biol audiences. We have raised following few points that would help authors to strengthen the manuscript further.

We would like to thank the reviewer for highlighting the novelty and importance of our study towards discovering a novel role of class 3 PI3K in regulating nuclear transcription, a function that our findings suggest has broad physiological significance for metabolic rhythmicity. We thank the reviewer for all comments that have allowed us to further improve the work by adding additional controls and clarifying already provided data.

1. As authors agree that class III PI 3-kinase function has been largely attributed to phosphoinositide-dependent function in endocytosis and endosomal vesicle trafficking in the cytoplasm, though there are paucities of the studies showing the nuclear transcriptional activation or repression function of class III PI 3-kinase. Thus, authors have uncovered a novel function of class III PI 3-kinase in transcriptional co-activation of clock-controlled gene expression. However, authors seem to have ignored to provide the molecular details of how VPS34 or adaptor subunit Vps15 can translocate into the nucleus by its own or via other interactomes. Authors need to discuss this important point in the manuscript.

Response:

We fully agree with the reviewer that finding a functional nuclear pool of class 3 PI3K represents a major novelty of our work. So far, most research has focused on the cytosolic roles of class 3 PI3K while the nuclear localization of both Vps15 and Vps34 was reported only in non-animal cells (yeast and plants)^{1,2}. We thank reviewer for this insightful comment that allowed us to provide further evidence of nuclear pool of class 3 PI3K. Given their high molecular weight (150kDa and 100kDa for Vps15 and Vps34, respectively), we hypothesized that Vps15 and Vps34 undergo active nuclear transport. In this revised manuscript, we provide following findings that tested this hypothesis:

1) We show that both Vps15 and Vps34 could undergo active importin-mediated nuclear import, as evidenced by the interaction of Vps15 and Vps34 with importin alpha (IPOA5) (**R-Fig. 1a**, corresponding to **Extended Data Fig. 5g** in the revised manuscript). This is further supported by decreased nuclear levels of Vps15 and Vps34 in MEFs treated for 4 hours with ivermectin (**R-Fig. 1b**, corresponding to **Extended Data Fig. 5f** in the revised manuscript). Finally, we did not observe major changes in nuclear Vps15 and Vps34 in response to inhibition of Crm1-dependent export by LMB (**R-Fig. 1c**).

2) Moreover, bioinformatic analyses predicted that Vps15 contains putative NLS motifs predicted by ELM, NLS mapper, NLStradamus, and PSORT II (**R-Fig. 2a**). We substituted the conserved K750_K751_R752 residues to neutral Ala amino acid. First, we did not observe major changes in the nuclear protein levels in Vps15-K750A_K751A_R752A-Flag (Vps15-NLS-mut) compared to Vps15wt-Flag construct (**R-Fig. 2b**). Second, the luciferase assay using E-box-Luc reporter did not reveal difference between Vps15-K750A_K751A_R752A-Flag (Vps15-NLS-mut) and Vps15wt in their capacity for co-activation of BMAL1-CLOCK, suggesting this predicted NLS signal is not involved in the nuclear function of Vps15. We cannot rule-out that the putative NLS that we mutated in Vps15 is not functional. Moreover, it is possible that Vps15 (and potentially Vps34) could contain complex NLS/NES signals (bipartite or non-canonical) or are co-transported with other proteins. We have briefly discussed these open questions in the revised manuscript (**line 530-534** in the revised manuscript).

R-Figure 1. **a**, Immunoblot analyses of IPOA5 co-immunoprecipitated with ectopically expressed Vps15 and Vps34 from HEK293T cells using anti-Flag antibody (ev, empty vector). **b**, **c**, Immunoblot analysis of nuclear soluble protein extracts using indicated antibodies in MEF cells treated with Ivermectin (**b**) or Leptomycin B (LMB) (**c**) for indicated times. Densitometric analyses of Vps15 and Vps34 protein normalized to Lamin B1 or Lamin A/C presented as folds over vehicle treated cells. Data are presented as means \pm SEM, n=2-5 independent experiments.

R-Figure 2. **a**, Schematic representation of a localization within the domain structure of Vps15 protein of the putative Nuclear Localization (NLS) sequence predicted by four algorithms (ELM, NLS mapper, NLStradamus and PSORT II). The evolutionary conservation of putative NLS and the amino substitutions in the NLS mutant are presented. **b**, Immunoblot analysis of nuclear soluble protein extracts using indicated antibodies in HEK293T cells transiently expressing either Vps15 wild type or Vps15-NLS-mut. **c**, Luciferase assay in HEK293T cells co-transfected with E-box-Luc reporter construct, BMAL1, CLOCK together with empty vector or plasmids expressing Vps15wt or Vps15-NLS-mut. The normalized luminescence presented as fold difference over the cells expressing only E-box-Luc reporter. Data are means \pm SEM of n=4 independent experiments. #: P<0.05, 2-tailed, unpaired Student's t test.

3) Furthermore, we have shown that nuclear soluble and euchromatin levels of Vps15 and Vps34 are diurnal and peak in light phase, coinciding with the peak of BMAL1 binding and transcriptional activity (**R-Fig. 3a-c**, corresponding to **Extended Data Fig.5b-d** in the revised manuscript).

R-Figure 3. Immunoblot analysis of nuclear soluble (**a**) and euchromatin fraction (**b**) from livers of WT mice collected at the indicated time points. Densitometric analysis of Vps15 and Vps34 normalized to total protein (TGX gels) are presented as folds over ZT0. Data are presented as means \pm SEM, n=4. Rhythmicity pattern was identified based on JTK_CYCLE analysis ($p < 0.05$). **c**, Average of normalized Vps15 and Vps34 proteins in light phase (ZT0 to ZT8) or Dark phase (ZT12 to ZT20) in nuclear soluble (as in **a**) and euchromatin levels (as in **b**). Data are presented as means \pm SEM, n=12, # $p < 0.05$ vs Light, 2-tailed, unpaired Student's t test.

Finally, given that BMAL1 transcriptional activity is inhibited by fasting (**R-Fig. 4a**, corresponding to **Extended Data Fig. 10a** in the revised manuscript), we tested whether nuclear levels and chromatin bound Vps15 respond to starvation in liver. Notably, Vps15 protein levels in nucleus were decreased in livers of fasted control mice but not in cytosol fraction (**R-Fig. 4b**, corresponding to **Extended Data Fig. 10b** in the revised manuscript). We found that neither BMAL1

protein expression nor its recruitment to promoters of *Nr1d1* or *Ppat* was impacted by fasting in livers of wild type mice (R-Fig. 4c, corresponding to Fig. 6k in the revised manuscript). However, consistent with decreased Vps15 nuclear levels, Vps15 recruitment to promoters of BMAL1 target genes was abrogated in livers of fasted mice (R-Fig. 4c, corresponding to Fig. 6k in the revised manuscript).

R-Figure 4. a, Relative transcript levels of indicated genes in liver tissue collected at ZT6 of WT and Vps15LKO mice *ad libitum* fed or 24h fasted. Data are presented as means fold over *ad libitum* fed control \pm SEM, $n=4$, $P<0.05$ # vs WT or fed, 2-tailed, unpaired Student's t test. **b**, Immunoblot analysis of cytosolic and nuclear soluble fractions of liver tissue of WT mice treated as in (a) using indicated antibodies. Densitometric analyses of Vps15 or BMAL1 proteins normalized to Tubulin (cytosolic fraction) or H3 (nuclear soluble fraction) levels presented as folds over control fed \pm SEM, $n=4$, $P<0.05$ # vs WT fed. **c**, ChIP-qPCR analyses of BMAL1 and Vps15 enrichment at promoter regions of indicated gene in chromatin of liver tissue of *ad libitum* fed and 24h fasted mice collected at ZT6. Data are presented as means fold enrichment over IgG \pm SEM, $n=3-4$, $P<0.05$ * vs IgG, 2-tailed, unpaired Student's t test.

Altogether, these findings in cells and *in vivo* in mouse liver suggest that nuclear shuttling and chromatin recruitment of Vps15 is a controlled process with the upstream signals yet to be identified. However, the modest changes in nuclear Vps15 and Vps34 in response to the IVM or LMB treatment in cells as well as the low amplitude of their diurnal nuclear expression in the liver could be also explained by constitutively present nuclear pools class 3 PI3K subunits, a hypothesis that we will pursue testing in future works.

2. The authors need to provide the rationale to start their study using the Vps15 subunit KO mice but not the lipid kinase Vps34, as the introduction shows a link between circadian lysosomal activity to class 3 PI3K, which is attributed to the PI-kinase function. Why do they check the regulatory subunit instead of the kinase in the first figure? The work could be improved by comparing to Vps34 KO mice.

Response:

We apologize for the lack of clarity of our previous version of the manuscript, and we thank reviewer for their comment that prompted us to provide further controls and interesting insights into the function of class 3 PI3K *in vivo*.

1) In our previous works (Nemazanyy et al., EMBO MM, 2013; Nemazanyy et al., Nature Comm., 2015 and Iershov et al., Nature Comm., 2019)³⁻⁵, we have thoroughly characterized the deleterious consequences of Vps15 deletion on Vps34 lipid kinase function. Namely, consistent with published reports in different models, the deletion of Vps15 results in the degradation of Vps34 and effectively represents a Vps15 and Vps34 depletion⁵. This is in line with the essential role of Vps15 for Vps34 stability and lipid kinase activity. We have also provided additional data showing Vps34 inhibition (decreased PI3P levels and autophagy defects) in Vps15-depleted MEFs in response to point #3 (see **R-Fig. 6 below**). Our motivation to employ Vps15 deletion as the model of class 3 PI3K inactivation stems from early findings in yeast and in mammalian cells that hypothesized wider roles of Vps15 regulatory subunit beyond its function in a complex with Vps34. To this end, in mammalian cells, immunodepletion studies showed that most of the Vps34 protein resides in the complex with Vps15⁶. However, presence of Vps15 in the cellular compartment lacking Vps34 was suggested by observations in HeLa cells in which a pool of Vps15 was resistant to extraction with Triton X100 while in the same conditions Vps34 was fully extracted⁷. Moreover, Vps15 was hypothesized to have putative protein kinase activity⁸⁻¹⁰. Yet, it lacks a P-loop and has non-typical amino acid residues in critical elements in activation loop (DFA instead of DFG) and catalytic loop (HGD instead of HRD)¹¹. Thus, it is still not certain whether Vps15 is pseudokinase or an active enzyme. Nevertheless, in yeast cells, the substitution of any of three critical for protein kinase activity residues in the aforementioned elements (D148, D165 or E200) was sufficient to prevent phosphorylation of Vps15, suggesting its autophosphorylation^{12,13}. These early findings advocated that Vps15 might have broader cellular roles in addition to its well described function in complex with Vps34. This motivated us for employing Vps15-null cell and liver-specific mutant mice. Our rationale is better presented in the revised manuscript (**line 105-110 in the revised manuscript**). Importantly, we now show that mutation of critical for yeast Vps15 function and conserved in human Vps15 protein residues D148 to R and E200 to R also abrogates the binding of Vps15 with Vps34 in mammalian cells (**R-Fig. 12c below**). This important finding thus allowed us to show that interaction of Vps15 and BMAL1 does not rely on presence of Vps34 (**R-Fig. 5a, corresponding to Fig.4g in the revised manuscript**).

2) Moreover, in line with the suggestion of the reviewer, and in addition to the already employed genetic and pharmacologic inactivation of Vps34 in MEF cells, we also conducted analyses in Vps34 hepatic mutants that for the first time were directly compared to Vps15 iLKO mice. For this, we have generated the TtrCre⁺;Vps34^{fl/fl} mouse mutant for acute Vps34 depletion in hepatocytes employing the same strategy as the depletion of Vps15. Surprisingly, unlike our observations in Vps34-depleted MEFs, the deletion of Vps34 in liver resulted in repression of REV-ERB α protein expression (analyses at peak of its expression in livers collected at ZT8) (**R-Fig. 5b**). Notably, depletion of Vps15 resulted in similar inhibition of lysosomal degradation by autophagy as in livers of Vps34 mutant mice as marked by of p62, LC3 and Lamp1 accumulation (**R-Fig. 5b**). Initially, we were puzzled by this comparable decrease of REV-ERB α protein in livers of Vps15 and Vps34 mutant mice as our findings in Vps34-depleted MEFs, use of structurally unrelated inhibitors of Vps34, rescue experiments with Vps15mut1 (E200R), and luciferase assays with Vps15 point mutant 1 (E200R) that doesn't interact with Vps34 all collectively showed that the function of Vps15 as a co-activator of BMAL1

R-Figure 5. a, Immunoblot analyses of Vps15-containing complexes immunoprecipitated from HEK293T cells transiently transfected with Vps15wt or Vps15mut1. Cells were collected 48 hours post-transfection, Vps15 was immunoprecipitated using anti-Flag antibody. **b**, Immunoblot analysis of total protein liver extracts of four months old *ad libitum* fed Vps15 iLKO, Vps34iLKO and control mice sacrificed at ZT8. Densitometric analyses of indicated proteins normalized to GAPDH levels are presented as graphs. Data are means \pm SEM, n=5, P<0.05 # vs WT, 2-tailed, unpaired Student's t test. **c**, Immunoblot analysis of nuclear soluble fractions of livers of WT, Vps15iLKO and Vps34iLKO mice treated as in (b). Densitometric analyses of Vps15 and Vps34 proteins normalized to Lamin B1 levels presented as folds over WT control. Data are means \pm SEM, n=5, P<0.05 # vs WT, 2-tailed, unpaired Student's t test.

for *Nr1d1* (REV-ERB α) transcription does not rely on Vps34 lipid kinase activity. In search for possible explanations, we reasoned that as in Vps34-depleted MEFs the expression levels of nuclear Vps15 could provide the clue. The analyses of nuclear levels of Vps15 in livers of Vps34 hepatic mutants showed that nuclear Vps15 was well below 30% of levels found in livers of control mice (R-Fig. 5c). This differs to the situation *in vitro*, in MEFs acutely depleted of Vps34, in which nuclear pool of Vps15 is above 60% of levels of control cells (R-Fig. 27, response to point 3 of the Reviewer 3). Altogether, our findings *in vitro* and *in vivo* stress the importance of employing both acute and chronic

depletion of class 3 PI3K to study its functions. These observations also point to likely differences in cytosolic and nuclear pool Vps15 turnover, a topic that we will pursue in future. If the reviewer agrees, we prefer to not include these data in the manuscript as this extra genetic model does not qualitatively provide further evidence to support our claim on Vps15 being a co-activator of BMAL1, in addition to already presented data.

3. What is the impact of PI3P generation upon Vps15 KO? Would impact on PI3P generation lead to a change in the circadian clock?

Response:

We thank the reviewer for prompting us to provide controls that further reinforced the conclusion on a Vps34-lipid kinase independent role of class 3 PI3K in transcriptional co-activation of BMAL1-CLOCK complex.

1) In **R-Fig. 6a** (corresponding to **Extended Data Fig. 2c** in the revised manuscript) we show that PI3P levels are significantly decreased in Vps15-depleted MEFs 5-days post depletion (the same time that we employed for Dexamethasone synchronization in Vps15-depleted MEFs). This is similar to the depletion observed in MEFs treated with Vps34-IN1 for 18hrs with, an inhibitor of Vps34 lipid kinase (**R-Fig. 6b**, corresponding to **Extended Data Fig. 3c** in the revised manuscript). The specificity of the HSR-Cherry probe binding to PI3P was further validated by the use of an HSRmut probe in which binding site for PI3P was mutated and that showed minimal background in this assay. These findings are consistent with our published observations of endocytosis defects and autophagy inhibition in Vps15-depleted cells (see below response to the Reviewer 4, points 8, **R-Fig. 33**).

2) Nuclear Vps34 (**R-Fig. 3**) and detection of PI3P in nucleus of control cells (**R-Fig. 6a, 6b**) backs a plausible implication of PI3P on BMAL1 driven transcription. In line, we observed transcription of *Per2* repressed by two different Vps34 lipid kinase inhibitors in MEFs (**Extended Data Fig. 3b** in the revised manuscript) and co-activating function of Vps34 (in the conditions of its overexpression at high level) on E-box-LUC in HEK293T cells shown in this revised version of the manuscript (**R-Fig. 28**, see the response to point 5 of the Reviewer 3 below). However, acute deletion of

R-Figure 6. a, PI3P detection with a recombinant protein HRS-RFP probe in Vps15-depleted (CRE) and control (GFP) MEFs 5 days post infection with GFP and CRE expressing adenoviruses. The incubation with HRS-mut and the condition without the probe served as controls of non-specific binding of HRS-RFP probe. The quantification of HRS-RFP puncta per cell (\pm SEM, n=50-70 cells, P<0.05 # vs GFP, 2-tailed, unpaired Student's t test) is presented. Scale bar, 10 μ m. **b**, PI3P detection with a recombinant protein HRS-RFP probe in Vps15^{f/f} MEFs treated with Vps34-IN1 (5 μ M) for 18 hours. The incubation with HRS-mut and the condition without the probe served as controls of non-specific binding. Scale bar, 10 μ m.

Vps34 in MEFs does not affect *Nr1d1* (REV-ERBa) transcription and protein levels. Notably, observations in MEFs treated with Vps34 inhibitors, the rescue of REV-ERBa expression in Vps15-null MEFs with Vps15mut1 (E200R) that does not interact with Vps34, and ability of this mutant form of Vps15 to co-activate BMAL1-CLOCK collectively advocate the scenario in which Vps15 does not rely on Vps34 for transcriptional co-activation of BMAL1. We thank the reviewer for prompting us to investigate PI3P levels as it led to an initiation of study on the roles of nuclear PI3P in regulating gene transcription.

4. Figure 1b, what is the expression level of Vps34 upon Vps15 KO in the WB? Also need to probe for Vps15 to validate the KO.

Response:

We apologize for leaving out this control in the initial submission. As seen on R-Fig. 7, chronic depletion of Vps15 results in decreased levels of Vps34 and is accompanied by accumulation of p62 (autophagy inhibition). This analysis is also included as **Extended Data Fig. 1a** in the revised manuscript.

R-Figure 7. Immunoblot analysis of total protein liver extracts of five-week-old *ad libitum* fed Vps15LKO and control mice collected at ZT6 using indicated antibodies. Densitometric analyses of Vps15, Vps34 and p62 protein normalized to GAPDH levels are presented as graphs. Data are presented as means \pm SEM, n=4, P<0.05 # vs WT, 2-tailed, unpaired Student's t test.

5. In the later part of the manuscript, the authors have shown Vps15 interaction with BMAL1-CLOCK complex. Does this interaction provide the mechanism for Vps15 to translocate into the nucleus?

Response:

We thank reviewer for this interesting suggestion. Indeed, nuclear translocation of BMAL1-CLOCK complex is active and is controlled by posttranslational modifications^{14,15}. In addition to being driven by the NLS of BMAL1, it is also positively impacted by RAE1 and NUP98 proteins^{16,17}. To investigate whether BMAL1 depletion impacts Vps15 and Vps34 nuclear localization, we knocked down BMAL1 using shRNA. As seen in R-Fig. 8, acute BMAL1 knock-down resulted in decreased CLOCK levels but did not significantly impact nuclear Vps15 or Vps34 levels in MEFs. Thus, BMAL1-CLOCK does not act upstream of Vps15 or Vps34 nuclear localization.

R-Figure 8. Immunoblot analysis of cytosolic and nuclear soluble protein extracts using indicated antibodies in control (GFP) and BMAL1-depleted (shRNA BMAL1) MEFs. Densitometric analyses of BMAL1, CLOCK, Vps15 and Vps34 proteins normalized to Tubulin (cytosolic fraction) and Lamin B1 (nuclear soluble fraction) levels presented as folds over GFP treated cells. Data are presented as means \pm SEM, n=4 independent experiments, # vs GFP, 2-tailed, unpaired Student's t test.

6. The authors need to provide IF staining of PI3P, Vps34, Vps15, BMAL1 in the wild-type cells and Vps15 KO cells with nuclear staining to demonstrate their subcellular localization, especially nuclear localization upon Vps15 KO. Their model shows subcellular translocation of Vps15 and nuclear complex of it with BMAL1, but they have only done the subcellular fractionation and performed WB, where imaging data would be essential for supporting those claims but are largely missing.

Response:

We thank the reviewer for raising these important points that further reinforce the conclusions of our study.

1) As requested by the reviewer, we also analyzed nuclear localization of BMAL1 in control and Vps15-depleted MEFs. As seen on **R-Fig. 9** (corresponding to **Extended Data Fig. 2h** in the revised manuscript), nuclear localization of BMAL1 is not impacted by acute depletion of Vps15 in MEFs. This is consistent with our findings of BMAL1 enrichment unaffected on promoters of its target genes; *Dbp1* and *Nr1d1* in ChIP-qPCR analyses (**Fig. 4b** in the revised manuscript).

R-Figure 9. Immunofluorescent microscopy analyses of BMAL1 in Vps15-depleted (CRE) and control (GFP) MEFs 5 days post deletion. The quantification of the BMAL1-positive signal in nucleus and in cytosol of GFP and CRE infected cell is presented. Data are \pm SEM, n=180-270 cells. Scale bar, 10 μ m.

2) We also provide staining of PI3P in Vps15-null MEFs as an additional control (**R-Fig. 6a**, above). As seen on **R-Fig. 6a**, Vps15 depletion results in decreased PI3P positive structures both in cytosol and in the nucleus.

3) We also pursued optimizing the immunofluorescence microscopy analyses to demonstrate the subcellular localization of Vps15 and Vps34. First, we tested three available antibodies against Vps15 and one antibody against Vps34, which we validated are specific in immunoblot analyses and immunoprecipitation. As could be seen from **R-Fig. 10a**, all antibodies against Vps15 are specific to Vps15 and show uniform cytosolic-nuclear staining. Unfortunately, commercially available antibodies against Vps34 did not provide reliable staining under different fixation and cell permeabilization protocols (*data not shown*), thus, we were unable to investigate sub-cellular distribution of endogenous Vps34 by immunofluorescence. We expect that, in line with the immunoblot analyses of subcellular fractions of cells and liver tissue of Vps15 mutants, Vps34 protein levels are decreased both in cytosol and nucleus of Vps15-depleted MEFs. Given the specificity of anti-Vps15 antibodies, we now provide the microscopic evidence of nuclear pool of endogenous Vps15 in MEF cells. Immunoblotting analyses of nuclear vs cytosolic fractions suggested that nuclear Vps15 represents a less pronounced pool compared to Vps15 in the cytosol. To better visualize nuclear Vps15 in MEFs,

R-Figure 10. a, Immunofluorescent microscopy analyses of Vps15 in MEF using three different anti-Vps15 antibodies (scale bar, 10µm). To validate the specificity of anti-Vps15 antibodies, MEF cells were transduced with the adenoviral particles expressing shRNA Vps15 and analyses were performed 72 hours post infection. The efficiency of depletion and the specificity of anti-Vps15 antibodies was validated by immunoblot with indicated antibodies. **b** and **c** Immunofluorescent microscopy analyses of Vps15 using two antibodies raised in different hosts (ab1 (mouse monoclonal) and ab2 (rabbit polyclonal)) in MEF cells treated for indicated times with CSK buffer. The staining with DAPI and anti-Histone 3 antibody served as nuclear staining control. Scale bar, 10µm. **d**, Immunofluorescent microscopy analyses of endogenous Vps15 and BMAL1 in MEF cells. Scale bar, 5µm.

we treated cells with CSK buffer before fixation; that allows to wash out the cytosolic soluble proteins¹⁷. Moreover, depending on time of incubation, CSK buffer treatment facilitates the visualization of tightly bound nuclear proteins. As seen from **R-Fig. 10b**, 1 min of incubation with CSK was sufficient to reveal prominent nuclear staining of endogenous Vps15. Notably, the signal weakens with longer (3 min) incubation. This is consistent with the wash-out of non-cytoskeletal proteins, an expected outcome of CSK buffer treatment. These findings were validated with two different anti-Vps15 antibodies (mouse monoclonal (Ab1) and rabbit polyclonal (Ab2)) further reassuring the observation of nuclear Vps15 (**R-Fig. 10b, 10c** corresponding to **Fig. 3b** in the revised manuscript). As suggested by the reviewer, we also co-stained endogenous Vps15 and BMAL1 in MEF cells which showed a partial nuclear co-localization as puncta (**R-Fig. 10d** corresponding to **Extended Data Fig. 6d** in the revised manuscript).

4) Nuclear Vps15 puncta staining prompted us to investigate further involvement of class 3 PI3K with RNA Pol2-driven nuclear transcription. For this, we labelled *de novo* transcription sites with BrUTP in MEF cells with ectopically expressed Flag-tagged Vps15 and Vps34. Strikingly, both Vps15 and Vps34 co-localize with BrUTP puncta (**R-Fig. 11a** corresponding to **Fig. 3c** in the revised manuscript). Next, we asked if Vps15 co-localization with *de novo*

transcription sites extends to endogenous Vps15 in other cell type such as hepatocytes. As presented on **R-Fig. 11b** (corresponding to **Fig. 3e** in the revised manuscript), endogenous Vps15 readily detected in nuclei and it co-localized with *de novo* transcription sites labelled with BrUTP and RNA Pol2 phospho-Ser5 (transcriptional elongation form). Notably, the triple positive BrUTP/RNA Pol2/Vps15 sites were detected in hepatocyte nuclei outside of nucleoli (**R-Fig. 11c** corresponding to **Fig. 3e** in the revised manuscript). Unfortunately, we could not perform these analyses with Vps34 as we lack antibodies that would specifically recognize endogenous Vps34 in immunofluorescence microscopy.

R-Figure 11. a, Transcription assay with ectopically expressed Vps15-Flag and Vps34-Flag proteins in MEFs cells. 24hrs post-transfection MEFs were incubated with BrUTP to label *de novo* transcription sites. For co-localization, immunofluorescent microscopy was performed with anti-Flag and anti-BrUTP antibodies. Scale bar 5 μ m. **b**, Immunofluorescent microscopy analyses of co-localization of endogenous RNA Pol2 phSer5, Vps15 and *de novo* transcription sites labelled with BrUTP in primary hepatocytes isolated from 2-month old control wild type mice. Scale bar 10 μ m and 5 μ m (inset).

5) Finally, we asked whether Vps15 and Vps34 interact with RNA Pol2. Initially, we detected specific binding of ectopically expressed Vps15 and Vps34 with endogenous RNA Pol2 in HEK293T cells (**R-Fig. 12a**, corresponding to **Fig. 3d** in the revised manuscript). Next, we extended this observation to endogenous Vps15 and RNA Pol2 in MEFs (**R-Fig. 12b**, corresponding to **Fig. 3f** in the revised manuscript). To get further insight into the requirement of Vps34 for interaction between Vps15 and RNA Pol2, we also employed two different point mutants of Vps15 (E200R and D165R) that do not form the complex with Vps34. Notably, the mutations that disrupted the interaction between Vps15 and Vps34 did not affect binding of RNA Pol2 or its Ser5 phosphorylated active form (**R-Fig. 12c**, corresponding to **Fig. 3g** in the revised manuscript). This observation is in favor of the scenario that interaction of Vps15 and RNA Pol2 does not rely on Vps34.

Altogether, these analyses further reinforce our initial findings of a nuclear pool both Vps15 and Vps34 as well as provide new evidence of Vps15 interaction with RNA Pol2 and their co-localization at *de novo* transcription sites.

R-Figure 12. **a**, Immunoblot analyses of Flag-containing immunoprecipitates from nuclear soluble fraction of HEK293T cells transfected with Vps15-Flag and Vps34-Flag expressing vectors. Co-immunoprecipitation of RNA Pol2 with ectopically expressed Vps15 and Vps34 proteins is evidenced in immunoblot. ev, empty vector transfected HEK293T condition used as a control of non-specific binding. **b**, Immunoblot analyses of Vps15-containing immunoprecipitates using anti-RNA Pol2 antibody from nuclear soluble fractions of MEF cells. **c**, Immunoblot analyses of Vps15-containing complexes immunoprecipitated from HEK293T cells transiently transfected with constructs of Vps15wt, Vps15mut 1 and Vps15mut 2 (ev, empty vector). Cells were collected 48 hours post-transfection, Vps15 was immunoprecipitated using anti-Flag antibody.

7. Figure 3b is at poor quality, no scale bar, only 2~3 cells are shown and without quantification. Also, is there PLA between endogenous Vps15 and BMAL1?

Response:

We apologize for these shortcomings of presentation, and we thank this reviewer and the others for their suggestions that further improved the presentation of PLA analyses.

1) We repeated the PLA assay between endogenous Vps15 and Flag-BMAL1 and provided the quantification of puncta corresponding to Vps15-BMAL1 proximity for this new analysis (**R-Fig. 13a**, corresponding to **Fig. 4e** in the revised manuscript). This new analysis reproduced our initial findings showing proximity between Vps15 and ectopically expressed BMAL1 with signal localized predominantly in the nucleus and a minority of puncta present in cytosol. As also proposed by **the Reviewer 3**, we have modified the labelling to make the color code more readable.

2) As suggested by the reviewer, we have also included the PLA analyses with endogenous Vps15 and BMAL1. These new analyses confirmed the observation of overexpressed BMAL1-Flag, namely the presence of both nuclear and cytosolic puncta (**R-Fig. 13b**, corresponding to **Extended Data Fig. 6c** in the revised manuscript). To note, although we have observed a significant enrichment of proximity signal in this assay compared to no-antibodies control, as expected, the number of puncta was less substantial compared to ectopically expressed BMAL1-Flag.

Thus, these observations together with new findings showing co-immunoprecipitation of BMAL1-Vps15 domains and luciferase assays further reinforce our findings of Vps15 functionally interacting with BMAL1.

R-Figure 13. a, Proximity ligation assay of endogenous Vps15 and ectopic Flag-BMAL1 in HEK293T cells (co-transfected with GFP-expressing vector for visualization of transfected cells). The no antibody condition served as a non-specific signal control. Scale bar 10 μ m. Quantification of proximity puncta per cell in n=20-520 cells is presented. P<0.05 #: vs no antibody, 2-tailed, unpaired Student's t test. **b**, Proximity ligation assay between endogenous Vps15 and Vps34 proteins in HEK293T cells. The no antibody condition served as a control of non-specific signal. Scale bar 10 μ m. The quantification of the proximity puncta per cell in n=20-36 cells is presented as graph. P<0.05 #: vs no antibody, 2-tailed, unpaired Student's t test.

8. The key finding of the authors in this manuscript is the co-transcriptional activator function of the VSP15 with BMAL1-CLOCK complex in the transcriptional regulation of the clock-controlled genes/regulators. However, the Vps15 knockout seems to have noticeable impact in the expression level of BMAL1, especially in the liver and authors seem ignoring this important point based on results in MEFs. In this context, VPS15 seems to have role in the stability of BMAL1 and BMAL1-CLOCK complex impacting on transcriptional co-activation of BMAL1-CLOCK complex target genes/regulators.

Response:

We thank the reviewer for this pertinent comment that prompted us to further investigate the status of BMAL1/CLOCK's transcriptional complex under conditions of Vps15 loss-of-function and gain-of-function.

1) In this revised manuscript, we show that nuclear BMAL1 and CLOCK proteins have a similar half-life of 10.4hrs, revealed in analyses of protein turnover in cycloheximide treated control and Vps15-null MEFs (**R-Fig. 14a**, corresponding to **Extended Data Fig. 2i** in the revised manuscript).

2) We also studied the BMAL1-CLOCK complex in cells depleted Vps15 or under its overexpression. We now show that neither Vps15 depletion (**R-Fig. 14b**, corresponding to **Fig. 4a** in the revised manuscript) nor its dose-course overexpression (**R-Fig. 14c**, corresponding to **Fig. 4c** in the revised manuscript) impact BMAL1-CLOCK complex in MEF and HEK293T cells. To note, dose course overexpression of Vps15 resulted in its respective increased interaction with the BMAL1-CLOCK complex (**R-Fig. 14c**, corresponding to **Fig. 4c** in the revised manuscript). This is also correlated with increased BMAL1-CLOCK transcriptional activity measured in luciferase assay with E-box-LUC reporter; which we provide in this revised version in response to the comment of the **Reviewer 2** (**R-Fig. 20a**, corresponding to **Fig. 5a** in the revised manuscript).

3) Finally, the ChIP-qPCR analyses in control compared to Vps15-overexpressing cells showed no significant difference in BMAL1 or RNA Pol2 phospho-Ser5 occupancy on promoter regions of *Nr1d1* and *Dbp1* genes (**R-Fig. 14d**, corresponding to **Extended Data Fig. 6a** in the revised manuscript).

R-Figure 14. a, Immunoblot analysis of nuclear soluble fractions of Vps15-depleted (CRE) or control (GFP) MEFs treated with cycloheximide for indicated times. Densitometric analyses of BMAL1 and CLOCK proteins normalized to total protein (TGX gels) are presented as fold over a vehicle treated condition (n=3). **b**, Immunoblot analyses of BMAL1-containing immunoprecipitates using anti-CLOCK and anti-BMAL1 antibody from total protein extract of Vps15-depleted (CRE) or control (GFP) MEFs. Quantification of CLOCK protein in BMAL1 immunoprecipitates normalized to BMAL1 levels in direct IP is presented. Data are means \pm SEM (n = 3). **c**, Immunoblot analyses of BMAL1-containing immunoprecipitates from total protein extract of HEK293T cells transfected with increasing amounts of Vps15wt-expressing vector. The quantification of CLOCK protein in BMAL1 immunoprecipitates normalized to BMAL1 is presented as fold difference to control empty vector transfected cells. Data are means \pm SEM (n = 2-3). **d**, BMAL1 and RNA Pol2 phSer5 recruitment to promoter of indicated genes by ChIP-qPCR in MEFs transduced with adenoviral vectors expressing Vps15wt or GFP as a control. Cells were Dexamethasone-synchronized and collected 24 hours after synchronization. Data are presented as log₂ fold change enrichment over IgG \pm SEM, n=8, P<0.05 # vs IgG, 2-tailed, unpaired Student's t test.

Altogether, these novel findings go in line with our initial conclusions that Vps15 acts as a transcriptional co-activator of BMAL1 without impacting its protein turnover or interaction with CLOCK. These findings are also coherent with the observations that we reported in the initial version of the manuscript using models of acute Vps15 depletion both *in vivo* (TtrCre⁺;Vps15^{fl/fl} mice, Vps15iLKO) and *in vitro* (Vps15^{fl/fl} MEFs infected with Cre or GFP as a control). We demonstrated that unlike upon chronic depletion in mouse liver of Vps15LKO mice (AlbCre⁺;Vps15^{fl/fl} mice), the levels of nuclear BMAL1 upon acute inactivation of Vps15 were not changed (**Extended Data Fig. 2g, Extended Data Fig. 9c in the revised manuscript**). This was also accompanied with no changes of BMAL1 chromatin recruitment in MEF cells and in liver upon acute Vps15 deletion (**Fig. 4b, Extended Data Fig. 9h in the revised manuscript**).

Reviewer #2

A. Summary of the key results

Here, the authors show that targeting the subunits of the Class 3 PI3K influences the activity of CLOCK-BMAL1 in mouse liver, including classic clock-controlled genes like *Per2*, *Dbp*, and *Nr1d1*, as well as those involved in purine biosynthesis. There is quite a lot of data here, from genome-wide studies of transcript abundance and chromatin occupancy to cellular biochemical studies looking at protein abundance, localization, interactions, and transcriptional activation. Overall, the major conclusion is that the regulatory subunit VPS15 binds to BMAL1 to coactivate transcriptional activation in the nucleus independent of the catalytic subunit VPS34. This conclusion seems largely supported by the data, although most of it is indirect (see below). In addition, some issues with the rigor of experimental design and discrepancies between the phenotypes (in constitutive liver knockout (LKO) of the regulatory subunit VPS15 in mice vs. the Cre-inducible deletion of VPS15 in MEFs or in the liver vs. chemical inhibition or Cre-inducible deletion of the catalytic subunit VPS34) confound a clear and compelling mechanistic interpretation here.

B. Originality and significance: if not novel, please include reference

To my knowledge, this work is the first to look at the effects of Class 3 PI3K subunits on CLOCK-BMAL1 activity. If the major conclusion was better supported by the data, it could provide a nice basis to understand the clock-dependent regulation of purine biosynthesis first discovered by Fustin and colleagues (ref. 43). Other recent work shows that CLOCK-BMAL1 activity is regulated directly by 'unexpected' proteins like the RelA subunit of NF- κ B (PMID 34807912), so this could add to a growing body of work on regulatory mechanisms of this central clock transcription factor.

We would like to thank the reviewer for finding our discovery of non-canonical functions of Vps15 as co-activator of BMAL1 transcription factor original, conclusions in large supported with the presented evidence and for great suggestions that we fully addressed in this revised version of the manuscript.

C. Data & methodology: validity of approach, quality of data, quality of presentation

1. The 6-hr sampling frequency is not ideal for studying circadian rhythms. According to the Guidelines for Genome-Scale Analysis of Circadian Rhythms (PMID 29098954), sampling should ideally occur every 2 hours (or every 4 hr at a minimum).

Response:

We fully agree with the reviewer that resolution of sample collection is a key factor for rhythmicity determination. In this revised manuscript, following the advice of the reviewer, we provide new data in line with the methodological recommendations:

1) **R-Fig. 15:** Transcript levels of clock components in livers of control and Vps15iLKO mice collected every 4 hours. JTK-CYCLE analyses of these data show that while rhythmic in livers of control mice, the deletion of Vps15 results in loss of rhythmicity of *Per2*, *Cry1*, *Clock*, *Nr1d2*, and *Ppat* (**R-Fig. 15**, corresponding to **Extended Data Fig. 9g** in the

R-Figure 15. Relative transcript levels of indicated genes in livers of three months old *ad libitum* fed Vps15 iLKO and control mice. The livers of mice were collected every 4 hours in indicated ZT times under 12h light-dark regiment. Data are presented as log₂ normalized fold change as loess best fit curve (n=3-10). Rhythmicity pattern was tested using JTK_CYCLE (**Supplementary Table 1**).

revised manuscript). While still rhythmic in livers of Vps15iLKO mice, the amplitude was decreased for *Nr1d1* and *Dbp* genes (R-Fig. 15, corresponding to Extended Data Fig. 9g in the revised manuscript).

2) R-Fig. 16 and R-Fig. 17: Protein levels of clock components in total (R-Fig. 16) and nuclear extracts (R-Fig. 17) of liver tissue of control and Vps15iLKO mice collected every 4 hours. First of all, we validated Vps15 deletion in this new cohort of mice. It was accompanied by a decrease in Vps34 protein level and a block of autophagy witnessed by accumulation of p62 and LC3 proteins (substrates of autophagy) (R-Fig. 16a, corresponding to Extended Data Fig. 9a in the revised manuscript). Second, the immunoblot analyses in the total liver extracts of Vps15iLKO mice showed that deletion of Vps15 largely inhibited rhythmic expression of REV-ERB α and BMAL1 proteins (R-Fig. 16b, corresponding to Extended Data Fig. 9b in the revised manuscript).

R-Figure 16. a,b Immunoblot analysis of total protein liver extracts of three months old ad libitum fed Vps15 iLKO and control mice using indicated antibodies. The livers of mice were collected every 4 hours in indicated ZT times under 12h light-dark regimen. Densitometric analyses of REV-ERB α , BMAL1 and CLOCK protein normalized to GAPDH levels is presented as loess best fit curve (n=3-4). Rhythmicity pattern was tested using JTK_CYCLE (Supplementary Table 1).

Moreover, we also analyzed the levels of BMAL1, CLOCK and REV-ERB α in the extracts of nuclear soluble fraction (R-Fig. 17, corresponding to Extended Data Fig. 9c in the revised manuscript). As expected, the pattern of expression of REV-ERB α protein in nuclear fractions was similar to that of total protein extracts (R-Fig.17a, corresponding to Extended Data Fig. 9c in the revised manuscript). Notably, we did not observe a significant difference between genotypes in nuclear protein levels of BMAL1 and CLOCK (R-Fig.17a, corresponding to Extended Data Fig. 9c in the revised manuscript). The analyses of other clock components, including ROR α , Per1, Per2, Cry1, Cry2 did not reveal major differences between genotypes (R-Fig.17b). We have also observed more pronounced animal-to-animal variability in this new dataset of control and Vps15iLKO mice compared to the analyses presented in the initial version of the manuscript. Likely, it stems from pooling samples of three separate mouse cohorts that were independently treated with tamoxifen and sacrificed, as a high number of animals was required for these expression analyses.

Finally, with regards to the data that were presented in the initial version of the manuscript, we would like to highlight three considerations for the experimental design of time series that we employed:

1) In cells, we conducted expression analyses after Dex-synchronization in Vps15-null MEFs, Vps34-null MEFs and when using Vps34 inhibitor treatment on samples collected every 4 hours (*now Fig. 1g, 2a, 2d, 4h in the revised manuscript*). Notably, these allowed us to observe defect of BMAL1 transcriptional activity measured by decreased *Nr1d1* and *Dbp1* transcript levels in Vps15-depleted MEFs. These were further complemented by *in vivo* bioluminescence analyses using *Nr1d1*-Luc reporter in freely moving mice¹⁸ that we conducted in our Vps15iLKO mouse model (*Fig. 6i and Extended Data Fig. 9e in the revised manuscript*).

2) We performed metabolomics analyses on samples collected every 6 hours and we observed in control mice, a set of rhythmic metabolites including nucleotides, amino acids and fatty acids, comparable to previously published dataset¹⁹ (Koronowski et al., *Cell*, 2019). Similarly, in recent work of Hepler et al., *Science*, 2022²⁰ a set-up of 6hr-collection was employed for the metabolomics analyses to discover rhythmic creatine-mediated thermogenesis downstream of BMAL1. Thus, we believe this approach represents a compromise between number of samples and cost of the analyses for selection of the most impacted hits for further validation studies.

3) Finally, as we hypothesized that Vps15 and Vps34 transcript and nuclear proteins have shallow amplitudes, we conducted their expression analyses in livers of control mice collected every 4 hours (*Extended Data Fig. 5b, c, e in the revised manuscript*).

Altogether, the new *in vivo* analyses and those already presented in initial version analyses in cells and mice support the central finding of our work that Vps15 depletion results in inhibition of BMAL1-CLOCK-driven transcription.

2. The immunohistological staining of BMAL1 in mouse liver presented in Ext. Data Fig. 1f shows an apparently high level of non-nuclear protein staining, but this does not seem to match the western blot data from panel d of the same figure, where cytosolic levels of BMAL1 are significantly reduced in VPS15LKO livers. Can the authors explain this discrepancy?

Response:

We thank the reviewer for bringing our attention to this methodological point that we clarified in this revised version of the manuscript. First, due to differences in the antigen preparation and antigen retrieval, the immunohistochemistry analyses and immunoblot analyses of cytosolic fraction of liver tissue are not directly comparable. For immunohistological analyses, liver tissue is fixed for 18 hours in 4% formalin-buffered solution before paraffin inclusion. For immunoblot analyses of the cytosolic fraction, soluble proteins are extracted in hypotonic buffer without detergent. In support, the differences in BMAL1 protein levels are much less apparent between control and Vps15LKO mice when total liver extracts prepared with 1%NP40 detergent for immunoblotting (compare **Fig. 1b** and **Extended Data Fig. 1e** in the revised manuscript). Moreover, antigen recognition in histological tissue samples depends on antigen retrieval, on tissue structure (e.g. presence of the vacuoles and protein aggregates in samples of Vps15LKO mice), on the freshness of the histological cuts of tissue as well as on level of dehydration of tissue. These make the immunohistochemistry with DAB chromophore reaction a qualitative method adapted for evaluating protein sub-cellular distribution (signal intensity within the sample).

In addition to these clarifications, we also present new immunohistological analyses (performed on slides of freshly cut liver blocks) in livers of Vps15 LKO mice (**R-Fig. 18a**, corresponding to **Extended Data Fig. 1g** in the revised manuscript). These new analyses are in line with previous observations that show less marked nuclear BMAL1 localization in livers of Vps15LKO mice compared to livers of control mice (ZT0). Moreover, we complement these observations with immunofluorescent analyses of BMAL1 (**R-Fig.18b**, corresponding to **Extended Data Fig. 1h** in the revised manuscript). This avoids the shortcomings of signal amplification and enzyme linked chemical substrate DAB revelation of the immunohistochemistry analyses. Finally, we also complement these findings in conditions of chronic Vps15 deletion with the histological analyses in livers of Vps15iLKO mice (acute model of Vps15 depletion in hepatocytes) (**R-Fig.18c**, corresponding to **Extended Data Fig. 9d** in the revised manuscript). These show no major differences in BMAL1 distribution between two genotypes in accordance with immunoblot analyses of nuclear fractions (**R-Fig.18c** and **R-Fig.17a**).

R-Figure 18. Immunohistochemistry (a) and immunofluorescent (b) microscopy analyses of BMAL1 in liver section of Vps15 LKO and control mice collected at ZT0. Scale bar 20 μ m and 10 μ m (inset). c, Immunohistochemistry analyses of BMAL1 in liver section of Vps15 iLKO and control mice collected at ZT0. Scale bar 20 μ m and 10 μ m (inset).

3. *The different effects on BMAL1 expression and chromatin binding in VPS15LKO mice and the acutely driven deletion by Cre in MEFs make it hard to conceptually integrate the whole animal and cell-based data, diminishing confidence in the conclusions.* Inhibition of lipid kinase activity with chemical inhibitors altered gene expression but with gene-specific effects that varied quite widely (see below). By contrast, **depletion of the VPS34 protein by Cre had minimal effect on the expression of clock-controlled genes, and it also reduced VPS15 levels, although apparently not enough to see the anticipated VPS15-dependent effects. Neither Cre-treated line reduced BMAL1 binding in MEFs as assessed by ChIP, although it was significantly decreased in the VPS15LKO livers. Therefore, it's unclear how the comparison of transcripts in WT and VPS15LKO livers (Fig. 3 and 4) adds to the clarity of the conclusions here. Data from the Cre-inducible deletion of VPS15 in the liver are compelling, as are data in primary hepatocytes, but there are enough confounding results here that were not clarified or explained amongst the different experimental conditions that the strength and clarity of the conclusions are diminished.**

We agree with the reviewer that the molecular interaction between class 3 PI3K and the molecular clock are complex and likely occur at several levels. This complexity stems from, first, multiple functions of class 3 PI3K in autophagy, endocytic trafficking and lysosomal activity and, second, from molecular complexity of molecular clock with interconnected transcriptional-translational loops functioning across 24hr. Our main finding is that nuclear Vps15 acts as a co-activator of BMAL1-CLOCK transcription factor complex. We show that this function of Vps15 does not rely on either lipid kinase activity of Vps34 (use of Vps34 lipid kinase inhibitors and Vps34-null cells) or on the interaction of Vps15 with Vps34 (use of Vps15 point mutant that does not interact with Vps34). Thus, we report a novel transcriptional co-activator of the circadian clock and unknown complexity of class 3 PI3K that so far was considered as an obligate complex of Vps15 with Vps34 lipid kinase for PI3P production. In this revised manuscript we have added a wealth of new data that support these conclusions with clarified explanations.

Specifically, we would like to stress following three points:

1) Initially, we hypothesized functional interaction between class 3 PI3K and BMAL1-CLOCK through transcriptional and ChIP-Seq analyses in livers of Vps15LKO mice. These suggested an inhibition of BMAL1-CLOCK. While Vp15LKO mouse model is a powerful tool, as rightly pointed by the reviewer, such chronic Vps15 inactivation and, as a result, prolonged class 3 PI3K dysfunction leads to profound endocytic trafficking and autophagy defects. The latter makes it challenging to dissociate from the former's primary effect on clock function. For this reason, we employed mouse and cell models for acute inactivation of Vps34 and Vps15. We also used the point mutant of Vps15 that does not interact with Vps34 as well as Vps34 lipid kinase inhibitors. Importantly, decreased nuclear expression of BMAL1 and CLOCK proteins and lower chromatin recruitment of BMAL1 (ChIP) is evident only in the chronic model of Vps15 inactivation (Vps15LKO mice), but not in the models of pharmacologic inhibition of Vps34, or acute inactivation of Vps15 or Vps34 in MEFs (compare **Fig. 1e and Fig. 1b, 2c, 2f, Extended Data Fig. 9h in the revised manuscript**). In this revised manuscript we also show that acute depletion of Vps15 does not affect nuclear BMAL1 protein turnover or BMAL1-CLOCK complex formation (**R-Fig. 14a, 14b, response to the Reviewer 1, point 8**). We decided to keep presenting the data with the chronic model of Vps15 inactivation as it illustrates the complexity and is insightful for the elucidation of long-term effects of class 3 PI3K dysfunction on metabolic rhythmicity.

2) As pointed by **this Reviewer and the Reviewer 3**, the difference in nuclear levels of Vps15 and Vps34 in Vps15-null and Vps34-null MEFs represents another unexpected finding of our work. We investigated further nuclear vs cytosolic levels of Vps15 in models of chronic and in acute depletion of Vps34. As our comparative analyses in Vps15iLKO and Vps34iLKO mice revealed, REV-ERB α protein expression is similarly decreased in both genotypes (**R-**

Fig. 5b, response to **the Reviewer 1**, point 2). This is accompanied by a similar 80% decrease in nuclear Vps15 protein levels in livers of Vps34iLKO and Vps15iLKO mice (**R-Fig. 5c**). This is not the case in MEFs acutely depleted of Vps34 that show about 40% decrease in nuclear Vps15 protein levels compared to the levels in control cells (**R-Fig. 27**, response to **the Reviewer 3**, point2). One possible interpretation of these findings could be a threshold for nuclear Vps15 that needs to be maintained for its nuclear functions and for control of the nuclear pool of class 3 PI3K subunits. We trust this question deserves an in-depth study and is beyond the scope of the current work.

3) Finally, we apologize for the lack of clarity in explaining differential effects of clock target gene expression in the experiments with structurally unrelated inhibitors of Vps34 lipid kinase. This shortcoming is addressed in the revised manuscript. Specifically, we have modified the text to improve clarity as well as added the data on *Nr1d1* transcript, REV-ERB α protein expression and *Nr1d1*-LUC activity using three different Vps34 lipid kinase inhibitors. The detailed response to this question is in **point 4** below.

Altogether, our findings presented in the revised manuscript are in favor of the hypothesis that Vps15 acts as a co-activator of the BMAL1-CLOCK transcription complex. In this work, we showcased the importance of using complementary tools for dissecting interactions between class 3 PI3K and the molecular clock. This included using chronic and acute gene inactivation *in vivo* and in cells combined with pharmacologic inhibition using structurally unrelated compounds.

4. Variable effects on the abundance of clock-controlled genes with Class 3 PI3K inhibitors do not rigorously support the conclusion/header of that paragraph as stated, that there is a "lipid kinase independent role of class 3 PI3K in the clock control." The range of effects on the several transcript levels range from decreased to increased abundance, or no effect at all, suggesting that something more complicated is going on.

Response:

We apologize for not being clear in our explanations for this important point (also highlighted by **the Reviewer 3**). First of all, we employed selective inhibitors of Vps34 as they represent a useful tool for acute inhibition of lipid kinase activity of Vps34 without impacting its protein levels. We agree with the reviewer that at first sight the effects of Vps34-inhibitors on the clock function are puzzling. Yet, these differences are likely due to general shortcomings when using inhibitors such as inhibition of "off-targets". Our observations highlight the importance of employing structurally different drugs in the studies which likely have non-overlapping off-targets. In this scenario, when using structurally unrelated compounds that effectively inhibit its intended target (e.g. lipid kinase Vps34), only common observations could be attributed to their intended target inhibition. At the same time the inconsistencies could be due to their action on other proteins.

In this revised manuscript, in addition to SAR405 and PIKIII, we utilized Vps34-IN1, another efficient inhibitor of Vps34 (IC₅₀ 4nM). Treatment with Vps34-IN1 inhibited Vps34, as witnessed by decrease in PI3P levels and p62 protein accumulation (**R-Fig. 19a, b**, corresponding to **Extended Data Fig. 3c, d** in the revised manuscript). At the same time, it had no effect on REV-ERB α protein levels (**R-Fig.19b**). Surprisingly, treatment of MEFs with Vps34-IN1 led to an increase in the transcript levels of *Nr1d1* and *Cry1* while levels of *Per2* decreased, and no effect was observed for *Dbp* transcript (**R-Fig.19c**).

Finally, in addition to presented expression studies, in the revised manuscript, we also show a rhythm analysis using bioluminescence of *Nr1d1*-Luc reporter in cells treated with three different Vps34 inhibitors (**R-Fig.25** response to **the Reviewer 3**, point 1).

R-Figure 19. a, PI3P detection with a recombinant protein HRS-RFP probe in Vps15^{fl/fl} MEFs treated with Vps34-IN1 (5 μ M) for 18 hours. The incubation with HRS-mut and the condition without the probe served as controls of non-specific binding. Scale bar, 10 μ m. Protein (n=3) (**b**) and relative transcript levels (n=9) (**c**) in synchronized with Dexamethasone MEFs treated with DMSO (control) or Vps34-IN1. Densitometry analyses of REV-ERB α protein normalized to Actin levels presented as fold over DMSO treated cells. Data are presented as means \pm SEM, P<0.05 # vs DMSO, 2-tailed, unpaired Student's t test.

Importantly, the three compounds that we employed are efficient in inhibiting lipid kinase activity of Vps34. As reported in Ronan *et al.*²¹ SAR405 has an IC₅₀ of 1 nM in the phosphorylation of a PtdIns substrate by human recombinant Vps34 enzyme while showing low inhibition up to 10 μ M of class I and class II PI3Ks as well as of mTOR. Yet, as could be seen in the Supplementary dataset, while largely specific to Vps34 lipid kinase, 1 μ M of SAR405 inhibited 50-75% of SMG1 kinase and above 60% of PIK3Cb and PIK3CD. As for PIKIII, Dowdle *et al.*²² tested the selectivity of this inhibitor on 44 Kinases and found it to be 100 folds more selective on Vps34 than other kinases. However, as presented in the Supplementary Table 1 of the original report, it inhibits other kinases at the following IC₅₀ concentrations: CDK2A (2.6 μ M), GSK3B (4.65 μ M), AURKA (3.9 μ M), ABL1 (8.8 μ M), EphB4 (9.4 μ M), FLT3 (7.2 μ M), JAK2 (8.7 μ M), KDR (9.1 μ M), MAPK1 (6.8 μ M). For Vps34-IN1, Bago *et al.*²³ described the high selectivity of this inhibitor for Vps34. However, we can see in Figure 2B and 2C in the original report, that Vps34-IN1 inhibits other kinases at the IC₅₀ concentration of: PI3Kd (1.89 μ M), PI3Kg (2.685 μ M), PIP5K1C (0.382 or 3.37 μ M), PIP5K1A (4.93 μ M), PI4K3B (1.069 μ M). Our non-exhaustive bibliography search showed that these potential off-targets of different Vps34 lipid kinase inhibitors were also reported to have functional links with the clock: CDK2a²⁴, GSK3b²⁵, ABL1²⁶, MAPK1²⁷, SMG1²⁸.

In sum, our findings point to the scenario that transcription of certain circadian clock genes such as *Per2* is inhibited by Vps34 lipid kinase activity while transcription of *Nr1dr1* relies on Vps15 presence. The future work will address the mechanisms of Vps34 lipid kinase in the nuclear gene transcription. We have modified the text of the revised manuscript to clarify this important point (**line 201-214 in the revised manuscript**).

5. *The statement that VPS15 protein displays diurnal rhythmicity based on an approximate change of ±10-20% in protein abundance (Ext. Data Fig. 4) should be modified to say "...showed a modest diurnal rhythmicity..."*

Response:

We have modified the text as suggested by the reviewer (**line 240 in the revised manuscript**).

D. Appropriate use of statistics and treatment of uncertainties

Seems appropriate.

E. Conclusions: robustness, validity, reliability

Please see the points made in sections C and F. The presence of key data needed to support the conclusions in both main and extended figures made it hard to read the manuscript, as it required constantly flipping back and forth to look at both sets of figures to critically evaluate the authors' conclusions. This could be improved by better streamlining data presentation in the manuscript.

Response:

We apologize for these shortcomings of data organization in the initial version of the manuscript. We have made an extensive effort to re-organize the manuscript text and corresponding figures to improve the comprehension and fluidity of the story. The modifications in the text of the revised manuscript are highlighted in BLUE.

F. Suggested improvements: experiments, data for possible revision

6. *Most of the data generally support a role for VPS15 in coactivation of CLOCK-BMAL1-dependent genes, but they only provide indirect evidence for this conclusion (e.g., genome occupancy, gene expression, nuclear proximity, and the co-IP of VPS15 and BMAL1). The luciferase reporter data in Fig. 3c showing enhancement of BMAL1-dependent E-box-driven luc reporter activity upon coexpression with VPS15 is one way to interrogate this activity in a relatively direct manner. The data look interesting, but the gold standard in the field is to co-express BMAL1 with its partner CLOCK to properly assess activity of the transcription factor in HEK293T cells. With the data displayed as "relative activity", it is not clear that this represents robust activity that is physiologically relevant. For example, how does coactivation by VPS15 activity compare to the activity of known CLOCK-BMAL1 coactivators p300 or CBP? These assays should be repeated looking at CLOCK-BMAL1 activity to support the statement that VPS15 coactivates transcriptional activity of the heterodimer. Alternatively, showing that VPS15 can coactivate BMAL1-dependent activation in a heterologous one-hybrid assay would also provide strong evidence for coactivation of the BMAL1 transcriptional activation domain (TAD) by VPS15, unless it works by binding a completely different region of BMAL1. Along those lines, does VPS15 require the BMAL1 TAD for binding, as other reported coactivators do?*

Response:

As suggested by the reviewer, we complemented the revised manuscript with experiments using the gold standard setup of the luciferase assay upon co-expression of BMAL1-CLOCK:

1) We now show that Vps15 overexpression of either its wild type form or the mutant that does not interact with Vps34 (Vps15-E200R, Vps15mut1) co-activates BMAL1-CLOCK co-expressed in HEK293T cells (**R-Fig. 20a**, corresponding to **Fig. 5a in the revised manuscript**). In the same settings, we show that ectopic expression of Vps15 results in activation of BMAL1-CLOCK comparable to expression of p300 or CBP (**R-Fig. 20a**, corresponding to **Fig. 5a in the revised manuscript**). To reassure the reviewer that we work with robust signals in the luciferase assay, we also present in **R-Fig. 20b** the non-normalized RLU values in one of the assays (the technical replicates in one assay are plotted). These show the magnitude of the non-normalized LUC signal that was commonly observed across the biological replicates of independent luciferase assays. Notably, the differences between conditions are of similar magnitude compared to normalized values presented in **R-Fig. 20a**.

R-Figure 20. a, Luciferase assay in HEK293T cells co-transfected with E-box-Luc reporter construct together with empty vector or plasmids expressing BMAL1 and CLOCK with or without Vps15wt, Vps15mut1, CBP or p300. The relative luminescence presented as fold difference over E-box-Luc only transfected cells (control). Data are means \pm SEM, n=4-12. #: P<0.05, 2-tailed, unpaired Student's t test. Representative immunoblot with anti-Flag antibody shows the expression levels of BMAL1, CLOCK and Vps15 proteins. GAPDH served as loading control. **b**, Representative biological repeat of Luciferase assays presented in **(a)** showing the raw values of RLU. Data are means \pm SEM of n=3 technical replicates of independent wells on the same plate. The expression of BMAL1, CLOCK, Vps15 and p300 constructs is validated by immunoblot.

2) Following the reviewer guidance (*this reviewer and in response to the point 3 of the Reviewer 4*) we have also investigated the protein region that mediates BMAL1 and Vps15 interaction. First, we employed a panel of BMAL1 deletion mutants that were reported in Tamaru et al., Plos Biol, 2009²⁹ (**R-Fig. 21a**, corresponding to **Fig. 4i** in the revised manuscript). We discovered that an interaction of BMAL1 with either CLOCK or Vps15 occurs through different regions of BMAL1 (**R-Fig. 21b**, corresponding to **Fig. 4i** in the revised manuscript). CLOCK interaction was abrupted by deletion of regions in BMAL1 containing PAS-A and PAC domains. At the same time, only deletion of the C-terminal region of BMAL1 was detrimental for its interaction with Vps15 while having no effect on binding of CLOCK (**R-Fig. 21b**, corresponding to **Fig. 4i** in the revised manuscript). This is in line with the insightful suggestion of the reviewer as the C-terminal region of BMAL1 was reported to be necessary for its interaction with Cry1 and CBP^{14,30,31}.

R-Figure 21. a, Schematic representation of HIS-tagged BMAL1 deletion mutants. **b**, Immunoblot analyses of BMAL1-containing immunoprecipitates using anti-Vps15 and anti-CLOCK antibody from total protein extracts of HEK293T cells expressing deletion mutants of BMAL1.

3) To investigate the domains of Vps15 involved in interaction with BMAL1, we generated a panel of truncated Vps15 mutants by deleting the functional domains defined by cryoEM analyses of Vps15-Vps34 complex³² (**R-Fig. 22a**, corresponding to **Extended Data Fig. 6f** in the revised manuscript). Co-immunoprecipitation analyses revealed the specific interaction between BMAL1 and all fragments of Vps15 (**R-Fig. 22b**, corresponding to **Extended Data Fig. 6f** in the revised manuscript). Thus, we concluded that HEAT domain of Vps15 is likely necessary for its interaction with BMAL1.

R-Figure 22. a, Schematic representation of Flag-tagged Vps15 deletion mutants. **b**, Immunoblot analyses of BMAL1-His-containing immunoprecipitates using anti-Flag antibody from total protein extracts of HEK293T cells expressing deletion mutants of Vps15 (Flag tagged) together with BMAL1-His full-length protein.

4) Finally, given that Vps15 binds to C-terminal domain of BMAL1, the same region as Cry1 binding, we asked whether Vps15 co-expression with Cry1 would interfere with its repressive action on BMAL1-CLOCK transcription complex. As expected, expression of Vps15 or Vps15mut1 (does not interact with Vps34) co-activated BMAL1-CLOCK while co-expression of Cry1 repressed BMAL1-CLOCK transcriptional activity (R-Fig. 23, corresponding to Fig. 5b in the revised manuscript). Notably, co-expression of Vps15 together with Cry1 significantly rescued the transcriptional activity of BMAL1-CLOCK compared to the levels observed in empty vector transfected cells (R-Fig. 23, corresponding to Fig. 5b in the revised manuscript). In sum, our findings show that Vps15 co-activates BMAL1-CLOCK and could interfere with repression by Cry1.

R-Figure 23. Luciferase assay in HEK293T cells co-transfected with E-box-Luc reporter construct together with empty vector or plasmids expressing BMAL1 and CLOCK with or without CRY1, Vps15wt or Vps15mut1. The relative luminescence is presented as fold difference over E-box-Luc only transfected cells (complemented with ev instead of BMAL1-CLOCK). Data are means \pm SEM, n=4. #: P<0.05, 2-tailed, unpaired Student's t test. Representative immunoblot shows the expression levels of CRY1, BMAL1, CLOCK and Vps15 proteins. GAPDH served as loading control.

G. References: appropriate credit to previous work?

Seems appropriate.

H. Clarity and context: lucidity of abstract/summary, appropriateness of abstract, introduction and conclusions.

7. The abstract is reasonably clear, although some editing could increase readability. Breaking the introduction and other sections down into paragraphs less than a page in length would help a lot with clarity. Technically, CLOCK and BMAL1 are subunits of the heterodimeric transcription factor and should not be referred to individually as transcription factors (line 55).

Response:

We apologize for this oversight that we have corrected in the revised manuscript text.

Reviewer #3

Summary: In this manuscript, Alkhoury et al. investigated the mechanistic connection between circadian clock-driven metabolic rhythmicity and a critical component in nutrient sensing pathways, class III PI3K, which consists of a kinase subunit Vps34 and a regulatory subunit Vps15. The mammalian molecular clock comprises the heterodimer of two transcription factors, CLOCK and BMAL1, along with their target genes, many of which are negative transcriptional regulators of the CLOCK-BMAL1 complex to form transcriptional/translational feedback loops. Circadian rhythm of this molecular clock can be regulated by multiple nutrient sensing pathways, including mTOR/Class I PI3K, AMPK signaling and autophagy. Class III PI3K are functionally connected to all these pathways, yet its direct functional interaction with the molecular clock remains poorly understood. Using a liver Vps15 knockout mouse model (Vps15LKO), the authors found that multiple gene targets of BMAL1, including *Nr1d1*, *Dbp*, and *Per2*, showed abnormal transcript levels and deregulated circadian rhythmicity in livers of Vps15LKO mice. Similar results were observed in Vps15-null MEF cells. Surprisingly, treatment of Vps34 kinase inhibitors and acute depletion of Vps34 in MEFs showed little effects in BMAL1 transcriptional activity. Therefore, the authors proposed that Vps15 functions independently from Vps34 as a transcriptional coactivator of BMAL1. They found that both Vps15 and Vps34 could localize to the nucleus, but only Vps15 interacted with BMAL1 in the nucleus and was capable of activating *in vitro* transcription from a promoter with BMAL1 binding motif E-box. Further, the authors showed that Vps15 was enriched at the promoter regions of *Nr1d1*, *Dbp* in wild-type (WT) mouse liver extracts, WT MEFs and Vps34-null MEFs. GO pathway analyses for data acquired from BMAL1 and Vps15 ChIP-Seq and bulk transcriptomics in mouse livers pointed out multiple metabolic processes under co-regulation of Vps15 and BMAL1. The authors then performed targeted metabolomics in livers harvested from wild-type or Vps15LKO mice, and they found 63 metabolites from diverse chemical classes showed circadian rhythmicity in WT livers, with 42 of them losing the rhythmicity in Vps15LKO livers. A number of these deregulated metabolites were involved in the *de novo* synthesis of purines, and they showed that a key enzyme in this biosynthetic pathway, PPAT, was under transcriptional control of Vps15/BMAL1, demonstrating rhythmic *de novo* purine synthesis in liver depends on Vps15. At last, to complement studies done with the chronic Vps15LKO mouse model, the authors developed a tamoxifen-inducible Vps15 liver knockout mouse model (Vps15iLKO), and again found that induced Vps15 depletion caused decreased transcript and protein levels and aberrant rhythmicity of BMAL1 target genes, including the newly confirmed Vps15/BMAL1 target *Ppat*. In conclusion, the authors reported that Vps15 has novel function in the nucleus, independent of Vps34, to serve as a transcriptional coactivator of BMAL1, a core molecular component, to regulate the transcriptions of circadian genes and couple the circadian clock to maintain energy homeostasis and metabolic rhythmicity. Overall, the findings here present an unexpected mechanism by which expression of BMAL1 target genes can be regulated. The physiological relevance of the findings remains underexplored, including what factors upstream of Vps15/34 are responsible for modulating the Vps15-BMAL1 interaction and the physiological consequences of the effects of the observed changes to purine metabolism.

Response:

We would like to thank the reviewer for in-depth analysis of our work and for the appreciation of our surprising discovery of class 3 PI3K participating in nuclear transcription and acting as a transcriptional co-activator of BMAL1-CLOCK. From the physiology perspective, our findings are important as **(1)** they expand the current paradigm of post-translational control of the molecular clock components by nutrient sensing signaling pathways, such as mTOR or AMPK signaling in response to feeding or at fasting to nuclear co-factor mechanisms by class 3 PI3K subunits; **(2)** we uncover non-canonical functions of class 3 PI3K beyond its well-known essential role in trafficking as a lipid kinase; **(3)** we uncover temporal control of purine *de novo* synthesis downstream of BMAL1 and Vps15. While the first two points open an entirely new field of future investigations, the latter is a step up compared to fundamental works linking purine synthesis to feeding activated mTORC1 signaling (Robitaille et al., Science, 2013; Ben-Sahra et al., Science, 2016; French et al., Science, 2016)³³⁻³⁵.

Moreover, in response to this reviewer's comment and **the Reviewer 1 (point 1)**, in this revised manuscript, we further link BMAL1, *de novo* purine synthesis, transcriptional response in fasting, and class 3 PI3K. As the zenith of the IMP levels (ZT12) corresponds to the onset of activity/feeding phase, we hypothesized that feeding-fasting might act upstream of BMAL1-Vps15 to regulate *Ppat* expression. Consistent with previous findings, we found that transcript levels of circadian clock genes, *Nr1d1* and *Dbp*, as well as *Ppat* were decreased in livers of fasted control mice (**R-Fig. 4a**, corresponding to **Extended Data Fig. 10a** in the revised manuscript). Notably, they were lower and not further

R-Figure 4. a, Relative transcript levels of indicated genes in liver tissue collected at ZT6 of WT and Vps15LKO mice *ad libitum* fed or 24h fasted. Data are presented as means fold over *ad libitum* fed control \pm SEM, n=4, $P < 0.05$ # vs WT or fed, 2-tailed, unpaired Student's t test. **b**, Immunoblot analysis of cytosolic and nuclear soluble fractions of liver tissue of WT mice treated as in (a) using indicated antibodies. Densitometric analyses of Vps15 or BMAL1 proteins normalized to Tubulin (cytosolic fraction) or H3 (nuclear soluble fraction) levels presented as folds over control fed \pm SEM, n=4, $P < 0.05$ # vs WT fed. **c**, ChIP-qPCR analyses of BMAL1 and Vps15 enrichment at promoter regions of indicated gene in chromatin of liver tissue of *ad libitum* fed and 24h fasted mice collected at ZT6. Data are presented as means fold enrichment over IgG \pm SEM, n=3-4, $P < 0.05$ * vs IgG, 2-tailed, unpaired Student's t test.

reduced in fasting of Vps15LKO mice (**R-Fig. 4a**, corresponding to **Extended Data Fig. 10a** in the revised manuscript). Yet, neither nuclear BMAL1 protein levels nor its chromatin recruitment to the promoters of *Nr1d1* and *Ppat* target genes were modified by fasting (**R-Fig. 4b,c**, corresponding to **Extended Data Fig. 10b,c** in the revised manuscript). Notably, Vps15 protein levels in nucleus were decreased in livers of fasted control mice but not in the cytosol (**R-Fig. 4b**, corresponding to **Extended Data Fig. 10b** in the revised manuscript). Moreover, consistent with decreased Vps15 nuclear levels, Vps15 recruitment to promoters of BMAL1 target genes was abrogated in livers of fasted mice (**R-Fig. 4c**, corresponding to **Fig. 6k** in the revised manuscript).

Taken together with the transcript and metabolomics analyses, these data suggest that Vps15 could act as a transcriptional co-activator of BMAL1 for expression of circadian clock genes and the key enzyme in *de novo* purine synthesis in response to nutrient levels *in vivo*. The mechanisms of how chromatin recruitment of Vps15 is controlled in response to nutrients is a new question that will be a focus of our future works.

Other major comments:

1. One important claim the authors made in the manuscript is that the lipid kinase subunit of the Class III PI3K, Vps34, is dispensable for the clock regulation. To support this claim, the authors used two structurally unrelated Vps34 inhibitors to block lipid kinase activity and generate Vps34^{ff} MEFs to semi-acutely deplete Vps34. The results of these experiments are a little perplexing. For example, different inhibitors showed mixed effects in Dbp levels, and Per2 was sensitive to Vps34 modulation but not Nr1d1. Do the two inhibitors use different mechanisms to suppress Vps34 activity? Why are some transcript levels strongly affected by Vps34 depletion/inhibition, but others were not (yet all of them are reported as gene targets of BMAL1)? Further, in Vps34^{ff} MEFs, the protein levels of REV-ERB α were shown not to be reduced by Vps34 depletion (Fig. 2d), but if we compare the levels of REV-ERB α in the GFP control of Vps34^{ff} and Vps15^{ff} cell lines (western blot in Fig. 1g), we can see much higher REV-ERB α level at earlier timepoints (16-24 h) compared to later timepoints (28 – 36 h) in Vps15^{ff} cells, but not as a big difference in Vps34^{ff} cells. Does this indicate that REV-ERB α levels were lower in Vps34^{ff} GFP control cells during those time points to start with, and thus upon Vps34 depletion no reduction was observed? This part of the study would be more convincing if the authors could provide more detailed explanations for the discrepancy in the responses to different treatments instead of glossing over the observations. Also, the authors should conduct the rhythm analysis on these inhibitor experiments like in the Vps15^{LKO} livers and Vps15-null MEFs, if possible. Currently, only the differences in the target gene transcript level are reported.

Response:

We thank the reviewer for these insightful comments that prompted us to better present our findings, to clarify our conclusions, and to provide additional controls. New findings, together with the data already presented in the manuscript, show that Vps15 does not rely on Vps34 for co-activation of BMAL1-CLOCK. This conclusion is based on the use of the point mutant of Vps15 that does not interact with Vps34 (Vps15E200R), while it co-activates BMAL1-CLOCK transcriptional complex as efficiently as the wild-type form of Vps15 (**R-Fig. 20, the response to the Reviewer 2, point 6**). Nevertheless, these findings do not rule out the involvement of Vps34 in nuclear transcription. In support of this wider role of class 3 PI3K, in this revised manuscript, we show that both Vps15 and Vps34 co-localize on *de novo* transcription sites and interact with RNA Pol2 (**R-Fig. 11a, the response to the Reviewer 1, point 6**). In sum, our findings discover novel nuclear role of both Vps15 and Vps34 in transcription, yet with regards to BMAL1-CLOCK, Vps15 does not rely on Vps34. Below, we provide detailed responses to the insightful comments of the reviewer:

1) We apologize for being not clear in our explanations regarding REV-ERB α expression in Vps15^{ff} and Vps34^{ff} MEFs. Different immunoblot panels, including those presented in **Fig.1g** and **Fig.2d**, to which the reviewer refers to, are derived from independent experiments, and cannot be compared directly. Notably, all experiments in cells were performed with their internal controls of non-depleted cells (transduced with GFP expressing adenovirus). To address the comment of the reviewer, we have directly compared the protein expression in total extracts of Vps15^{ff} and Vps34^{ff} MEF lines. For this, we analyzed REV-ERB α , Vps15, and Vps34 protein levels in cells collected at different times after synchronization with Dexamethasone and in unsynchronized cells (**R-Fig. 24**). These immunoblot analyses showed comparable expression of all analyzed proteins between two cells lines of MEFs.

R-Figure 24. Immunoblot analysis of total protein extracts using indicated antibodies in unsynchronized and Dexamethasone-synchronized Vps15^{ff} and Vps34^{ff} MEFs. Densitometric analyses of BMAL1 protein normalized to Actin levels presented as folds over CT16 condition. Data are presented as means \pm SEM.

2) We recognize that the observations with different inhibitors of Vps34 could appear confusing and we apologize for providing insufficient explanations for this important point. Essentially, **Reviewer 2 (point 4)** has raised the same concern, thus, we copy here our response to both of the reviewers.

First of all, we employed selective inhibitors of Vps34 as they represent a useful tool for acute inhibition of lipid kinase activity of Vps34 without impacting its protein levels. We agree with the reviewer that at first sight the effects of Vps34-inhibitors on the clock function are puzzling. Yet, these differences are likely due to general shortcomings when using inhibitors such as inhibition of “off-targets”. Our observations highlight the importance of employing structurally different drugs in the studies which likely have non-overlapping off-targets. In this scenario, when using structurally unrelated compounds that effectively inhibit its intended target (e.g. lipid kinase Vps34), only common observations could be attributed to repression of their intended target. At the same time the inconsistencies are likely due to their action on other proteins.

In this revised manuscript, in addition to SAR405 and PIKIII, we have utilized Vps34-IN1, another efficient inhibitor of Vps34 (IC₅₀ 4nM). Treatment with Vps34-IN1 inhibited Vps34, as witnessed by decrease in PI3P levels and p62 protein accumulation (**R-Fig. 19a, b**, corresponding to **Extended Data Fig. 3c, d** in the revised manuscript). At the same time, it had no effect on REV-ERB α protein levels (**R-Fig.19b**). Surprisingly, treatment of MEFs with Vps34-IN1

R-Figure 19. a, PI3P detection with a recombinant protein HRS-RFP probe in Vps15^{fl/fl} MEFs treated with Vps34-IN1 (5 μ M) for 18 hours. The incubation with HRS-mut and the condition without the probe served as controls of non-specific binding. Scale bar, 10 μ m. Protein (n=3) (**b**) and relative transcript levels (n=9) (**c**) in synchronized with Dexamethasone MEFs treated with DMSO (control) or Vps34-IN1. Densitometry analyses of REV-ERB α protein normalized to Actin levels presented as fold over DMSO treated cells. Data are presented as means \pm SEM, P<0.05 # vs DMSO, 2-tailed, unpaired Student's t test.

led to an increase in the transcript levels of *Nr1d1* and *Cry1* while levels of *Per2* decreased and no effect was observed for *Dbp* transcript (**R-Fig.19c**).

Importantly, the three compounds that we employed are efficient in inhibiting lipid kinase activity of Vps34. As reported in *Ronan et al.*²¹ SAR405 has an IC₅₀ of 1 nM in the phosphorylation of a PtdIns substrate by human recombinant Vps34 enzyme while showing low inhibition up to 10 μM of class I and class II PI3Ks as well as of mTOR. Yet, as could be seen in the Supplementary dataset, while largely specific to Vps34 lipid kinase, 1 μM of SAR405 inhibited 50-75% of SMG1 kinase and above 60% of PIK3Cb and PIK3CD. As for PIKIII, *Dowdle et al.*²² tested the selectivity of this inhibitor on 44 Kinases and found it to be 100 folds more selective on Vps34 than other kinases. However, as presented in the Supplementary Table 1 of the original report, it inhibits other kinases at the following IC₅₀ concentrations: CDK2A (2.6 μM), GSK3B (4.65 μM), AURKA (3.9 μM), ABL1 (8.8 μM), EphB4 (9.4 μM), FLT3 (7.2 μM), JAK2 (8.7 μM), KDR (9.1 μM), MAPK1 (6.8 μM). For Vps34-IN1, *Bago et al.*²³ described the high selectivity of this inhibitor for Vps34. However, we can see in Figure 2B and 2C in the original report, that Vps34-IN1 inhibits other kinases at the IC₅₀ concentration of: PI3Kd (1.89 μM), PI3Kg (2.685 μM), PIP5K1C (0.382 or 3.37 μM), PIP5K1A (4.93 μM), PI4K3B (1.069 μM). Our non-exhaustive bibliography search showed that these potential off-targets of different Vps34 lipid kinase inhibitors were also reported to have functional links with the clock: CDK2a²⁴, GSK3b²⁵, ABL1²⁶, MAPK1²⁷, SMG1²⁸.

Finally, in this revised manuscript, we also present rhythm analyses using bioluminescence of *Nr1d1*-Luc reporter in cells treated with three different Vps34 inhibitors (**R-Fig. 25**, corresponding to **Extended Data Fig. 3f,g** in the revised manuscript). We found that treatment with all inhibitors in two doses (2.5 and 5 μM) did not show any major effect on the *Nr1d1*-Luc reporter activity in MEF cells (**R-Fig. 25a**, corresponding to **Extended Data Fig. 3f** in the revised manuscript). At the same time, treatment with three inhibitors in both doses efficiently inhibited autophagy as judged by immunoblot showing accumulation of p62 and LC3 proteins (**R-Fig. 25b**, corresponding to **Extended Data Fig. 3f** in the revised manuscript). This is consistent with the expression analyses showing unmodified levels of REV-ERBa protein and its transcript in MEF cells treated with these compounds.

R-Figure 25. a, Bioluminescence recordings of *Nr1d1*-Luc oscillations in synchronized with Dexamethasone MEFs treated with DMSO or Vps34 lipid kinase inhibitors (PIKIII, SAR405, and Vps34-IN1) in indicated doses during the entire recording. Data are expressed as average of detrended values from n=4 independent experiments. **b**, Representative immunoblot analysis of LC3 and p62 proteins in MEF cells treated with DMSO (control), SAR405, PIKIII or Vps34-IN1 collected at the end of the recording (**a**).

In sum, our findings point to the scenario that the transcription of certain circadian clock genes such as *Per2* is inhibited by Vps34 lipid kinase activity while transcription of *Nr1d1* relies on Vps15. The future work will address the mechanisms of Vps34 lipid kinase in the nuclear gene transcription. We have modified the text of the revised manuscript to clarify this important point (**line 201-214** in the revised manuscript).

2. The transcript levels of *BMAL1* target genes in *Vps34*-null MEFs were similar to the GFP control (Fig. 2e, Extended Data Fig. 3b), which was drastically different when compared to *Vps15*-null cells (Fig. 1h, Extended Data 2c). However, in *Vps34^{fl/fl}* MEFs, the overall protein level of *Vps15* was also reduced to ~30% (Fig. 2d, Extended Data Fig. 3a). The authors explained that this observation was due to higher retention of *Vps15* in the nucleus of *Vps34*-null cells (70% reduction in cytosol vs 30% reduction in the nucleus). However, since no cytosolic protein markers were included in the representative western blots, it is hard to tell the purity of the nuclear soluble fractions and thus fully trust the quantification.

Response:

We apologize for omitting this fractionation purity control that we performed but did not show to keep the figure less busy. We now show an immunoblot with Lamin A/C and Tubulin for respective cytoplasmic and nuclear fractionation controls from the presented experiment (R-Fig. 26, corresponding to **Extended Data Fig. 7d** in the revised manuscript). Quantification of *Vps15* and *Vps34* in the nuclear fraction was done as the ratio to β -Catenin as it showed least of variation between GFP and CRE treated cells (consistent with equal protein loading after Bradford protein assay).

R-Figure 26. Immunoblot analysis of cytosolic and nuclear soluble protein extracts using indicated antibodies in Dexamethasone-synchronized control (GFP) and *Vps34*-depleted (CRE) MEFs. Densitometric analyses of *BMAL1* protein normalized to Tubulin (cytosol) and to β -Catenin (nuclear soluble) levels presented as fold over GFP treated cells. Data are presented as means \pm SEM, n=8. The bottom panel shows immunoblot of Tubulin and Lamin A/C as controls for nuclear and cytosolic fraction purity.

3. Also, the authors should include the ratio of cytosolic vs. nuclear pools of *Vps15* in control GFP MEFs and in *Vps34*-null MEFs? These data would support the authors' explanation if they can show the remaining level of *Vps15* in the nucleus of *Vps34*-null cells is comparable to the *Vps15* level in the nucleus of the control GFP cells, so the remaining *Vps15* in the *Vps34*-null nucleus is sufficient to support its function as a transcriptional coactivator.

Response:

We thank the reviewer for this suggestion that prompted us to provide additional analyses that indeed reinforce the initial conclusions on the presence of a *Vps15* pool that could be independent of *Vps34*. As suggested by the reviewer, we analyzed *Vps15* and *Vps34* protein expression in cytosolic and nuclear fractions of *Vps15* and *Vps34* depleted MEFs in the same experiment (R-Fig. 27, corresponding to **Extended Data Fig. 7e** in the revised manuscript). As evidenced on R-Fig. 27, the expression of nuclear *Vps15* in *Vps34*-depleted MEFs is twice higher compared to

Vps15-depleted MEFs. These observations are also in line with our findings with Vps15mut1 that does not interact with Vps34, yet could co-activate BMAL1-CLOCK as potently as Vps15wt in the luciferase assay (R-Fig. 20a, the response to the Reviewer 2, point 6).

R-Figure 27. Immunoblot analysis of cytosolic and nuclear soluble protein extracts using indicated antibodies in Dexamethasone-synchronized control (GFP) and Vps15-depleted (CRE) MEFs. Densitometric analyses of Vps15 and Vps34 proteins normalized to Tubulin (cytosolic fraction) and Lamin A/C (nuclear soluble fraction) levels presented as folds over GFP treated cells. Data are presented as means \pm SEM, n=3. P<0.05 #: vs GFP, 2-tailed, unpaired Student's t

4. For subcellular fractionation experiments done in the manuscript, only western blots shown in Extended Data Fig. 4a and b included a nuclear protein marker for the cytosolic fraction, and a cytosolic protein for the nuclear fraction to demonstrate the purity of these fractions. This practice should be extended to at least one representative western blot for subcellular fractionation performed in different cell lines (e.g. Vps15^{fl/fl} MEF, Vps34^{fl/fl} MEF, etc) and from mouse liver extracts to show the robustness of the procedure.

Response:

We apologize if we were not sufficiently clear in our explanations, and we thank the reviewer for helping us to improve presenting the data. We employ a protocol for cytosolic-nuclear fractionation that consistently produced the fractions with minimal cross-contamination as could be observed from purity controls presented on Fig. 3a, 3f, Extended Data Fig. 5a, 5f, 6b, 7d and 7e in the revised manuscript. Specifically, the Fig. 3f, Extended Data Fig. 5f, 6b, 7d and 7e are new data added to the revised manuscript.

5. The authors performed co-IP, PLA and E-box-Luciferase reporter experiments to demonstrate that Vps15 interacts with BMAL1 in the nucleus, is able to initiate in vitro transcription, and that Vps34 was not needed for all these processes. The following experiments are necessary to support the proposed mechanism.

Response:

We thank the reviewer for these insightful questions that improved the revised manuscript. The main conclusion of our revised work is that both Vps15 and Vps34 participate in nuclear transcription and the function of Vps15 as a transcriptional co-activator of BMAL1-CLOCK does not rely on its interaction with Vps34. However, our findings do not rule out, and are actually in favor of Vps34 having functions in nuclear transcription. First, as was presented in the previous version of the manuscript, we found both Vps34 and Vps15 in the nucleus of liver and cells (Fig. 3, Extended Data Fig. 5 in the revised manuscript). Second, in the revised manuscript, we also show that ectopically expressed Vps34 as Vps15 localize on *de novo* transcription sites (R-Fig. 11a, corresponding to Fig. 3c in the revised manuscript) and they interact with RNA Pol2 (R-Fig. 12a, corresponding to Fig. 3d in the revised manuscript). Third, we observed nuclear PI3P, and its levels are sensitive to Vps34 lipid kinase inhibition (R-Fig. 6,

corresponding to **Extended Data Fig. 2c, 3c** in the revised manuscript). However, the presence of a functional clock in Vps34-null MEFs and the findings that Vps15-mut1 which does not interact with Vps34 but could co-activate BMAL1-CLOCK are in favor of Vps15 being a co-activator of BMAL1-CLOCK independently of Vps34. Thus, in this revised manuscript, we further investigated the capacity of ectopically expressed Vps34 to interact and to co-activate BMAL1-CLOCK. First, we observed proximity puncta between overexpressed Vps34 and endogenous BMAL1 (**R-Fig. 28a**, corresponding to **Extended Data Fig. 7a** in the revised manuscript). Second, we conducted luciferase assays in conditions of BMAL1-CLOCK co-expression with increasing amounts of Vps15 or Vps34. Surprisingly, we observed that high level of Vps34 overexpression also led to E-box-LUC reporter activation (**R-Fig. 28b**, corresponding to **Extended Data Fig. 7b** in the revised manuscript). However, only over 10-fold overexpression of Vps34 (dose 3) resulted in significant co-activation of BMAL1-CLOCK (**R-Fig. 28b**, corresponding to **Extended Data Fig. 7b** in the revised manuscript). This is unmatched to Vps15, that resulted in BMAL1-CLOCK co-activation with 2.4-fold overexpression compared to endogenous Vps15 (dose 2) (**R-Fig. 28b**, corresponding to **Extended Data Fig. 7b** in the revised manuscript). Moreover, given the findings of Vps15 overexpression in mitigating the repressive effect of CRY1

R-Figure 28. a, Proximity ligation assay between endogenous BMAL1 and ectopic Flag-Vps34 protein in HEK293T cells (co-transfection with GFP-expressing plasmid was used to visualize transfected cells). The no antibody condition served as a control of non-specific signal. Scale bar 10 μ m. The quantification of the proximity puncta per cell in n=20-290 cells is presented. #: P<0.05, 2-tailed, unpaired Student's t test. **b**, Luciferase assay in HEK293T cells co-transfected with E-box-Luc reporter construct together with empty vector or plasmids expressing BMAL1 and CLOCK with or without increasing doses of Vps15wt or Vps34wt. The relative luminescence presented as fold difference over E-box-Luc only transfected cells. Data are means \pm SEM, n=3-4, #: P<0.05, 2-tailed, unpaired Student's t test. Representative immunoblot with Vps15 and Vps34 antibodies shows the overexpression levels. GAPDH served as loading control. **c**, Luciferase assay in HEK293T cells co-transfected with E-box-Luc reporter construct together with empty vector or plasmids expressing BMAL1 and CLOCK with or without CRY1, Vps15wt or Vps34 (**complementary to R-Fig. 23**). The relative luminescence presented as fold difference over E-box-Luc only transfected cells. Data are means \pm SEM, n=4. #: P<0.05, 2-tailed, unpaired Student's t test.

on BMAL1-CLOCK (**R-Fig. 23** corresponding to **Fig. 5b** in the revised manuscript), we also tested whether high level of Vps34 overexpression (over 10-fold compared to the endogenous levels) would produce a similar effect. Remarkably, while Vps34 overexpression activated BMAL1-CLOCK, it did not interfere with the repressor action of CRY1 (**R-Fig. 28c**, corresponding to **Extended Data Fig. 7b** in the revised manuscript). Thus, these observations, together with other Vps34-related findings, open to the exciting possibility that Vps34 and Vps15 have distinct roles in nuclear transcription that we aim to investigate in future works. We thank the reviewer for guiding us through the comments and questions into performing these mechanistic analyses.

(a) For the co-IP experiment shown in Extended Data Fig. 5a and b, the authors should probe for Vps34 in the co-IP from the different fractions to assess if Vps34 and Vps15 complex formation is different in the cytosol and in the nucleus. Supposedly, Vps34 might show less interaction with Vps15 in the nucleus.

(b) Alternatively, the authors could consider repeating the FLAG IP shown in Fig.3a in cytosol and nuclear fractions, and similarly probe for Vps34 in the co-IP in addition to BMAL1.

(c) For the PLA experiment shown in Fig. 3b, the authors should add a panel for FLAG-BMAL1 and Vps34 to show negative interaction for these two proteins.

Response:

Following the suggestion of the reviewer, we analyzed Vps15-Vps34 complex in cytosolic and nuclear fractions. As expected, the complex between Vps15 and Vps34 was apparent in the cytosol and the interaction between Vps15 and BMAL1 was readily observed in the nuclear fraction from liver and from MEFs (**R-Fig. 29a,b**, corresponding to **Fig. 4f** and **Extended Data Fig. 6b** in the revised manuscript). Also, the complex between Vps15-Vps34 was readily detected in nuclear extracts of liver and to a lesser extent in MEFs. Moreover, the PLA assay between endogenous Vps15 and overexpressed Vps34-Flag showed predominantly cytosolic localization of the complexes with minority of PLA dots detected in the nucleus (**R-Fig. 29c**, corresponding to **Extended Data Fig. 5h** in the revised manuscript). The latter is consistent with lower amount of nuclear Vps15 and Vps34 compared to levels present in cytosolic fraction. To consolidate these findings with overexpressed Vps34-Flag, we also tested the PLA between endogenous Vps15 and Vps34. As could be seen on **R-Fig. 29d**, there were much fewer PLA dots detected between endogenous Vps15 and Vps34 compared to Vps34-Flag and endogenous Vps15. Moreover, they almost exclusively localized to the cytosol, with the exception of few cells possessing some nuclear PLA signal of endogenous Vps15 and Vps34 proteins (**R-Fig. 29d**). Although we cannot rule out that anti-Vps34 antibodies are not adapted for the PLA assay, these observations of mostly cytosolic complex between Vps15-Vps34 are in line with the PI3P detection, predominantly in the extra-nuclear compartment, as could be seen on **R-Fig. 6** in response to **the point 6 of the Reviewer 1**. Finally, as suggested by the reviewer, we also tested PLA between Flag-Vps34 and endogenous BMAL1 as we had all confidence in the specificity of these antibodies that we validated in PLA assay. As could be seen on **R-Fig. 28a**, we detected specific PLA dots in GFP+ cells that were co-transfected with Vps34-Flag. Notably, the majority of those PLA dots were localized in the cytosol with only minor dots present in the nucleus (**R-Fig. 28a**, corresponding to **Extended Data Fig. 7a** in the revised manuscript).

(d) Also, using these same antibodies for FLAG (BMAL1), Vps15 and potentially Vps34 to perform IF in parallel with the PLA procedures would allow assessment of the subcellular localizations of these proteins (BMAL1, Vps15 and Vps34 were all reported to show both cytosolic and nuclear localization), and would further support the authors about the Vps15-BMAL1 interaction claim if the PLA signal is only observed in the nucleus for Vps15 and FLAG-BMAL1.

Response:

As also suggested by **the Reviewer 1 (point 6)**, we have extensively validated the presence of a nuclear pool of Vps15 in immunofluorescent microscopy analyses. First, we have tested three available antibodies against Vps15

R-Figure 29. Immunoblot analyses of Vps15-containing immunoprecipitates from nuclear soluble and cytosol fractions of liver tissue of *ad libitum* fed WT mice collected at ZT6 (**a**) or MEF cells (**b**) using indicated antibodies. **c**, Proximity ligation assay between endogenous Vps15 and ectopic Flag-Vps34 protein in HEK293T cells (co-transfection with GFP-expressing plasmid was used to visualize transfected cells). The no antibody condition served as a control of non-specific signal. Scale bar 10 μ m. The quantification of the proximity puncta per cell in n=20-380 cells is presented as graph, #: P<0.05 vs no antibody, 2-tailed, unpaired Student's t test. **d**, Proximity ligation assay between endogenous Vps15 and Vps34 proteins in HEK293T cells. The no antibody condition served as a control of non-specific signal. Scale bar 10 μ m. The quantification of the proximity puncta per cell in n=10-20 cells is presented as graph. #: P<0.05 vs no antibodies, 2-tailed, unpaired Student's t test.

and one antibody against Vps34, which we have validated are specific in immunoblot analyses and immunoprecipitation. As could be seen from **R-Fig. 10a**, all antibodies against Vps15 are specific to Vps15 and show uniform cytosolic-nuclear staining. Unfortunately, commercially available antibodies against Vps34 did not provide reliable staining under different fixation and cell permeabilization protocols (*data not shown*). Thus, we could not investigate the sub-cellular distribution of endogenous Vps34 by immunofluorescence. We expect that, in line with immunoblot analyses of subcellular fractions of cells and liver tissue of Vps15 mutants, Vps34 protein levels are decreased both in cytosol and in the nucleus in Vps15-depleted cells. Given the specificity of anti-Vps15 antibodies, we now provide microscopic evidence of nuclear pool of endogenous Vps15 in MEF cells. As the immunoblot analyses of nuclear vs cytosolic fractions suggested that nuclear Vps15 represents a less important pool compared to Vps15 in the cytosol, to better visualize nuclear Vps15 in MEFs, we have employed CSK buffer treatment of cells before fixation that allows washing out of cytosolic soluble proteins¹⁷. Moreover, depending on incubation time, the CSK buffer treatment facilitates the visualization of tightly bound fraction of nuclear proteins. As seen from **R-Fig. 10b** already 1 min of incubation with CSK revealed prominent nuclear staining of endogenous Vps15. Notably, it weakens with longer

R-Figure 10. a, Immunofluorescent microscopy analyses of Vps15 in MEF using three different anti-Vps15 antibodies (scale bar, 10 μ m). To validate the specificity of anti-Vps15 antibodies, MEF cells were transduced with the adenoviral particles expressing shRNA Vps15 and analyses were performed 72 hours post infection. The efficiency of depletion and the specificity of anti-Vps15 antibodies was validated by immunoblot with indicated antibodies. **b** and **c** Immunofluorescent microscopy analyses of Vps15 using two antibodies raised in different hosts (ab1 (mouse monoclonal) and ab2 (rabbit polyclonal)) in MEF cells treated for indicated times with CSK buffer. The staining with DAPI and anti-Histone 3 antibody served as nuclear staining control. Scale bar, 10 μ m. **d**, Immunofluorescent microscopy analyses of endogenous Vps15 and BMAL1 in MEF cells. Scale bar, 5 μ m.

(3 min) incubation. This is consistent with the wash-out of non-cytoskeletal proteins, an expected outcome of CSK buffer treatment. These findings were validated with two different anti-Vps15 antibodies (mouse monoclonal (Ab1) and rabbit polyclonal (Ab2) further reassuring the observation of nuclear Vps15 (**R-Fig. 10b, 10c** corresponding to **Fig. 2b** in the revised manuscript). As suggested by the reviewer, we have also co-stained endogenous Vps15 and BMAL1 in MEF cells which showed a partial co-localization in the nucleus as puncta (**R-Fig. 10d** corresponding to **Extended Data Fig. 6d** in the revised manuscript). Finally, we extended these analyses to a *de novo* transcription assay in cell that demonstrated co-localization of ectopically expressed Vps15 and Vps34 with active transcription sites in MEFs (**R-Fig. 11a** corresponding to **Fig. 3c** in the revised manuscript). These were validated with endogenous Vps15 that showed prominent colocalization with *de novo* transcription sites labelled with BrUTP and RNA Pol2 phospho-Ser5 (transcriptional elongation form). Notably, the triple positive BrUTP/RNA Pol2/Vps15 sites were detected in hepatocyte nuclei outside of nucleoli (**R-Fig. 11c** corresponding to **Fig. 3e** in the revised manuscript). Unfortunately, we could not perform these analyses with Vps34 as we lack antibodies that would specifically recognize endogenous Vps34 in immunofluorescence microscopy.

R-Figure 11. a, Transcription assay with ectopically expressed Vps15-Flag and Vps34-Flag proteins in MEFs cells. 24hrs post-transfection MEFs were incubated with BrUTP to label *de novo* transcription sites. For co-localization, immunofluorescent microscopy was performed with anti-Flag and anti-BrUTP antibodies. Scale bar 5 μ m. **b**, Immunofluorescent microscopy analyses of co-localization of endogenous RNA Pol2 phSer5, Vps15 and *de novo* transcription sites labelled with BrUTP in primary hepatocytes. Scale bar 10 μ m and 5 μ m (inset).

(e) Showing no Vps34 enrichment at the promoters of *Nr1d1* and *Dbp* would also strengthen the statement that Vps34 was dispensable for the transcription of clock related genes.

Response:

In response to the suggestion of the reviewer to ChIP Vps34, one should keep in mind that this assay is highly dependent on the quality of antibody. Thus, a lack of specific enrichment does not irrefutably demonstrate absence of binding and should be interpreted with caution. Importantly, there are no commercially available ChIP-grade validated anti-Vps34 antibodies. We have validated anti-Vps34 antibodies from Cell Signaling (cat#: 4263S) as the one that

R-Figure 30. a, Immunoblot analyses of Vps15 in immunoprecipitates using anti-Vps34 antibody from total protein extract of WT livers collected at ZT6. **b**, ChIP-qPCR analyses of BMAL1 and Vps34 enrichment at promoter regions of indicated genes in liver tissue collected at ZT6. Data are presented as means fold enrichment over IgG \pm SEM, n=3, P<0.05 * vs IgG, 2-tailed, unpaired Student's t test. No specific enrichment above 5-fold was detected for Vps34 immunoprecipitates.

efficiently immunoprecipitates Vps34 (R-Fig. 30a). We used this antibody in ChIP and observed no specific enrichment above 5-fold over the IgG background in samples with anti-Vps34 antibody, unlike with anti-BMAL1 that was used in this assay as a positive control (R-Fig. 30b). This response was observed on three target genes of BMAL1 that we tested – *Nr1d1*, *Dbp*, *Ppat*. To note, we have optimized the ChIP protocol by using more chromatin material for reaction as could be seen from high level of BMAL1 enrichment. Given that we cannot exclude the technical issue of non-adapted anti-Vps34 antibodies for ChIP, we did not include these findings in the manuscript. We believe the role of Vps34 on the chromatin is highly likely given its nuclear presence, co-IP with RNA Pol2, co-localization of ectopically expressed Vps34 with *de novo* transcription sites, and co-activation effect of BMAL1-CLOCK by highly overexpressed Vps34 that we show in this revised manuscript. We are planning to investigate Vps15- and Vps34-bound cistromes in the future.

6. The authors used the JTK-CYCLE algorithm from R for circadian rhythmicity analysis of proteins, mRNAs and metabolites levels in the manuscript. The authors should provide brief explanation of the analyses in the method section to help the audiences better understand how the calculations were done and interpret the data correctly. For example, in Extended Data Fig.4, the authors demonstrated that the protein and transcript levels of Vps15 but not Vps34 show rhythmic pattern according to the analysis by the JTK-CYCLE algorithm. However, just from looking at the representative western blots and the quantification plots for the protein and mRNA levels of these two proteins, sometimes both Vps15 and Vps34 seemed to show oscillating expression pattern (Extended Data Fig.4d), or neither seemed to show a pattern (Extended Data Fig.4f), but the p-values from the rhythmic analysis for the two proteins are very different.

Response:

We thank the reviewer to noting to this oversight that we have now corrected in the revised manuscript. We have added the information of the JTK-CYCLE parameters that we employed for rhythmicity analyses. Moreover, we are grateful for reviewer noting the mistake on labeling of the p-value for analyses of Vps34 protein rhythmicity in the euchromatin fraction (Extended Data Fig. 5c in the revised manuscript). Indeed, as evident from the figure, it is cycling, and the mistake was caused by accidental switching of the columns in the Excel data file. The raw data and JTK calculations are provided as accompanying files in the revised manuscript (Supplementary Table 1 and Source Data-Raw data).

7. ChIP-Seq and ChIP-qPCR analyses for Vps15 showed much lower enrichment at transcription start sites and at promoter regions of Nrdr1, Dbp and Ppat than BMAL1, indicating that the Vps15 and BMAL1 interaction is sub-stoichiometric (many molecules of BMAL1 not bound to Vps15). The authors should discuss how a sub-stoichiometric Vps15 interaction affects the overall BMAL1 transcriptional activity so drastically (as seen by tens and hundreds fold differences in some BMAL1 target genes transcript levels in WT vs Vps15LKO or Vps15iLKO livers or Vps15-null cells).

Response:

We thank the reviewer for raising this point that we now clarify in the revised version of the manuscript. Indeed, the consistent yet modest enrichment of Vps15 in the same promoter regions bound by BMAL1 is about 20-30-fold lower compared to the one observed with anti-BMAL1 antibody (compare fold enrichment of BMAL1 and Vps15 in **Fig. 5c or Fig. 6h in the revised manuscript**). This could be explained by technical particularity of the ChIP assay as well as biological aspects of Vps15-BMAL1 interaction. First, the ChIP assay is an antibody-based technique and depends on the efficacy of the antibody to pull down crosslinked protein-DNA complexes. Thus, we cannot exclude that the anti-Vps15 antibody we used in ChIP are not ideal for this application. Second, BMAL1 binds directly to DNA as it is a transcription factor while Vps15 is most likely indirectly associated with chromatin as it does not have DNA-binding domain. Thus, enrichment of Vps15 is inherently lower compared to BMAL1's due to its lower recuperation from chromatin. It is a common observation that the highest to lowest ChIP enrichment is obtained in the following order: histones, RNA Pol2, transcription factors, and finally co-factors. We briefly mentioned this in the revised manuscript (**lines 370-375**). New data that we included in this revised manuscript further reinforce the confidence that nuclear Vps15 has a role in RNA Pol2 driven transcription (**R-Fig. 11-12**) and interacts with BMAL1 through the TAD domain, commonly associated with the binding of BMAL1 co-factors (**R-Fig. 21**).

8. Based on data shown in the manuscript, Vps15 did not seem to facilitate the recruitment of BMAL1 to chromatin (Fig. 2g, Extended Fig. 7f, line 202-206; reduced BMAL1 occupancy at target gene promoters observed in the chronic Vps15LKO liver was probably due to decreased BMAL1 protein level), and there was not enough data showing Vps15 depletion negatively regulates the BMAL1-CLOCK heterodimer formation. Also, the nuclear pool of Vps15 seemed to show rather stable levels despite a minor reduction during the dark phase (Extended Data Fig. 4c-e). Therefore, for the proposed working model that Vps15 couples transcriptional activity of the circadian clock with regulation of energy homeostasis, the authors should provide some more evidence of how Vps15 may facilitate the activation of BMAL1 as its transcription coactivator, and how could this activated transcriptional machinery for clock genes and purine de novo synthesis be terminated/paused when necessary. Does Vps15 constitutively interact with BMAL1 in the nucleus to bolster its transcriptional activity, and thus the rhythm of the system only comes from the regulation of BMAL1 expression from other clock components, or the Vps15-BMAL1 interaction itself is also rhythmic and subject to some regulations upon receiving signaling cues?

Response:

We fully agree with the reviewer that our work opens to the follow-up questions on the mechanisms upstream of the Vps15 interaction with BMAL1. In this revised manuscript we provide additional evidence that Vps15 expression does not impact BMAL1-CLOCK complex (**R-Fig. 14b, 14c, the response to the Reviewer 1, point 8**). Also, as described in our response **point 1** above as well as to **the Reviewer 1 (point 1)**, in this revised manuscript, we provided insights of correlating BMAL1 transcriptional activity under fasting and Vps15 nuclear levels as well as its chromatin recruitment (**R-Fig. 4**). We show that the nuclear levels and chromatin recruitment of Vps15 to BMAL1 promoter target genes were decreased in livers of fasted mice (**R-Fig. 4b,c**). Thus, our findings suggest that Vps15 undergoes active controlled nuclear transport and chromatin recruitment, warranting future mechanistic studies of this process in response to fasting and potentially other physiological stresses.

Minor comments for specific figures:

1. Fig. 1a: For the time course plots like the two shown in Fig.1a, if there are no data points between adjacent time points, the data point at each time point should not be connected by smoothed/curve lines, which may show visually deceiving trends, but rather by straight lines.

Response:

We thank the reviewer for the suggestion. In the revised manuscript all graphs are re-plotted with individual replicates or animals indicated as scatter-plot graphs. Also, for the transcript analyses around the clock, a loess regression curve was fitted to the log2-normalized data in R, and rhythmicity adjusted p-values extracted from JTK_CYCLE are presented in figures and all available in **Supplementary Table 1**.

2. Fig. 1b (and for the rest): Representative Western blots should include molecular weight markers (The authors included MW in source data, but not in figures).

Response:

We have complemented with MW indicators for all immunoblot panels in the manuscript.

3. For Fig. 1b and Extended Data Fig. 1b and 1d: the protein level of BMAL1 in Vps15LKO liver at ZT0 and ZT12 for these two experiments were inconsistent. There was no obvious difference in BMAL1 level at ZT0 or ZT12 in WT vs Vps15LKO livers from data shown in Fig. 1b/Extended 1b, but then in Extended 1d, BMAL1 level in Vps15LKO liver was lower in both cytosol and in nucleus at both time points. This discrepancy should be explained/resolved.

Response:

We thank the reviewer for the comment that helped us to improve presentation of the data. This comment resonates with **the point 2 of Reviewer 2**. The immunoblot analyses in **Fig.1b** and **Extended Data Fig. 1c** (in the revised manuscript) were performed on total protein extracts of liver tissue that are prepared with detergent (1%NP40), assuring efficient extraction. We validated this extraction protocol in our previous publications by detecting plasma membrane, nuclear, and mitochondrial proteins in immunoblots³⁻⁵. However, immunoblot analyses presented in **Extended Data Fig. 1e** (in the revised manuscript) were done with samples of cytosolic and nuclear fractions. In this protocol, the cytosolic proteins are extracted in hypotonic buffer without detergent and nuclear proteins are extracted with hypertonic buffer³⁶. We believe the observed differences in detection of BMAL1 protein in total protein extracts compared to enriched nuclear and cytosolic fractions are due to the differences in these extraction procedures.

4. Fig. 1e (and for the rest): In accordance with changing standards in the field toward better demonstration of all data points, the authors should replace bar graphs with scatter plots to show individual data points for quantifications.

Response:

We have re-plotted all the graphs in the manuscript as scatter plots in line with the guidelines.

5. Extended Data Fig. 1e: the authors stated that less CLOCK was in complex with BMAL1 in Vps15LKO livers because of lower level of CLOCK the co-IP with BMAL1. However, there was less CLOCK in the lysate input as well. Did the author take this into account when performing quantification?

Response:

We thank the reviewer for their help in improving the clarity of our manuscript. In this assay, we used equal amounts of protein extracts for all conditions. In the quantification, we normalized the signal of co-immunoprecipitated CLOCK to the signal of the BMAL1 in the respective direct IP. We modified the text to indicate that inhibition of BMAL1-CLOCK transcriptional activity in mouse livers upon chronic deletion of Vps15 is likely due to decreased protein levels of both proteins. This is reflected in lower levels of their transcriptional complex. Moreover, in response to **Reviewer 1 (point 8)**, in this revised manuscript we now provide data that shows nuclear BMAL1 protein turnover is unchanged

and BMAL1-CLOCK complex formation is unaffected in MEF cells upon acute depletion of Vps15 (R-Fig. 14a,b). These are further backed by findings *in vivo* in livers of Vps15iKO mice that show similar levels of nuclear BMAL1 and CLOCK proteins (R-Fig. 17a) and unmodified chromatin enrichment of BMAL1 upon acute Vps15 depletion (*Extended Data Fig. 9h in the revised manuscript*). Although we cannot rule out secondary effects of chronic Vps15 depletion on BMAL1-CLOCK protein levels, findings in cells and *in vivo* in the models of acute Vps15 depletion together with data of luciferase assays support our conclusion that Vps15 has a novel role as a transcriptional co-activator of BMAL1-CLOCK.

6. *Extended Data Fig. 1e: it would be more helpful if the authors could add arrow heads pointing to the staining of BMAL1 in the nucleus.*

Response:

We thank the reviewer for the suggestion that improved the presentation of the findings (see R-Fig.18 in response to **the Reviewer 2, point 2**).

7. *Fig. 3b: Images shown here are missing scale bars, and it would be convenient for the readers if the authors could indicate what color is corresponding to which signal (like writing DAPI in blue font and PLA in red font, for example). In this particular experiment, the GFP fluorescence was only observed in the nucleus of the cell, which is strange, as it should be in cytosol as well. Also, co-expression of GFP from another plasmid is actually not helpful to visualize cells successfully transfected with the FLAG-BMAL1, since some cells may only receive one of the plasmids.*

Response:

We thank the reviewer for these helpful insights that added clarity to the figure. We have labelled all PLA figures following the suggestion of the reviewer (R-Fig. 13, R-Fig. 28, R-Fig. 29). Also, we apologize if the quality of the image was not sufficient to see GFP+ signal. In the updated PLA figures, we employed co-transfection approach with GFP and, as expected, GFP is present both in cytosol and in nucleus of transfected HEK293T cells (R-Fig. 13a, R-Fig. 28a, R-Fig. 29c). Moreover, the plasmid co-expression approach is a common practice and there is no reason for selective uptake of one plasmid versus another when the transfection mix is prepared with two vectors added to the same tube at the same time (that is the case in our protocol). The efficacy of this approach could be seen in luciferase assays or in co-IP analyses when multiple plasmids are co-transfected. This approach is also efficient in PLA assay, as could be seen on R-Fig.13a, in the field of about 10 cells the proximity dots are apparent and are higher in number in GFP+ cells.

8. *Fig. 3e: the Vps15mut is a point mutant of E200R, but does that single mutation explain its different migration from the WT form in this blot? The WT and mutant proteins showed similar migration in Fig. 3d (FLAG blots).*

Response:

We apologize for omitting this technical information that is now included in the material and method section. The adenoviral vector construct of Vps15E200R is tagged with 1xFlag while Vps15wt has been tagged with 6xHis-V5 tag. This difference in tag length and charge explains the difference in protein migration. To be sure that the effect of Vps15 overexpression on BMAL1-CLOCK transcriptional activity is not impacted by a type of tag, in luciferase assays, we have employed Vps15 wt-1xFlag and Vps15E200R-1xFlag, both showing similar expression levels and capacity to co-activate BMAL-CLOCK transcription factor complex in HEK293T cells (R-Fig. 20).

9. *For numbers with decimal on the y-axis of many plots, periods should be used instead of commas.*

Response:

We modified the labelling on all plots.

Reviewer #4

*In this manuscript (NCB-A48597), "Class3 PI3K transcriptionally co-activates circadian clock (CC) to drive purine synthesis" Alkhoury et al., have utilized various approaches in several model systems to provide evidence for VPS15 to be a co-activator of BMAL1 and regulate metabolism, specifically purine synthesis. **Experiments are well controlled, and presentation of results are clear. The statistics is appropriate.** As a general comment, I previous investigations have provided evidence for links between other classes of PI3K signaling (but not class 3) and 'Clock' system, thus somewhat diluting the. Additionally, the present study raises several concerns/questions (see below).*

We would like to thank the reviewer for their appreciation of our work, for noting its integrative aspect in using different experimental approaches and the importance of our findings for the field. We are grateful for the reviewer's comments that allowed us to significantly improve our work.

Major Points:

1.The manuscript initiates (Fig.1a, Extended Data Fig. 1a) and finishes (Extended Data Fig.7e) with analyzing the expression (transcripts) of various CC-components. It is clear from the results with VPS15iLKO mice that: mutation of VPS15 kills the CC-oscillator, with significant reduction in the expression of all the CC genes including BMAL1 and CLOCK. This is an extremely crucial observation and raises the question- Why in absence of VPS15 the RORE-dependent transcription of BMAL1 and CLOCK is reduced?

Response:

We thank reviewer for pointing to this functional link between ROR α and class 3 PI3K/Vps15 that we believe warrants in-depth future investigation. Given that ROR α is inhibited by NCoR1 transcriptional repressor and its inhibition is especially evident in the chronic model of class 3 PI3K inactivation in liver (Vps15LKO mice), it could be due to accumulation of NCoR1 and Hdac3 repressors that we reported in livers of Vps15LKO mice³⁷. Together with the team of Prof. Komatsu³⁸, we demonstrated that these transcriptional repressors are the targets of selective autophagy and their accumulation in autophagy defective hepatocytes results in inhibition of nuclear receptor PPAR α transcriptional activity. Most likely, in livers of Vps15LKO mice, the repressive effect of accumulated NCoR1/Hdac3 is not restricted to PPAR α and could contribute to ROR α dysfunction. It is also important to note, that although acute deletion of Vps15 in cells and in livers of mice results in a visible autophagy block, it does not yet lead to accumulation of NCoR1 and Hdac3 to the levels seen in the models of chronic deletion of Vps15 (Vps15LKO mice). This is potentially one of the explanations of differences in clock phenotypes between models of acute and chronic Vps15 depletion. [Redacted].

2. Continuing with VPS15iLKO system: How does the authors explain the reduced nuclear accumulation of BMAL1. Is this reduction in nuclear accumulation also happens for other CC-components too? Could the authors speculate on the mechanism for this?

Response:

The reviewer points to an important question of BMAL1 protein turnover in the models of chronic vs acute Vps15 depletion. As also noted by **the Reviewer 1**, chronic deletion of Vps15 in livers of Vps15LKO mice results in decreased levels of BMAL1 protein. This is not evident in the models of acute Vps15 depletion both *in vivo* (TtrCre⁺;Vps15^{ff} mice) and *in vitro* (Vps15^{ff} MEFs). Following the advice of this reviewer and **the Reviewer 1** (point 8) we have complemented these initial findings by additional analyses:

1) In this revised manuscript, we show that BMAL1 and CLOCK proteins have a similar half-life of 10.4hrs revealed in analyses of protein turnover in cycloheximide treated control and Vps15-null MEFs (**R-Fig. 14a**, corresponding to **Extended Data Fig. 2i** in the revised manuscript).

2) We also studied the BMAL1-CLOCK complex in cells with depleted Vps15 or upon its overexpression. We now show that neither Vps15 depletion (**R-Fig. 14b**, corresponding to **Fig. 4a** in the revised manuscript) nor its dose-course overexpression (**R-Fig. 14c**, corresponding to **Fig. 4c** in the revised manuscript) impact BMAL1-CLOCK complex in MEF and HEK293T cells. To note, dose course overexpression of Vps15 resulted in its respective increased interaction with the BMAL1-CLOCK complex (**R-Fig. 14c**, corresponding to **Fig. 4c** in the revised manuscript). This is also correlated with increased BMAL1-CLOCK transcriptional activity measured in luciferase assay with E-box-LUC reporter; which we provide in this revised version in response to the comment of **the Reviewer 2** (**R-Fig. 20a**, corresponding to **Fig. 5a** in the revised manuscript).

3) We performed immunoblot analyses of BMAL1 and CLOCK levels in total and nuclear extracts of livers Vps15iLKO mice collected every 4 hours (**R-Fig. 16a**). The immunoblot analyses in total liver extracts of Vps15iLKO mice showed that deletion of Vps15 resulted largely inhibited rhythmic expression of REV-ERB α and BMAL1 proteins (**R-Fig. 16b**, corresponding to **Extended Data Fig. 9b** in the revised manuscript). Next, we analyzed levels of BMAL1, CLOCK and REV-ERB α in the extracts of nuclear soluble fraction (**R-Fig. 17**, corresponding to **Extended Data Fig. 9c** in the revised manuscript). As expected, the pattern of expression of REV-ERB α protein in nuclear fractions was similar to that of total protein extracts (**R-Fig.17a**, corresponding to **Extended Data Fig. 9c** in the revised manuscript). Notably, we did not observe a significant difference between genotypes in nuclear protein levels of BMAL1 and CLOCK (**R-Fig.17a**, corresponding to **Extended Data Fig. 9c** in the revised manuscript). The analyses of other clock components, including ROR α , Per1, Per2, Cry1, Cry2 did not reveal major differences between genotypes (**R-Fig.17b**, data not included in the revised manuscript). The mild accumulation of Cry1 that we have observed in Vps15iLKO mice is consistent with autophagy block and is in line with its role for Cry1 protein degradation³⁹. We have also observed more pronounced animal-to-animal variability in this new dataset of control and Vps15iLKO mice compared to the analyses presented in the initial version of the manuscript. Likely, it stems from pooling samples of three separate mouse cohorts that were independently treated with tamoxifen and sacrificed as a high number of animals was required for these expression analyses.

R-Figure 14. a, Immunoblot analysis of nuclear soluble fractions of Vps15-depleted (CRE) or control (GFP) MEFs treated with cycloheximide for indicated times. Densitometric analyses of BMAL1 and CLOCK proteins normalized to total protein (TGX gels) are presented as fold over a vehicle treated condition ($n=3$). **b**, Immunoblot analyses of BMAL1-containing immunoprecipitates using anti-CLOCK and anti-BMAL1 antibody from total protein extract of Vps15-depleted (CRE) or control (GFP) MEFs. Quantification of CLOCK protein in BMAL1 immunoprecipitates normalized to BMAL1 levels is presented. Data are means \pm SEM ($n=3$). **c**, Immunoblot analyses of BMAL1-containing immunoprecipitates from total protein extract of HEK293T cells transfected with increasing amounts of Vps15wt-expressing vector. The quantification of CLOCK protein in BMAL1 immunoprecipitates normalized to BMAL1 is presented as fold difference to control empty vector transfected cells. Data are means \pm SEM ($n=2-3$). **d**, BMAL1 and RNA Pol2 phSer5 recruitment to promoter of indicated genes by ChIP-qPCR in MEFs transduced with adenoviral vectors expressing Vps15wt or GFP as a control. Cells were Dexamethasone-synchronized and collected 24 hours after synchronization. Data are presented as log₂ fold change enrichment over IgG \pm SEM, $n=8$, $P<0.05$ # vs IgG, 2-tailed, unpaired Student's t test.

3. The study demonstrates a protein-protein interaction between the BMAL1 and VPS15. It will be a great help if the region or the residues in both BMAL1 and VPS15 which helps in the interaction could be determined. Is it a direct interaction or indirect?

Response:

We thank reviewer for this insightful suggestion that resonated with **the point 6 of the Reviewer 2** and which helped us further reinforce our main finding of Vps15 as a co-activator of BMAL1-CLOCK transcription complex.

1) To investigate the domain of BMAL1 involved in its interaction with Vps15, we employed a panel of BMAL1 deletion mutants that were reported in *Tamaru et al., Plos Biol, 2009*²⁹ (R-Fig. 21a, corresponding to Fig. 4i in the revised manuscript). We discovered that an interaction of BMAL1 with either CLOCK or Vps15 occurs through different regions of BMAL1 (R-Fig. 21b, corresponding to Fig. 4i in the revised manuscript). CLOCK interaction was abrogated by deletion of regions in BMAL1 containing PAS-A and PAC domains. At the same time, only deletion of the C-terminal region of BMAL1 was detrimental for its interaction with Vps15 while having no effect on complex with CLOCK (R-Fig. 21b, corresponding to Fig. 4i in the revised manuscript). This finding is in line with C-terminal region of BMAL1 reported to be necessary for its interaction with other co-factors such as Cry1 and CBP^{14,30,31}.

R-Figure 21. a, Schematic representation of HIS-tagged BMAL1 deletion mutants. **b**, Immunoblot analyses of BMAL1-containing immunoprecipitates using anti-Vps15 and anti-CLOCK antibody from total protein extracts of HEK293T cells expressing deletion mutants of BMAL1.

2) To investigate the domains of Vps15 involved in interaction with BMAL1, we generated a panel of truncated Vps15 mutants by deleting the functional domains defined by cryoEM analyses of Vps15-Vps34 complex³² (R-Fig. 22a, corresponding to Extended Data Fig. 6f in the revised manuscript). Co-immunoprecipitation analyses revealed the specific interaction between BMAL1 and all fragments of Vps15 (R-Fig. 22b, corresponding to Extended Data Fig. 6f in the revised manuscript). Thus, we concluded that HEAT domain of Vps15 is likely necessary for its interaction with BMAL1.

R-Figure 22. a, Schematic representation of Flag-tagged Vps15 deletion mutants. **b**, Immunoblot analyses of BMAL1-His-containing immunoprecipitates using anti-Flag antibody from total protein extracts of HEK293T cells expressing deletion mutants of Vps15 (Flag tagged) together with BMAL1-His full-length protein.

4. The study claims VPS15 as a co-activator of BMAL1-dependent transcription for a certain class of genes. Considering BMAL1/CLOCK are pioneer transcription factors (in mouse liver), how VPS15 co-activates this transcriptional activity? This is significant given the fold enrichment for VPS15 in ChIP assays seem to be manifold less than that of BMAL1? Can authors determine the stoichiometry?

Response:

Indeed, BMAL1-CLOCK are known to mediate circadian DNA accessibility for binding of other transcription factors, thus acting as circadian pioneering factors⁴⁰. Notably, Vps15 does not have classical DNA binding domain and our findings of ChIP-qPCR and ChIP-Seq with Vps15 show much lower enrichment compared to BMAL1. The latter is consistent with Vps15 binding to a transcription factor and transcriptional machinery likely away from chromatin. Our findings that chromatin binding sites of Vps15 also overlap with BMAL1 open to future investigations of Vps15 recruitment on chromatin around the clock especially in light that we find it interacting with RNA Pol2 (R-Fig.12, above) and its localization in *de novo* transcription sites (R-Fig.11, above).

Moreover, in line with the suggestions of **this reviewer and the Reviewer 2**, in this revised manuscript, we provide additional mechanistic insights of how Vps15 is involved in BMAL1-CLOCK co-activation (**point 6 of the Reviewer 2**). We find TAD domain of BMAL1 and HEAT domain of Vps15 necessary for their interaction. Given that Vps15 binds to C-terminal domain of BMAL1, the same region as Cry1 binding, we also asked whether Vps15 co-expression with Cry1 would interfere with its repressive action on BMAL1-CLOCK transcriptional complex. As expected, expression of Vps15 or Vps15mut1 that does not interact with Vps34 co-activated BMAL1-CLOCK while co-expression of Cry1 repressed BMAL1-CLOCK transcriptional activity (R-Fig. 23, corresponding to Fig. 5b in the revised manuscript). Notably, co-expression of Vps15 together with Cry1 significantly rescued the transcriptional activity of BMAL1-CLOCK compared to the levels observed in empty vector transfected cells (R-Fig. 23, corresponding to Fig. 5b in the revised manuscript). In sum, our findings show that Vps15 co-activates BMAL1-CLOCK and could interfere with repression by Cry1.

R-Figure 23. Luciferase assay in HEK293T cells co-transfected with E-box-Luc reporter construct together with empty vector or plasmids expressing BMAL1 and CLOCK with or without CRY1, Vps15wt or Vps15mut1. The relative luminescence is presented as fold difference over E-box-Luc only transfected cells (complemented with ev instead of BMAL1-CLOCK). Data are means \pm SEM, n=4. #: P<0.05, 2-tailed, unpaired Student's t test. Representative immunoblot shows the expression levels of CRY1, BMAL1, CLOCK and Vps15 proteins. GAPDH served as loading control.

5. The ChIP assays were performed at ZT6, which is considered to be the peak of BMAL1 binding to DNA. But, it is well known that enhancer remodeling starts earlier with production of eRNAs. Does VPS15 plays any role in this process for its target genes?

Response:

We thank the reviewer for an interesting suggestion to test whether Vps15 could have an impact on the enhancer activity of BMAL1 that we did not consider. We did not perform GRO-Seq adapted for detection of fast degraded eRNA transcripts and, thus, we are fully aware of the limitations of our approach of using the bulk RNA-Seq data. Nevertheless, to investigate the question of the reviewer, we overlaid the H3K27ac and BMAL1 ChIP-seq with the bulk RNA-seq performed at ZT6. We used identified eRNAs from Fang et al., Cell, 2014⁴¹ to test where these eRNAs overlap with the transcriptome and cistrome of BMAL1 and H3K27ac in the control livers and in livers of Vps15LKO mice. Initially, we decided to focus on three candidate genes to investigate Vps15 dependent BMAL1 enhancer binding. While no eRNA transcripts were identified +/- 12kb or 50kb surrounding *Dbp* or *Ppat* genes, we did discover a set of identified eRNAs published in Fang et al., Cell, 2014⁴¹ that were present upstream of *Nr1d1* (R-Fig. 32). In line with the limitations of bulk RNA-Seq for the identification of eRNAs, we did not detect in this region more than one or two transcripts based on the coverage map. Despite this, we observed BMAL1 and H3K27ac binding roughly 6kb upstream of the *Nr1d1* TSS (R-Fig. 32a). This is indicative of enhancer binding, and there is substantial decreased H3K27ac signal as well as binding of BMAL1 at this region in the Vps15LKO samples. However, an overlap with H3K4me1 in our data set would be critical to better validate the Vps15-dependent effects on enhancer mediated transcription. While these preliminary observations suggest that enhancer binding/activity of BMAL1 is diminished in the absence of Vps15, we cannot conclude on the implication of Vps15 for eRNA mediated transcriptional regulation. Furthermore, most likely lower BMAL1 binding that is observed in livers of Vps15LKO mice is due to decreased BMAL1 protein levels observed upon chronic Vps15 depletion. Future works are needed to directly assess the role of Vps15 on eRNA production by using GRO-Seq in the models of acute depletion of Vps15, a better adapted methodology and experimental model to address these questions. We thank the reviewer for pointing to this direction that we likely pursue in future.

6. Fig. 3h: Analyses of VPS15 genomic recruitment indicates that its recruitment seems to be far more downstream of the TSS rather than upstream. Is VPS15 a part of the elongating RNA PolII complex rather than initiation complex? This distinction is necessary to provide possible mechanism of its suggested role as a co-activator.

Response:

We thank the reviewer for this insightful comment that made us investigate link between RNA Pol2 and Vps15. Indeed, although the peak of binding for Vps15 surrounds TSS, it shows an evident shoulder in the region of +3kb. Thus, we investigated the involvement of class 3 PI3K with RNA Pol2-driven nuclear transcription. For this, we have labelled *de novo* transcription sites with BrUTP in MEF cells with ectopically expressed Flag-tagged Vps15 and Vps34. Strikingly, both Vps15 and Vps34 co-localize with BrUTP puncta (R-Fig. 11a corresponding to Fig. 3c in the revised manuscript). Next, we asked if Vps15 co-localization with *de novo* transcription sites extends to endogenous Vps15 in other cell type such as hepatocytes. As presented on R-Fig. 11b (corresponding to Fig. 3e in the revised manuscript), endogenous Vps15 readily detected in nuclei and it co-localized with *de novo* transcription sites labelled with BrUTP and RNA Pol2 phospho-Ser5 (transcriptional elongation form). Notably, the triple positive BrUTP/RNA Pol2/Vps15 sites were detected in hepatocyte nuclei outside of nucleoli (R-Fig. 11c corresponding to Fig. 3e in the revised manuscript). Unfortunately,

R-Figure 32. RNA seq and ChIP-seq of BMAL1 and H3K27ac in WT and Vps15LKO livers overlaid on a, *Nr1d1*, b, *Dbp*, and c, *Ppat* using IGV genome browser. Identified eRNAs shown in tables to the right of each panel for each gene region. eRNAs identified in Fang et al., *Cell*, 2014 highlighted in yellow.

we could not perform these analyses with Vps34 as we lack antibodies that would specifically recognize endogenous Vps34 in immunofluorescence microscopy. Next, we asked whether Vps15 and Vps34 interact with RNA Pol2. Initially, we detected specific binding of ectopically expressed Vps15 and Vps34 with endogenous RNA Pol2 in HEK293T cells (**R-Fig. 12a**, corresponding to **Fig. 3d** in the revised manuscript). Next, we extended this observation to endogenous Vps15 and RNA Pol2 in MEFs (**R-Fig. 12b**, corresponding to **Fig. 3f** in the revised manuscript). To get further insight into the requirement of Vps34 for interaction between Vps15 and RNA Pol2, we also employed two different point

R-Figure 11. a, Transcription assay with ectopically expressed Vps15-Flag and Vps34-Flag proteins in MEFs cells. 24hrs post-transfection MEFs were incubated with BrUTP to label *de novo* transcription sites. For co-localization, immunofluorescent microscopy was performed with anti-Flag and anti-BrUTP antibodies. Scale bar 5 μ m. **b**, Immunofluorescent microscopy analyses of co-localization of endogenous RNA Pol2 phSer5, Vps15 and *de novo* transcription sites labelled with BrUTP in primary hepatocytes. Scale bar 10 μ m and 5 μ m (inset).

R-Figure 12. a, Immunoblot analyses of Flag-containing immunoprecipitates from nuclear soluble fraction of HEK293T cells transfected with Vps15-Flag and Vps34-Flag expressing vectors. Co-immunoprecipitation of RNA Pol2 with ectopically expressed Vps15 and Vps34 proteins is evidenced in immunoblot. ev, empty vector transfected HEK293T condition used as a control of non-specific binding. **b**, Immunoblot analyses of Vps15-containing immunoprecipitates using anti-RNA Pol2 antibody from nuclear soluble fractions of MEF cells. **c**, Immunoblot analyses of Vps15-containing complexes immunoprecipitated from HEK293T cells transiently transfected with constructs of Vps15wt, Vps15mut 1 and Vps15mut 2 (ev, empty vector). Cells were collected 48 hours post-transfection, Vps15 was immunoprecipitated using anti-Flag antibody.

mutants of Vps15 (E200R and D165R) that do not form the complex with Vps34. Notably, the mutations that disrupted the interaction between Vps15 and Vps34 did not affect binding of RNA Pol2 or its Ser5 phosphorylated active form (**R-Fig. 12c**, corresponding to **Fig. 3g** in the revised manuscript). This observation is in favor of the scenario that the interaction of Vps15 and RNA Pol2 does not rely on Vps34.

Altogether, these analyses further reinforced our initial findings of a nuclear pool of Vps15 and Vps34 as well as provided new evidence of Vps15 interaction with RNA Pol2 and their co-localization at *de novo* transcription sites. Our findings are in line with an early published work in yeast showing that Vps factors including Vps15 and Vps34 are involved in RNA Pol2-mediated transcription elongation thus pointing to evolutionary conserved role of nuclear class 3 PI3K in all eukaryotes¹.

7.Fig.3c: This experiment describes the result of luciferase assay from the transfected cells. This is not an actual classically described 'in vitro transcription assay' as mentioned in the text.

Response:

We apologize for this misleading formulation that we have corrected in the revised manuscript. To note, following the suggestions of **the Reviewer 2 (point 6, R-Fig. 20)**, we now provide additional luciferase assays upon Vps15 overexpression that largely support our main finding of Vps15 acting as a co-activator of BMAL1-CLOCK.

8.At least some microscopic evidence is necessary (along with presented western blots) to demonstrate that autophagy is indeed affected by VPS15 mutation.

Response:

We apologize for omitting these controls of Vps15 inactivation. This comment of the reviewer also resonates with **point 3 of Reviewer 1**. In the revised version of the manuscript, in addition to presented expression analyses, we now provide multiple evidence that Vps15 depletion results in Vps34 lipid kinase inhibition and manifests in autophagy defects. Notably, Vps15-null MEFs show decreased levels of PI3P (**R-Fig. 33a**, corresponding to **Extended Data Fig. 2c** in the revised manuscript) and blocked autophagy (**R-Fig. 33b,c**, corresponding to **Extended Data Fig. 2d,e** in the revised manuscript). The latter is evidenced by accumulation of p62 granules in Vps15-null MEFs (**R-Fig. 33b**, corresponding to **Extended Data Fig. 2d** in the revised manuscript). In addition, we employed the gold standard reporter to monitor autophagy flux, a tandem fluorescent-tagged LC3 (mRFP-EGFP-LC3)⁴² which we expressed using adenoviral vectors in control and Vps15-depleted MEFs. As can be seen on **R-Fig. 33c** (corresponding to **Extended Data Fig. 2e** in the revised manuscript) Vps15-depleted cells already in nutrient-rich media accumulated large size dysfunctional autophagolysosomes (co-localization of mRFP and EGFP in the non-acidic vacuoles; *open yellow triangles*). Importantly, while in control cells treatment with the amino acid withdrawal (2hr EBSS) resulted in increased number of active autolysosomes (red vacuolar structures corresponding to fused autophagosomes with the lysosomes characterized by acidic pH; *filled white triangles*), the Vps15-depleted MEFs were largely insensitive to the induction of autophagy (**R-Fig. 33c**, corresponding to **Extended Data Fig. 2e** in the revised manuscript). Altogether, these findings are in agreement with our published observations of defective autophagic degradation leading to profound proteostasis defects in Vps15-null cells^{4,5,37}.

R-Figure 33. **a**, PI3P detection with a recombinant protein HRS-RFP probe in Vps15-depleted (CRE) and control (GFP) MEFs 5 days post transduction with GFP and CRE expressing adenoviruses. The incubation with HRS-mut and the condition without the probe served as controls of non-specific binding of HRS-RFP probe. The quantification of HRS-RFP puncta per cell is presented \pm SEM, n=50-70 cells, P<0.05 # vs GFP, 2-tailed, unpaired Student's t test. Scale bar, 10 μ m. **b**, Immunofluorescent microscopy analyses of p62 in Vps15-depleted (CRE) MEFs 5 days post infection with GFP and CRE expressing adenoviruses. Scale bar, 20 μ m. **c**, Fluorescent microscopy analyses of mRFP-EGFP-LC3 reporter expressed in Vps15-depleted MEFs (CRE) or in control MEFs that were either kept in nutrient rich media or incubated with EBSS for 2 hours to induce autophagic flux. The number of red and yellow puncta per cell were quantified and the data presented as percentage of different puncta (left panel) or number of puncta per cell (right panel), n=20-100 cells, #: P<0.05, 2-tailed, unpaired Student's t test. Scale bar, 5 μ m.

9. If possible a demonstration of reduced purine incorporation in nucleotides in VPS15 and BMAL1 mutant cells will be extremely convincing.

Response:

We thank the reviewer for the suggestion that has further reinforced our study. To address this point, we have used the AML12 cell line (immortalized non-tumoral murine hepatocytes) and acutely depleted either Vps15 or BMAL1 using adenoviral vectors expressing shRNA Vps15 or shRNA BMAL1. As evident from immunoblot analyses, 3 days post-infection was sufficient to deplete BMAL1 (and CLOCK) as well as Vps15 (R-Fig. 34, corresponding to Fig. 6f in the revised manuscript). Treatment of cells with heavy labelled Gln-15N revealed significant inhibition of its incorporation into IMP (m+2), AMP (m+2) and ADP (m+2) in cells depleted of BMAL1 or Vps15 (R-Fig. 34, corresponding to Fig. 6f in the revised manuscript). These findings are in line with our observations of decreased IMP m+2 synthesis *in vivo* in livers of Vps15LKO mice (Fig. 6e in the revised manuscript).

R-Figure 34. Peak area of ¹⁵N-glutamine-amide incorporation into IMP, AMP and ADP. AML12 cells 72hours post transduction with viral vectors expressing shRNA BMAL1, shRNA Vps15 or GFP as a control were treated with ¹⁵N-glutamine-amide for 30min. Data are presented as means ±SEM, n=9, P<0.05 # vs GFP, 2-tailed, unpaired Student's t test. Representative immunoblot of total protein extract with indicated antibodies is shown as a control of depletion (right panel).

Minor Point: The scheme and its legend could be improved.

Response:

We have modified the summary schema of the study and its legend by incorporating the novel findings including interaction of Vps15 with RNA Pol2 (**R-Fig. 35** corresponding to **Fig. 6I** in the revised manuscript).

R-Figure 35. Schema presenting the functions of class 3 PI3K in cytosol in trafficking to the lysosome and in nuclear gene transcription *de novo* and as a co-activator of the core circadian transcription factor complex BMAL1-CLOCK for rhythmic *de novo* purine synthesis (created with BioRender.com).

References

- 1 Gaur, N. A. *et al.* Vps factors are required for efficient transcription elongation in budding yeast. *Genetics* **193**, 829-851, doi:10.1534/genetics.112.146308 (2013).
- 2 Bunney, T. D. *et al.* Association of phosphatidylinositol 3-kinase with nuclear transcription sites in higher plants. *Plant Cell* **12**, 1679-1688, doi:10.1105/tpc.12.9.1679 (2000).
- 3 Iershov, A. *et al.* The class 3 PI3K coordinates autophagy and mitochondrial lipid catabolism by controlling nuclear receptor PPAR α . *Nature communications* **10**, 1566, doi:10.1038/s41467-019-09598-9 (2019).
- 4 Nemazanyy, I. *et al.* Class III PI3K regulates organismal glucose homeostasis by providing negative feedback on hepatic insulin signalling. *Nature communications* **6**, 8283, doi:10.1038/ncomms9283 (2015).
- 5 Nemazanyy, I. *et al.* Defects of Vps15 in skeletal muscles lead to autophagic vacuolar myopathy and lysosomal disease. *EMBO molecular medicine* **5**, 870-890, doi:10.1002/emmm.201202057 (2013).
- 6 Kim, J. *et al.* Differential regulation of distinct Vps34 complexes by AMPK in nutrient stress and autophagy. *Cell* **152**, 290-303 (2013).
- 7 Murray, J. T., Panaretou, C., Stenmark, H., Miaczynska, M. & Backer, J. M. Role of Rab5 in the recruitment of hVps34/p150 to the early endosome. *Traffic* **3**, 416-427, doi:10.1034/j.1600-0854.2002.30605.x (2002).
- 8 Panaretou, C., Domin, J., Cockcroft, S. & Waterfield, M. D. Characterization of p150, an adaptor protein for the human phosphatidylinositol (PtdIns) 3-kinase. Substrate presentation by phosphatidylinositol transfer protein to the p150.PtdIns 3-kinase complex. *J.Biol.Chem.* **272**, 2477-2485 (1997).
- 9 Stack, J. H., DeWald, D. B., Takegawa, K. & Emr, S. D. Vesicle-mediated protein transport: regulatory interactions between the Vps15 protein kinase and the Vps34 PtdIns 3-kinase essential for protein sorting to the vacuole in yeast. *J.Cell Biol.* **129**, 321-334 (1995).
- 10 Stack, J. H., Herman, P. K., Schu, P. V. & Emr, S. D. A membrane-associated complex containing the Vps15 protein kinase and the Vps34 PI 3-kinase is essential for protein sorting to the yeast lysosome-like vacuole. *EMBO J.* **12**, 2195-2204 (1993).
- 11 Rostislavleva, K. *et al.* Structure and flexibility of the endosomal Vps34 complex reveals the basis of its function on membranes. *Science* **350**, aac7365, doi:10.1126/science.aac7365 (2015).
- 12 Herman, P. K., Stack, J. H., DeModena, J. A. & Emr, S. D. A novel protein kinase homolog essential for protein sorting to the yeast lysosome-like vacuole. *Cell* **64**, 425-437 (1991).
- 13 Herman, P. K., Stack, J. H. & Emr, S. D. A genetic and structural analysis of the yeast Vps15 protein kinase: evidence for a direct role of Vps15p in vacuolar protein delivery. *EMBO J* **10**, 4049-4060, doi:10.1002/j.1460-2075.1991.tb04981.x (1991).
- 14 Lee, Y. *et al.* Coactivation of the CLOCK-BMAL1 complex by CBP mediates resetting of the circadian clock. *J Cell Sci* **123**, 3547-3557, doi:10.1242/jcs.070300 (2010).
- 15 Kwon, I. *et al.* BMAL1 shuttling controls transactivation and degradation of the CLOCK/BMAL1 heterodimer. *Mol Cell Biol* **26**, 7318-7330, doi:10.1128/MCB.00337-06 (2006).
- 16 Zheng, X. *et al.* RAE1 promotes BMAL1 shuttling and regulates degradation and activity of CLOCK: BMAL1 heterodimer. *Cell Death Dis* **10**, 62, doi:10.1038/s41419-019-1346-2 (2019).
- 17 Sawasdichai, A., Chen, H. T., Abdul Hamid, N., Jayaraman, P. S. & Gaston, K. In situ subcellular fractionation of adherent and non-adherent mammalian cells. *Journal of visualized experiments : JoVE*, doi:10.3791/1958 (2010).
- 18 Sinturel, F. *et al.* Circadian hepatocyte clocks keep synchrony in the absence of a master pacemaker in the suprachiasmatic nucleus or other extrahepatic clocks. *Genes Dev* **35**, 329-334, doi:10.1101/gad.346460.120 (2021).
- 19 Koronowski, K. B. *et al.* Defining the Independence of the Liver Circadian Clock. *Cell* **177**, 1448-1462 e1414, doi:10.1016/j.cell.2019.04.025 (2019).
- 20 Hepler, C. *et al.* Time-restricted feeding mitigates obesity through adipocyte thermogenesis. *Science* **378**, 276-284, doi:10.1126/science.abl8007 (2022).
- 21 Ronan, B. *et al.* A highly potent and selective Vps34 inhibitor alters vesicle trafficking and autophagy. *Nat Chem Biol* **10**, 1013-1019, doi:10.1038/nchembio.1681 (2014).
- 22 Dowdle, W. E. *et al.* Selective VPS34 inhibitor blocks autophagy and uncovers a role for NCOA4 in ferritin degradation and iron homeostasis in vivo. *Nat Cell Biol* **16**, 1069-1079, doi:10.1038/ncb3053 (2014).
- 23 Bago, R. *et al.* Characterization of VPS34-IN1, a selective inhibitor of Vps34, reveals that the phosphatidylinositol 3-phosphate-binding SGK3 protein kinase is a downstream target of class III phosphoinositide 3-kinase. *The Biochemical journal* **463**, 413-427, doi:10.1042/BJ20140889 (2014).
- 24 Horiguchi, M. *et al.* Rhythmic control of the ARF-MDM2 pathway by ATF4 underlies circadian accumulation of p53 in malignant cells. *Cancer research* **73**, 2639-2649, doi:10.1158/0008-5472.CAN-12-2492 (2013).
- 25 Besing, R. C. *et al.* Circadian rhythmicity of active GSK3 isoforms modulates molecular clock gene rhythms in the suprachiasmatic nucleus. *J Biol Rhythms* **30**, 155-160, doi:10.1177/0748730415573167 (2015).
- 26 Tamai, T. K. *et al.* Identification of circadian clock modulators from existing drugs. *EMBO molecular medicine* **10**, doi:10.15252/emmm.201708724 (2018).
- 27 Hirota, T. *et al.* High-throughput chemical screen identifies a novel potent modulator of cellular circadian rhythms and reveals CK1 α as a clock regulatory kinase. *PLoS Biol* **8**, e1000559, doi:10.1371/journal.pbio.1000559 (2010).
- 28 Katsioudi, G. *et al.* A conditional Smg6 mutant mouse model reveals circadian clock regulation through the nonsense-mediated mRNA decay pathway. *Sci Adv* **9**, eade2828, doi:10.1126/sciadv.ade2828 (2023).
- 29 Tamaru, T. *et al.* CRY Drives Cyclic CK2-Mediated BMAL1 Phosphorylation to Control the Mammalian

- Circadian Clock. *PLoS Biol* **13**, e1002293, doi:10.1371/journal.pbio.1002293 (2015).
- 30 Kiyohara, Y. B. *et al.* The BMAL1 C terminus regulates the circadian transcription feedback loop. *Proc Natl Acad Sci U S A* **103**, 10074-10079, doi:10.1073/pnas.0601416103 (2006).
 - 31 Sato, T. K. *et al.* Feedback repression is required for mammalian circadian clock function. *Nat Genet* **38**, 312-319, doi:10.1038/ng1745 (2006).
 - 32 Baskaran, S. *et al.* Architecture and dynamics of the autophagic phosphatidylinositol 3-kinase complex. *Elife* **3**, doi:10.7554/eLife.05115 (2014).
 - 33 Robitaille, A. M. *et al.* Quantitative phosphoproteomics reveal mTORC1 activates de novo pyrimidine synthesis. *Science* **339**, 1320-1323, doi:10.1126/science.1228771 (2013).
 - 34 Ben-Sahra, I., Hoxhaj, G., Ricourt, S. J. H., Asara, J. M. & Manning, B. D. mTORC1 induces purine synthesis through control of the mitochondrial tetrahydrofolate cycle. *Science* **351**, 728-733, doi:10.1126/science.aad0489 (2016).
 - 35 French, J. B. *et al.* Spatial colocalization and functional link of purinosomes with mitochondria. *Science* **351**, 733-737, doi:10.1126/science.aac6054 (2016).
 - 36 Zhang, X., Shu, L., Hosoi, H., Murti, K. G. & Houghton, P. J. Predominant nuclear localization of mammalian target of rapamycin in normal and malignant cells in culture. *J Biol Chem* **277**, 28127-28134, doi:10.1074/jbc.M202625200 (2002).
 - 37 Iershov, A. *et al.* The class 3 PI3K coordinates autophagy and mitochondrial lipid catabolism by controlling nuclear receptor PPAR α . *Nature communications* **10**, doi:10.1038/s41467-019-09598-9 (2019).
 - 38 Saito, T. *et al.* Autophagy regulates lipid metabolism through selective turnover of NCoR1. *Nature communications* **10**, 1567, doi:10.1038/s41467-019-08829-3 (2019).
 - 39 Toledo, M. *et al.* Autophagy Regulates the Liver Clock and Glucose Metabolism by Degrading CRY1. *Cell Metab* **28**, 268-281, e264, doi:10.1016/j.cmet.2018.05.023 (2018).
 - 40 Menet, J. S., Pescatore, S. & Rosbash, M. CLOCK:BMAL1 is a pioneer-like transcription factor. *Genes Dev* **28**, 8-13, doi:10.1101/gad.228536.113 (2014).
 - 41 Fang, B. *et al.* Circadian enhancers coordinate multiple phases of rhythmic gene transcription in vivo. *Cell* **159**, 1140-1152, doi:10.1016/j.cell.2014.10.022 (2014).
 - 42 Klionsky, D. J. *et al.* Guidelines for the use and interpretation of assays for monitoring autophagy (4th edition)(1). *Autophagy* **17**, 1-382, doi:10.1080/15548627.2020.1797280 (2021).

Decision Letter, first revision:

*Please delete the link to your author homepage if you wish to forward this email to co-authors.

Dear Ganna,

Thank you for your patience while your revised manuscript, "Class 3 PI3K participates in nuclear gene transcription and co-activates the circadian clock to promote de novo purine synthesis", was evaluated by the original reviewers.

As you will see from their comments (attached for Rev#2 and otherwise copied below), all the reviewers appreciated the strong revision efforts and they remain supportive of the work. They have raised a few comments that are consistent with their concerns from the first round of review and pertain to the revision data. In particular, the nomenclature describing PAS-A, PAS-B and PAC motifs needs to be corrected, as per Rev#2, and Rev#3 has a remaining important question about the concentrations used for the Vps34 inhibitor studies.

We continue to be very interested in the study. As you know, our policy is to limit all manuscripts to a single round of major experimental revision in order to limit the overall time spent in peer review. However, as the reviewers' points are relatively minor, we are open to a final round of minor revisions, as we believe that their concerns should be addressed before we can consider publication in Nature Cell Biology.

Please let us know if you wish to discuss the revisions further. In our view, resolving the final points from Revs#2 and #3 is important, in particular Rev#3's question about the concentrations used for the Vps34 inhibitors.

Finally, as before, please pay close attention to our guidelines on statistical and methodological reporting (listed below) as failure to do so may delay the reconsideration of the revised manuscript. In particular please provide:

- a Supplementary Figure including unprocessed images of all gels/blots in the form of a multi-page pdf file. Please ensure that blots/gels are labeled and the sections presented in the figures are clearly indicated.
- a Supplementary Table including all numerical source data in Excel format, with data for different figures provided as different sheets within a single Excel file. The file should include source data giving rise to graphical representations and statistical descriptions in the paper and for all instances where the figures present representative experiments of multiple independent repeats, the source data of all repeats should be provided.

We therefore invite you to take these points into account when revising the manuscript. In addition, when preparing the revision please:

- ensure that it conforms to our format instructions and publication policies (see below and

<https://www.nature.com/nature/for-authors>).

- provide a point-by-point rebuttal to the full referee reports verbatim, as provided at the end of this letter.
- provide the completed Reporting Summary (found here <https://www.nature.com/documents/nr-reporting-summary.pdf>). This is essential for reconsideration of the manuscript and will be available to editors and referees in the event of peer review. For more information see <http://www.nature.com/authors/policies/availability.html> or contact me.

When submitting the revised version of your manuscript, please pay close attention to our [href="https://www.nature.com/nature-portfolio/editorial-policies/image-integrity">Digital Image Integrity Guidelines](https://www.nature.com/nature-portfolio/editorial-policies/image-integrity). and to the following points below:

Nature Cell Biology is committed to improving transparency in authorship. As part of our efforts in this direction, we are now requesting that all authors identified as 'corresponding author' on published papers create and link their Open Researcher and Contributor Identifier (ORCID) with their account on the Manuscript Tracking System (MTS), prior to acceptance. ORCID helps the scientific community achieve unambiguous attribution of all scholarly contributions. You can create and link your ORCID from the home page of the MTS by clicking on 'Modify my Springer Nature account'. For more information please visit www.springernature.com/orcid.

This journal strongly supports public availability of data. Please place the data used in your paper into a public data repository, or alternatively, present the data as Supplementary Information. If data can only be shared on request, please explain why in your Data Availability Statement, and also in the correspondence with your editor. Please note that for some data types, deposition in a public repository is mandatory - more information on our data deposition policies and available repositories appears below.

[Redacted]

*This url links to your confidential home page and associated information about manuscripts you may

have submitted or be reviewing for us. If you wish to forward this email to co-authors, please delete the link to your homepage.

We would like to receive the revision within four weeks. If submitted within this time period, reconsideration of the revised manuscript will not be affected by related studies published elsewhere, or accepted for publication in Nature Cell Biology in the meantime. We would be happy to consider a revision even after this timeframe, but in that case we will consider the published literature at the time of resubmission when assessing the file.

We hope that you will find our referees' comments and editorial guidance helpful. Please do not hesitate to contact me if there is anything you would like to discuss. Thank you again for considering NCB for your work,

Best wishes,

Melina

Melina Casadio, PhD
Senior Editor, Nature Cell Biology
ORCID ID: <https://orcid.org/0000-0003-2389-2243>

Reviewers' Comments:

Reviewer #1:

Remarks to the Author:

The revised manuscript by Alkhoury et. al. has clearly addressed all of our recommended revisions. The revised manuscript has added clarity, specificity, and robustness to the work. The revised manuscript has revealed and expanded a novel role for class III PI3K in the cell nucleus and linked VPS34 and VPS15 to circadian transcriptional-translation oscillation regulation. We strongly recommend publication in NCB.

Reviewer #2:

Remarks to the Author:

See attached comments

Reviewer #3:

Remarks to the Author:

In this revised manuscript, the authors provided a significant amount of high-quality new data to support their main findings for a novel role of class 3 PI3K in the nucleus and their proposed mechanistic model, where the nuclear pool of Vps15 serves as a transcriptional co-activator of the

BMAL1-CLOCK complex to couple energy homeostasis regulation with circadian clock transcriptional activity. Importantly, in the revised manuscript, the authors presented much more solid support for multiple major findings compared to the original version:

1. The role of Vps15 as a co-activator for BMAL1-CLOCK transcriptional complex is better supported with 1) new microscopic data from immunofluorescence and PLA analyses for Vps15, BAML1 and RNA Pol2, which showed co-localization of these proteins in the nucleus in MEFs and primary hepatocytes (Figure 9 to 11 in Rebuttal letter), 2) more detailed analysis of Vps15-BAML1 interaction with domain-mapping IPs (Fig 21 and 22), and 3) the rather comprehensive sets of E-Box driven luciferase reporter assays (Figure 20 and 23).
2. The argument that Vps15's co-activator function on BMAL1 does not rely on Vps34 kinase activity or the Vps15-Vps34 complex is investigated more in-depth: 1) one more Vps34 inhibitor was used and authors provided more thorough explanations for different phenotypes observed using different inhibitors, which can be due to non-overlapping off-target effects; 2) The Vps15 mutant that cannot bind to Vps34 can still show co-activator function and rescue the Cry1 suppression from the luciferase assays; 3) The observation that Vps15-BMAL1 PLA puncta was mainly in the nucleus, but the Vps34-BMAL1 PLA puncta was predominantly in the cytosol also highlight the functional interaction between Vps15 and BMAL1 independent of Vps34; 4) clearer characterization of nuclear levels of Vps15 in Vps34-null and Vps15-null cell lines and its interaction with BMAL1 in these different cellular fractions; and 5) the requirement of much higher ectopically expressed Vps34 to co-activate BMAL1-CLOCK compared to Vps15 .
3. At the same time, the authors included new discoveries that Vps34 also exhibited potential transcriptional regulation activity.
4. The authors demonstrated that Vps15 did not exert its transcriptional regulation function through modulating BMAL1-CLOCK complex formation, protein turnover or chromosome recruitment to target genes' promoter sites, but a layer of regulation of its co-activation on BMAL1 can come from controlling its active nuclear import in response to physiological cues, such as fasting (Figure 1 and 4 in rebuttal letter). The feeding-fasting experiments also provided a demonstration of the physiological relevance of the nuclear function of Vps15.

Overall, I commend the authors for addressing the comments well with more careful interpretation and accurate description/summary of data in this revised manuscript. Also, I find that the presentation of the data and organization of the written manuscript show significant improvement. There are a few minor issues that remain, listed below:

1. For the Vps34 kinase inhibitor studies, the authors explained that the discrepancy from different small molecules can be due to their non-overlapping off-target effects. From the rebuttal letter response to Reviewer 2, the authors listed the IC₅₀ values of these three inhibitors towards Vps34, which were all in the single-digit nM range (1-5 nM), while the IC₅₀ for other unwanted targets are in the low μ M range (1-10 μ M). It is therefore puzzling that the authors chose to use 2.5 μ M and 5 μ M in their studies to target Vps34, which is about a 1000-fold higher than the IC₅₀ for Vps34 and similar to IC₅₀ for other off-targets. The authors should find a lower concentration to use that these inhibitors can still inhibit Vps34 but will not target other kinases to potentially clear up the inhibitor data on the clock genes, or else provide a reason for the use of these high inhibitor concentrations.
2. The authors provided some explanation about the potential causes of the differences for BMAL1-CLOCK expression levels in the chronic Vps15 depletion model (Vps15LKO) vs. the other acute Vps15 depletion models (Vps15-null MEFs and Vps15iLKO) in the rebuttal letter (e.g. response to Reviewer 2 point 3 and Reviewer 4 point 1). It would be better if the authors could incorporate these ideas into the manuscript, either in the discussion or near line 170 where the clock phenotype differences in

Vps15-nulls MEFs and the chronic KO model is mentioned.

3. For statistical analysis, if more than two groups of data are compared (e.g., Extended figure 3b), instead of Student's t-test, one-way ANOVA should be used with appropriate post-hoc tests, such as Tukey or Sidak.

4. Extended data fig 3d: REV-ERBa showed 2 bands, which is different from other REV-ERBa blots. Authors should either mark the non-specific band with an asterisk or explain what the two bands are if they both correspond to REV-ERBa.

Reviewer #4:

Remarks to the Author:

In the revised manuscript " Class 3 PI3K participates in nuclear gene transcription and co-activates the circadian clock to promote de novo purine synthesis" (NCB-A48597A-Z), the authors have meticulously answered all the raised questions very satisfactorily.

Furthermore, the new experiments (nicely controlled with appropriate statistics) reveals critical insight. The revision has vastly improved the claims and makes significant contribution to the field.

I also agree with the authors that for the sake of simplicity of this very complex work, they might not include the observations on RORalpha interactions with Vps15 (Major Question1) in the final 'published' manuscript.

I have no further questions/comments for the authors.

GUIDELINES FOR SUBMISSION OF NATURE CELL BIOLOGY ARTICLES

ARTICLE FORMAT

ABSTRACT – should not exceed 150 words and should be unreferenced. This paragraph is the most visible part of the paper and should briefly outline the background and rationale for the work, and accurately summarize the main results and conclusions. Key genes, proteins and organisms should be specified to ensure discoverability of the paper in online searches.

TEXT – the main text consists of the Introduction, Results, and Discussion sections and must not exceed 3500 words including the abstract. The Introduction should expand on the background relating to the work. The Results should be divided in subsections with subheadings, and should provide a concise and accurate description of the experimental findings. The Discussion should expand on the findings and their implications. All relevant primary literature should be cited, in particular when discussing the background and specific findings.

REFERENCES – are limited to a total of 70 in the main text and Methods combined,. They must be numbered sequentially as they appear in the main text, tables and figure legends and Methods and must follow the precise style of Nature Cell Biology references. References only cited in the Methods should be numbered consecutively following the last reference cited in the main text. References only associated with Supplementary Information (e.g. in supplementary legends) do not count toward the total reference limit and do not need to be cited in numerical continuity with references in the main text. Only published papers can be cited, and each publication cited should be included in the numbered reference list, which should include the manuscript titles. Footnotes are not permitted.

Methods should be written concisely, but should contain all elements necessary to allow interpretation and replication of the results. As a guideline, Methods sections typically do not exceed 3,000 words.

The Methods should be divided into subsections listing reagents and techniques. When citing previous methods, accurate references should be provided and any alterations should be noted. Information must be provided about: antibody dilutions, company names, catalogue numbers and clone numbers for monoclonal antibodies; sequences of RNAi and cDNA probes/primers or company names and catalogue numbers if reagents are commercial; cell line names, sources and information on cell line identity and authentication. Animal studies and experiments involving human subjects must be reported in detail, identifying the committees approving the protocols. For studies involving human subjects/samples, a statement must be included confirming that informed consent was obtained. Statistical analyses and information on the reproducibility of experimental results should be provided in a section titled "Statistics and Reproducibility".

All Nature Cell Biology manuscripts submitted on or after March 21 2016, must include a Data availability statement as a separate section after Methods but before references, under the heading "Data Availability". For Springer Nature policies on data availability see <http://www.nature.com/authors/policies/availability.html>; for more information on this particular policy see <http://www.nature.com/authors/policies/data/data-availability-statements-data-citations.pdf>. The Data availability statement should include:

- Accession codes for primary datasets (generated during the study under consideration and designated as "primary accessions") and secondary datasets (published datasets reanalysed during the study under consideration, designated as "referenced accessions"). For primary accessions data should be made public to coincide with publication of the manuscript. A list of data types for which submission to community-endorsed public repositories is mandated (including sequence, structure, microarray, deep sequencing data) can be found here <http://www.nature.com/authors/policies/availability.html#data>.
- Unique identifiers (accession codes, DOIs or other unique persistent identifier) and hyperlinks for datasets deposited in an approved repository, but for which data deposition is not mandated (see here for details <http://www.nature.com/sdata/data-policies/repositories>).
- At a minimum, please include a statement confirming that all relevant data are available from the authors, and/or are included with the manuscript (e.g. as source data or supplementary information), listing which data are included (e.g. by figure panels and data types) and mentioning any restrictions on availability.
- If a dataset has a Digital Object Identifier (DOI) as its unique identifier, we strongly encourage including this in the Reference list and citing the dataset in the Methods.

We recommend that you upload the step-by-step protocols used in this manuscript to the Protocol Exchange. More details can be found at www.nature.com/protocolexchange/about.

DISPLAY ITEMS – main display items are limited to 6-8 main figures and/or main tables. For Supplementary Information see below.

FIGURES – Colour figure publication costs \$395 per colour figure. All panels of a multi-panel figure must be logically connected and arranged as they would appear in the final version. Unnecessary figures and figure panels should be avoided (e.g. data presented in small tables could be stated briefly

in the text instead).

All imaging data should be accompanied by scale bars, which should be defined in the legend. Cropped images of gels/blots are acceptable, but need to be accompanied by size markers, and to retain visible background signal within the linear range (i.e. should not be saturated). The boundaries of panels with low background have to be demarked with black lines. Splicing of panels should only be considered if unavoidable, and must be clearly marked on the figure, and noted in the legend with a statement on whether the samples were obtained and processed simultaneously. Quantitative comparisons between samples on different gels/blots are discouraged; if this is unavoidable, it has to be performed for samples derived from the same experiment with gels/blots were processed in parallel, which needs to be stated in the legend.

- For line art, graphs, charts and schematics we prefer Adobe Illustrator (.AI), Encapsulated PostScript (.EPS) or Portable Document Format (.PDF). Files should be saved or exported as such directly from the application in which they were made, to allow us to restyle them according to our journal house style.
- We accept PowerPoint (.PPT) files if they are fully editable. However, please refrain from adding PowerPoint graphical effects to objects, as this results in them outputting poor quality raster art. Text used for PowerPoint figures should be Helvetica (preferred) or Arial.
- We do not recommend using Adobe Photoshop for designing figures, but we can accept Photoshop generated (.PSD or .TIFF) files only if each element included in the figure (text, labels, pictures, graphs, arrows and scale bars) are on separate layers. All text should be editable in 'type layers' and line-art such as graphs and other simple schematics should be preserved and embedded within 'vector smart objects' - not flattened raster/bitmap graphics.
- Some programs can generate Postscript by 'printing to file' (found in the Print dialogue). If using an application not listed above, save the file in PostScript format or email our Art Editor, Allen Beattie for advice (a.beattie@nature.com).

Regardless of format, all figures must be vector graphic compatible files, not supplied in a flattened raster/bitmap graphics format, but should be fully editable, allowing us to highlight/copy/paste all text and move individual parts of the figures (i.e. arrows, lines, x and y axes, graphs, tick marks, scale bars etc). The only parts of the figure that should be in pixel raster/bitmap format are photographic images or 3D rendered graphics/complex technical illustrations.

Unprocessed scans of all key data generated through electrophoretic separation techniques need to be presented in a supplementary figure that should be labeled and numbered as the final supplementary figure, and should be mentioned in every relevant figure legend. This figure does not count towards the total number of figures and is the only figure that can be displayed over multiple pages, but should be provided as a single file, in PDF or TIFF format. Data in this figure can be displayed in a relatively informal style, but size markers and the figures panels corresponding to the presented data must be indicated.

The total number of Supplementary Figures (not including the “unprocessed scans” Supplementary Figure) should not exceed the number of main display items (figures and/or tables (see our Guide to Authors and March 2012 editorial <http://www.nature.com/ncb/authors/submit/index.html#suppinfo>; <http://www.nature.com/ncb/journal/v14/n3/index.html#ed>). No restrictions apply to Supplementary Tables or Videos, but we advise authors to be selective in including supplemental data.

GUIDELINES FOR EXPERIMENTAL AND STATISTICAL REPORTING

REPORTING REQUIREMENTS – We ask authors to complete a Reporting Summary that collects information on experimental design and reagents. We hope this will aid in your evaluation of the paper. The Reporting Summary can be found here <https://www.nature.com/documents/nr-reporting-summary.pdf>) Please note that these forms are dynamic 'smart pdfs' and must therefore be downloaded and completed in Adobe Reader. We will then flatten them for ease of use. If you would like to reference the guidance text as you complete the template, please access these flattened versions at <http://www.nature.com/authors/policies/availability.html>.

Author Rebuttal, second revision:

We thank the reviewers for their positive feedback regarding our revised manuscript. We are grateful for their helpful comments and suggestions that allowed us to reinforce further our initial findings on the unexpected crosstalk between class 3 PI3K and the circadian clock. In this revised version, we have made the corrections following the recommendations of the Reviewer 2 as well as answered the remaining questions of the Reviewer 3. We believe the additional information that we have provided below in point-by-point comments fully addressed the reviewer's questions.

The modifications to text file of the manuscript are highlighted in **green** to distinguish from the previous version.

Reviewers' Comments:

Reviewer 1:

The revised manuscript by Alkhoury et. al. has clearly addressed all of our recommended revisions. The revised manuscript has added clarity, specificity, and robustness to the work. The revised manuscript has revealed and expanded a novel role for class III PI3K in the cell nucleus and linked VPS34 and VPS15 to circadian transcriptional-translation oscillation regulation. We strongly recommend publication in NCB.

We thank reviewer for positive evaluation of our revised manuscript and for all the suggestions that have improved the presentation of our work.

Reviewer 2:

The revised manuscript does an excellent job of addressing prior concerns—this work describes the integration of two complicated pathways, and although some questions remain that will be addressed in future work, it provides a solid foundation to begin to understand co-transcriptional regulation of clock-controlled genes by subunits of the class III PI 3-kinase. In particular, new data describing a direct interaction between Vps15 and BMAL1 and activation of CLOCK: BMAL1 by Vps15 helps to root this a study in the larger framework of CLOCK: BMAL1 regulation. I think this work should be of interest to a broad audience and serve as a foundation for considerable future work by the field.

We thank the reviewer for all the guidance in this process and for such positive feedback on our revised manuscript.

I am satisfied with the revision with one major exception. The nomenclature describing PAS-A, PAS-B and PAC motifs is outdated and incorrect; the crystal structure of CLOCK: BMAL1 (Huang et al., 2012 Science) revealed unambiguous boundaries for the tandem PAS domains that represent the actual folded domains—the panel of BMAL1 deletion mutants that were reported in Tamaru et al., Plos Biol, 2009 refers to conserved bioinformatic motifs that represent only a portion of the PAS domains and is therefore misleading. In this context, the authors actually show that partial deletion of the PAS-A or PAS-B domains in BMAL1 disrupts binding to CLOCK. Although partial deletion of a domain is not ideal, these data are consistent with prior results demonstrating the importance of the PAS domains for heterodimerization of the transcription factor. BMAL1 fragments from this

manuscript: CLOCK: gray BMAL1: 1-134 (blue, disordered N-terminus and bHLH); 135-265 (orange, part of PAS-A); 266-399 (yellow, part of PAS-A and part of PAS-B); and 400-506 (green, part of PAS-B up to residue 442 and some of the disordered C-terminus)

The critical data for this manuscript relies on deletion of the BMAL1 C-terminus, which corresponds to the transcriptional activation domain, or TAD, of BMAL1, so it is not affected by these partial domain deletions. Moreover, the observed requirement of the TAD for Vps15 binding is consistent with its role in transcriptional activation of CLOCK: BMAL1 as described here.

To correct this, the authors should represent the proper domain boundaries in the cartoon (alongside their dashed lines for the partial PAS domain deletions) in Figure 4 and briefly describe this in the text to avoid perpetuating an incorrect description of the PAS domains. Based on the crystal structure PDB 4F3L, the domain boundaries of mouse PAS-A are residues 157-321 and PAS-B are 338-442. The authors may also want to give consideration for red-green coloring in figures for color vision deficient readers.

We thank the reviewer for detailed explanations and for the suggestions on figure presentation that we fully took on board. We have prepared a new figure that shows BMAL1 protein domain boundaries as indicated in PDB database for the PDBsum entry 4f3l (<https://www.ebi.ac.uk/thornton-srv/databases/cgi-bin/pdbsum/GetPage.pl?pdbcode=4f3l&template=protein.html&r=wiring&l=1&chain=B>). We also modified the color code to improve figure readability (**R-Fig.1**, corresponding to **Fig.4i** in the revised manuscript).

R-Fig.1. The domain organization of mouse BMAL1 protein (top panel) and its truncated mutants that were used for co-immunoprecipitation analyses. The amino acid positions of deletions in BMAL1 protein are indicated at the bottom.

Reviewer #3:

In this revised manuscript, the authors provided a significant amount of high-quality new data to support their main findings for a novel role of class 3 PI3K in the nucleus and their proposed mechanistic model, where the nuclear pool of Vps15 serves as a transcriptional co-activator of the BMAL1-CLOCK complex to couple energy homeostasis regulation with circadian clock transcriptional activity. Importantly, in the revised manuscript, the authors presented much more solid support for multiple major findings compared to the original version:

1. The role of Vps15 as a co-activator for BMAL1-CLOCK transcriptional complex is better supported with 1) new microscopic data from immunofluorescence and PLA analyses for Vps15, BAML1 and RNA Pol2, which showed co-localization of these proteins in the nucleus in MEFs and primary hepatocytes (Figure 9 to 11 in Rebuttal letter), 2) more detailed analysis of Vps15-BAML1 interaction with domain-mapping IPs (Fig 21 and 22), and 3) the rather comprehensive sets of E-Box driven luciferase reporter assays (Figure 20 and 23).

2. The argument that Vps15's co-activator function on BMAL1 does not rely on Vps34 kinase activity or the Vps15-Vps34 complex is investigated more in-depth: 1) one more Vps34 inhibitor was used and authors provided more thorough explanations for different phenotypes observed using different inhibitors, which can be due to non-overlapping off-target effects; 2) The Vps15 mutant that cannot bind to Vps34 can still show co-activator function and rescue the Cry1 suppression from the luciferase assays; 3) The observation that Vps15-BMAL1 PLA puncta was mainly in the nucleus, but the Vps34-BMAL1 PLA puncta was predominantly in the cytosol also highlight the functional interaction between Vps15 and BMAL1 independent of Vps34; 4) clearer characterization of nuclear levels of Vps15 in Vps34-null and Vps15-null cell lines and its interaction with BMAL1 in these different cellular fractions; and 5) the requirement of much higher ectopically expressed Vps34 to co-activate BMAL1-CLOCK compared to Vps15 .

3. At the same time, the authors included new discoveries that Vps34 also exhibited potential transcriptional regulation activity.

4. The authors demonstrated that Vps15 did not exert its transcriptional regulation function through modulating BMAL1-CLOCK complex formation, protein turnover or chromosome recruitment to target genes' promoter sites, but a layer of regulation of its co-activation on BMAL1 can come from controlling its active nuclear import in response to physiological cues, such as fasting (Figure 1 and 4 in rebuttal letter). The feeding-fasting experiments also provided a demonstration of the physiological relevance of the nuclear function of Vps15.

Overall, I commend the authors for addressing the comments well with more careful interpretation and accurate description/summary of data in this revised manuscript. Also, I find that the presentation of the data and organization of the written manuscript show significant improvement.

We thank the reviewer for positive evaluation of our revision efforts. We would like to thank again for the in-depth insightful suggestions of the reviewer that allowed us to ameliorate our work.

There are a few minor issues that remain, listed below:

1. For the Vps34 kinase inhibitor studies, the authors explained that the discrepancy from different small molecules can be due to their non-overlapping off-target effects. From the rebuttal letter response to Reviewer 2, the authors listed the IC50 values of these three inhibitors towards Vps34, which were all in the single-digit nM range (1-5 nM), while the IC50 for other unwanted targets are in the low μ M range (1-10 μ M). It is therefore puzzling that the authors chose to use 2.5 μ M and 5 μ M in their studies to target Vps34, which is about a 1000-fold higher than the IC50 for Vps34 and similar to IC50 for other off-targets. The authors should find a lower concentration to use that these inhibitors can still inhibit Vps34 but will not target other kinases to potentially clear up the inhibitor data on the clock genes, or else provide a reason for the use of these high inhibitor concentrations.

We apologize for not being sufficiently clear in our explanations for this important point. In three original reports on the selective inhibitors of Vps34 that we have cited and used in our study, the IC50 were reported either for purified recombinant protein following *in vitro* Vps34 lipid kinase assay (PIKIII and Vps34-IN1)^{1,2} or following the high-content image-based screening to identify the autophagy inhibitors that was also validated with recombinant Vps34 protein (SAR405)³. As expected, the effective doses of these inhibitors in cells are considerably higher (the micromolar range) compared to the doses used *in vitro*. For example, in the original publication and in the follow-up studies (and as we have employed in our work), PIKIII was used in different cell lines in doses of 2.5 μ M, 5 μ M and 10 μ M¹. Similarly, for SAR405, its effect on autophagy inhibition was tested in a dose course in different cells showing marked effect on p62 accumulation, lysosomal mass and autophagy block in doses between 1-10 μ M (HeLa, H1299, RKO cells)³. Notably, for SAR405, potential off-targets were suggested by KiNativ chemoproteomics (ActivX

Biosciences) in Jurkat cells treated with 1 μ M dose of the inhibitor (Supplementary Data Set in the original publication³). Furthermore, the Vps34-IN1 was used in 1 μ M dose in cells in the original publication but the autophagy was not assessed². However, similar to our study, Vps34-IN1 was used in 1-10 μ M doses in cells to inhibit Vps34⁴⁻⁶. Finally, when choosing the dose of the inhibitors, we also took into consideration the long-term incubation with the compound post synchronization and, thus, we opted for low micromolar range (2.5 μ M, 5 μ M and 10 μ M) to achieve a persistent inhibition of Vps34 without noticeable toxicity (fluorescent microscopy analyses presented in *Extended Figure 2c* and *3c* showing comparable cell shape and nuclei shape while decreased PI3P labelling).

2. The authors provided some explanation about the potential causes of the differences for BMAL1-CLOCK expression levels in the chronic Vps15 depletion model (Vps15LKO) vs. the other acute Vps15 depletion models (Vps15-null MEFs and Vps15iLKO) in the rebuttal letter (e.g. response to Reviewer 2 point 3 and Reviewer 4 point 1). It would be better if the authors could incorporate these ideas into the manuscript, either in the discussion or near line 170 where the clock phenotype differences in Vps15-nulls MEFs and the chronic KO model is mentioned.

As suggested by the reviewer, we have made brief text additions to the manuscript (*line 173-177*).

3. For statistical analysis, if more than two groups of data are compared (e.g., Extended figure 3b), instead of Student's t-test, one-way ANOVA should be used with appropriate post-hoc tests, such as Tukey or Sidak.

We gratefully thank the reviewer for bringing it to our attention. In this revised version, we carefully reviewed statistical analyses and changed t-test to one-way or two-way ANOVA following a post hoc test where appropriate (e.g. *Extended Figure 3b*). Further, figure legends and material and methods were revised to provide the information about the statistics applied.

4. Extended data fig 3d: REV-ERBa showed 2 bands, which is different from other REV-ERBa blots. Authors should either mark the non-specific band with an asterisk or explain what the two bands are if they both correspond to REV-ERBa.

The reviewer is right, the bottom band in the immunoblot panels of Rev-Erb α in *Extended Data Fig. 3d* is a non-specific band that is detected with this antibody and on most of the WB panels it is well separated from the specific band as could be seen on uncropped WB panels provided in Source Data file. We have added the “ns” next to the non-specific band in the Rev-Erb α panel in *Extended Data Fig. 3d* to avoid confusion.

References:

- 1 Dowdle, W. E. *et al.* Selective VPS34 inhibitor blocks autophagy and uncovers a role for NCOA4 in ferritin degradation and iron homeostasis in vivo. *Nat Cell Biol* **16**, 1069-1079, doi:10.1038/ncb3053 (2014).
- 2 Bago, R. *et al.* Characterization of VPS34-IN1, a selective inhibitor of Vps34, reveals that the phosphatidylinositol 3-phosphate-binding SGK3 protein kinase is a downstream target of class III phosphoinositide 3-kinase. *The Biochemical journal* **463**, 413-427, doi:10.1042/BJ20140889 (2014).

- 3 Ronan, B. *et al.* A highly potent and selective Vps34 inhibitor alters vesicle trafficking and autophagy. *Nat Chem Biol* **10**, 1013-1019, doi:10.1038/nchembio.1681 (2014).
- 4 Munson, M. J. *et al.* mTOR activates the VPS34-UVRAG complex to regulate autolysosomal tubulation and cell survival. *EMBO J* **34**, 2272-2290, doi:10.15252/embj.201590992 (2015).
- 5 Williams, C. G. *et al.* Inhibitors of VPS34 and fatty-acid metabolism suppress SARS-CoV-2 replication. *Cell reports* **36**, 109479, doi:10.1016/j.celrep.2021.109479 (2021).
- 6 Jang, W. *et al.* Endosomal lipid signaling reshapes the endoplasmic reticulum to control mitochondrial function. *Science* **378**, eabq5209, doi:10.1126/science.abq5209 (2022).

Reviewer #4:

In the revised manuscript " Class 3 PI3K participates in nuclear gene transcription and co-activates the circadian clock to promote de novo purine synthesis" (NCB-A48597A-Z), the authors have meticulously answered all the raised questions very satisfactorily.

Furthermore, the new experiments (nicely controlled with appropriate statistics) reveals critical insight. The revision has vastly improved the claims and makes significant contribution to the field.

I also agree with the authors that for the sake of simplicity of this very complex work, they might not include the observations on RORalpha interactions with Vps15 (Major Question1) in the final 'published' manuscript.

I have no further questions/comments for the authors.

We are grateful to the reviewer for their input and for such positive evaluation of our revised manuscript.

Decision Letter, second revision:

Our ref: NCB-A48597B

17th April 2023

Dear Dr. Panasyuk,

Thank you very much for submitting your revised manuscript "Class 3 PI3K participates in nuclear gene transcription and co-activates the circadian clock to promote de novo purine synthesis" (NCB-A48597B) and thank you for revising the study again. We have editorially assessed the revisions and appreciate all the clarifications and changes made in response to Reviewers #2 and #3. Therefore, we'll be happy in principle to publish the manuscript in Nature Cell Biology, pending minor revisions to comply with our editorial and formatting guidelines and to edit the text as follows. We appreciated your clarification regarding the definition of PAS-A and PAS-B BMAL-1 domains as explained in the paper. However, given that the PDB and the review you cited from 2015 <https://pubs.acs.org/doi/10.1021/bi500731f> provide yet distinct definitions for these domain boundaries, we think it is important that the manuscript note this in 1-2 sentences, explaining all the potential boundaries put forth, citing the Review you shared with us in your response, and stressing, as Rev#2 had indicated, that the mutants you generated had partial PAS domain deletions and that your data are consistent with the literature demonstrating the importance of the PAS domains for heterodimerization of the transcription factor. We greatly appreciate your efforts to edit the manuscript one more time. Please let us know if you wish to discuss further or have any questions.

****Please note that the current version of your manuscript is in a PDF format. Could you please email us a copy of the file in an editable format (Microsoft Word or LaTeX), as we can not proceed with PDFs at this stage? Many thanks for your attention to this point.****

Once we have the Word file, we will begin performing detailed checks on your paper and will send you a checklist detailing our editorial and formatting requirements in about 1-2 weeks. Please do not upload the final materials and make any revisions until you receive this additional information from us.

Thank you again for your interest in Nature Cell Biology. Please do not hesitate to contact me if you have any questions.

Sincerely,

Melina

Melina Casadio, PhD
Senior Editor, Nature Cell Biology
ORCID ID: <https://orcid.org/0000-0003-2389-2243>

Decision Letter, final checks

Our ref: NCB-A48597B

25th April 2023

Dear Dr. Panasyuk,

Thank you for your patience as we've prepared the guidelines for final submission of your Nature Cell Biology manuscript, "Class 3 PI3K participates in nuclear gene transcription and co-activates the circadian clock to promote de novo purine synthesis" (NCB-A48597B). Please carefully follow the step-by-step instructions provided in the attached file, and add a response in each row of the table to indicate the changes that you have made. Please also check and comment on any additional marked-up edits we have proposed within the text. Ensuring that each point is addressed will help to ensure that your revised manuscript can be swiftly handed over to our production team.

In recognition of the time and expertise our reviewers provide to Nature Cell Biology's editorial process, we would like to formally acknowledge their contribution to the external peer review of your manuscript entitled "Class 3 PI3K participates in nuclear gene transcription and co-activates the circadian clock to promote de novo purine synthesis". For those reviewers who give their assent, we will be publishing their names alongside the published article.

Nature Cell Biology offers a Transparent Peer Review option for new original research manuscripts submitted after December 1st, 2019. As part of this initiative, we encourage our authors to support increased transparency into the peer review process by agreeing to have the reviewer comments, author rebuttal letters, and editorial decision letters published as a Supplementary item. When you submit your final files please clearly state in your cover letter whether or not you would like to participate in this initiative. Please note that failure to state your preference will result in delays in accepting your manuscript for publication.

Cover suggestions

As you prepare your final files we encourage you to consider whether you have any images or illustrations that may be appropriate for use on the cover of Nature Cell Biology.

Nature Cell Biology has now transitioned to a unified Rights Collection system which will allow our Author Services team to quickly and easily collect the rights and permissions required to publish your work. Approximately 10 days after your paper is formally accepted, you will receive an email in providing you with a link to complete the grant of rights. If your paper is eligible for Open Access, our Author Services team will also be in touch regarding any additional information that may be required to arrange payment for your article.

Please note that *Nature Cell Biology* is a Transformative Journal (TJ). Authors may publish their research with us through the traditional subscription access route or make their paper immediately open access through payment of an article-processing charge (APC). Authors will not be required to make a final decision about access to their article until it has been accepted. Find out more about Transformative Journals

Please use the following link for uploading these materials:
[Redacted]

Best regards,

Kendra Donahue
Staff
Nature Cell Biology

On behalf of

Melina Casadio, PhD
Senior Editor, Nature Cell Biology
ORCID ID: <https://orcid.org/0000-0003-2389-2243>

Final Decision Letter:

Dear Dr Panasyuk,

I am pleased to inform you that your manuscript, "Class 3 PI3K coactivates the circadian clock to promote rhythmic de novo purine synthesis", has now been accepted for publication in *Nature Cell Biology*. Congratulations on this beautiful study!

Over the next few weeks, your paper will be copyedited to ensure that it conforms to *Nature Cell Biology* style. Once your paper is typeset, you will receive an email with a link to choose the appropriate publishing options for your paper and our Author Services team will be in touch regarding any additional information that may be required.

Publication is conditional on the manuscript not being published elsewhere and on there being no announcement of this work to any media outlet until the online publication date in *Nature Cell Biology*.

Please note that *Nature Cell Biology* is a Transformative Journal (TJ). Authors may publish their research with us through the traditional subscription access route or make their paper immediately open access through payment of an article-processing charge (APC). Authors will not be required to make a final decision about access to their article until it has been accepted. Find out more about Transformative Journals

Authors may need to take specific actions to achieve compliance with funder and institutional open access mandates. If your research is supported by a funder that requires immediate open access (e.g. according to Plan S principles) then you should select the gold OA route, and we will direct you to the compliant route where possible. For authors selecting the subscription publication route, the journal's standard licensing terms will need to be accepted, including self-archiving policies. Those licensing terms will supersede any other terms that the author or any third

party may assert apply to any version of the manuscript.

If you have not already done so, we strongly recommend that you upload the step-by-step protocols used in this manuscript to the Protocol Exchange (www.nature.com/protocolexchange), an open online resource established by Nature Protocols that allows researchers to share their detailed experimental know-how. All uploaded protocols are made freely available, assigned DOIs for ease of citation and are fully searchable through nature.com. Protocols and Nature Portfolio journal papers in which they are used can be linked to one another, and this link is clearly and prominently visible in the online versions of both papers. Authors who performed the specific experiments can act as primary authors for the Protocol as they will be best placed to share the methodology details, but the Corresponding Author of the present research paper should be included as one of the authors. By uploading your Protocols to Protocol Exchange, you are enabling researchers to more readily reproduce or adapt the methodology you use, as well as increasing the visibility of your protocols and papers. You can also establish a dedicated page to collect your lab Protocols. Further information can be found at www.nature.com/protocolexchange/about

With kind regards,

Melina

Melina Casadio, PhD
Senior Editor, Nature Cell Biology
ORCID ID: <https://orcid.org/0000-0003-2389-2243>
